# Evolution-based mathematical models significantly prolong response to abiraterone in metastatic castrate-resistant prostate cancer and identify strategies to further improve outcomes

**Jingsong Zhang[1], Jessica Cunningham[2†], Joel Brown[2,3], Robert Gatenby[2,4]\***

[1]Department of Genitourinary Oncology, Moffitt Cancer Center and Research Institute, Tampa, United States; [2]Department of Integrated Mathematical Oncology, Moffitt Cancer Center and Research Institute, Tampa, United States; [3]Department of Biological Sciences, University of Illinois at Chicago, Chicago, United States; [4]Cancer Biology and Evolution Program, Moffitt Cancer Center and Research Institute, Tampa, United States

**\*For correspondence:**
Robert.Gatenby@Moffitt.org

**Present address:** †Fralin Biomedical Research Institute, Roanoke, United States

## Abstract

**Background:**
Abiraterone acetate is an effective treatment for metastatic castrate-resistant prostate cancer (mCRPC), but evolution of resistance inevitably leads to progression. We present a pilot study in which abiraterone dosing is guided by evolution-informed mathematical models to delay onset of resistance.

**Methods:**
In the study cohort, abiraterone was stopped when PSA was <50% of pretreatment value and resumed when PSA returned to baseline. Results are compared to a contemporaneous cohort who had >50% PSA decline after initial abiraterone administration and met trial eligibility requirements but chose standard of care (SOC) dosing.

**Results:** 17 subjects were enrolled in the adaptive therapy group and 16 in the SOC group. All SOC subjects have progressed, but four patients in the study cohort remain stably cycling (range 53–70 months). The study cohort had significantly improved median time to progression (TTP; 33.5 months; p<0.001) and median overall survival (OS; 58.5 months; hazard ratio, 0.41, 95% confidence interval (CI), 0.20–0.83, p<0.001) compared to 14.3 and 31.3 months in the SOC cohort. On average, study subjects received no abiraterone during 46% of time on trial. Longitudinal trial data demonstrated the competition coefficient ratio ($\alpha_{RS}/\alpha_{SR}$) of sensitive and resistant populations, a critical factor in intratumoral evolution, was two- to threefold higher than pre-trial estimates. Computer simulations of intratumoral evolutionary dynamics in the four long-term survivors found that, due to the larger value for $\alpha_{RS}/\alpha_{SR}$, cycled therapy significantly decreased the resistant population. Simulations in subjects who progressed predicted further increases in OS could be achieved with prompt abiraterone withdrawal after achieving 50% PSA reduction.

**Conclusions:** Incorporation of evolution-based mathematical models into abiraterone monotherapy for mCRPC significantly increases TTP and OS. Computer simulations with updated parameters from longitudinal trial data can estimate intratumoral evolutionary dynamics in each subject and identify strategies to improve outcomes.

**Funding:** Moffitt internal grants and NIH/NCI U54CA143970-05 (Physical Science Oncology Network).

## Editor's evaluation

Zhang et al. use evolution-guided mathematical models to guide the timing and dosing of abiraterone treatment in castrate-resistant prostate cancer. While the sample size is limited, the implications of the study outcome are broad and compelling, and the article importantly highlights the transformative potential of deeply interdisciplinary research.

## Introduction

While often initially effective, nearly all cancer treatments ultimately fail due to evolution of resistance (*Sarmento-Ribeiro et al., 2019*). Prior efforts to disrupt the molecular machinery of resistance (such as P-glycoprotein during chemotherapy administration) have led to small or no improvement in outcomes (*Vasan et al., 2019*; *Wang et al., 2019*), indicating that tumor cells generally have multiple available mechanisms of resistance. However, resistant tumors, regardless of the precise mechanism, require both deployment of molecular mechanisms of resistance and proliferation of surviving (i.e., resistant) populations to become clinically significant (*Gatenby et al., 2009a*; *Reed et al., 2020*). We hypothesize the former is an inevitable response to treatment, but the latter is governed by eco-evolutionary principles and potentially manageable by Darwinian controls (*McKee, 2010*). Evolution-based models show that current treatment strategies, which apply therapy at maximum tolerated dose until progression, are often not evolutionarily optimal. While an initial response may be large, therapy fails because it strongly selects for resistance while eradicating all treatment-sensitive cells. The resistant cells are now free from competition with sensitive cells – an evolutionary dynamic termed 'competitive release' - which allows for rapid proliferation (*Silva et al., 2012*; *Newton and Ma, 2019*).

One evolutionary strategy to delay population growth of the resistant phenotype, termed 'adaptive therapy' (*Gatenby et al., 2009b*), exploits the fitness costs incurred by production, maintenance, and operation of the molecular machinery required for treatment resistance (*Szakács et al., 2014*). These resource demands are compensated for by increased fitness when treatment is applied. However, in the absence of treatment, the phenotypic costs reduce fitness compared to competing-sensitive cells (*Szakács et al., 2014*), particularly in resource-limited tumor microenvironments. Thus, when multiple mechanisms of resistance are available, the precise cost will likely vary but seldom will there be *no* associated cost. In nature, similar cost/benefit trade-offs are seen in, for example, loss of eyes by cavefish as the resource costs of producing and maintaining them are balanced against their lack of utility in a continuously dark environment (*Gatenby et al., 2011*). Population dynamics dependent on the cost of resistance are now fundamental principles in pest management (*Ehler, 2006*) and have been observed experimentally in cancer populations (*Enriquez-Navas et al., 2016*).

The phenotypic cost of resistance can be explicitly measured in, for example, membrane extrusion pumps (*Shen et al., 2008*). However, there are no data regarding the molecular dynamics leading to abiraterone resistance in metastatic castrate-resistant prostate cancer (mCRPC). Furthermore, the precise mechanism of resistance may vary (*Pal et al., 2018*). When such measurements cannot be obtained, census methods developed in ecology (*Pfister, 1995*) can estimate relative fitness based on population distributions. That is, if abiraterone-sensitive cells are the dominant tumor subpopulation prior to treatment, they must be fitter than the resistant phenotypes in the absence of treatment even if the specific reason for this fitness difference is not known.

Adaptive therapy (*Gatenby et al., 2009b*; *Zhang et al., 2017*) limits application of treatment to produce a moderate decrease in tumor volume while explicitly retaining a significant population of treatment-sensitive cancer cells. Following an initial response, treatment is withdrawn, allowing the cancer populations to proliferate. But, in the absence of selection pressures from treatment, the sensitive cells have a fitness advantage and will proliferate at the expense of the resistant cells. Thus, when the tumor returns to its pretreatment volume, the subpopulation distribution is similar, allowing the initial therapy to remain effective.

Because adaptive therapy starts and stops treatment, it is conceptually similar to 'intermittent therapy' trials in which men with metastatic castration-sensitive prostate cancer (mCSPC) were

randomized into continuous or intermittent androgen deprivation therapy (ADT) treatment using gonadatropin releasing hormone (GnRH) analogs. Neither regimen proved superior (*Hussain et al., 2013*). Although trial design has similarities to the adaptive therapy protocol, we note that the fundamental strategy of evolution-based treatment uses cycling of treatment-sensitive cells as a forcing function to control the smaller resistant population. However, in the intermittent therapy trial, treatment cycling began only after 8 months of 'induction therapy' with ADT. Furthermore, intermittent dosing was permitted only if the PSA reached <4 ng/ml. With computer simulations (*Cunningham, 2019*; *Cunningham et al., 2018*), we have demonstrated that the prolonged induction period consistently reduced the sensitive population to near-extinction levels, as confirmed by the reduction of PSA to the normal range. Since adaptive therapy relies on the presence of a sensitive cell that is initially larger than the resistant cells, the induction therapy rendered these evolutionary dynamics impossible. Thus, computer simulations predicted that this strategy would produce tumor control identical to standard continuous full dose of ADT, which was the observed outcomes with the intermittent arm indistinguishable from continuous therapy. Similar limitations apply to a prior study that used intermittent ADT with fixed 8-month intervals (*Crook et al., 2012*). Model simulations showed that the intervals were too long and promoted the dominance of resistant cancer cells.

These analyses illustrate the critical role for inclusion of mathematical models, evolutionary first principles, and computer simulations in trial design. As complex adaptive systems (*Uthamacumaran, 2021*), cancers frequently exhibit nonlinear dynamics that cannot be predicted intuitively but can be captured using mathematical models. Here, we present such a trial. Computer simulations of the model were used to predict optimal trial design. Later, the same models could be evaluated and parameterized to patient-specific data, allowing for novel approaches to trial analysis in which longitudinal trial data and observed outcomes are used to update pretreatment parameter estimates. Simulations using the updated model can then be applied to each patient in the trial to analyze intratumoral evolutionary dynamics during treatment. Unlike conventional clinical trials, this approach allows both cohort and patient-specific analyses. Furthermore, the simulations critique trial design and performance, thus providing guidance for alternative strategies and future investigations.

Thus, we hypothesize that formally integrating evolutionary dynamics into abiraterone treatment will delay proliferation of resistant cells prolonging time to progression (TTP). Our trial objectives were twofold: (1) test the underlying hypothesis in a small patient cohort and (2) investigate our novel trial design in which the treatment protocol is based on a mathematical model and analyzed through an iterative process in which trial data informs model parameter estimates and computer simulations from the updated model investigate intratumoral evolutionary dynamics in each trial patient.

Here, we provide follow-up on an initial report submitted when the benefits of adaptive therapy compared to standard of care (SOC) treatment achieved statistical significance (*Zhang et al., 2017*). However, at that time, we could only demonstrate the median TTP was >27 months. We confirm the superiority of adaptive therapy over SOC. Additionally, we demonstrate how our multidisciplinary approach to treatment design and analysis provides novel patient-specific information that may reduce the need for large, expensive clinical trials. Finally, the results of this iterative approach can be used to design follow-up clinical treatment plans to further improve outcomes.

## Methods
### Pilot clinical trial

This is a single-institution investigator-initiated pilot study (NCT02415621) carried out at the Moffitt Cancer Center, Tampa, FL. The protocol was approved by central IRB and monitored by Moffitt Cancer Center's protocol monitoring committee. Details of the trial design have been previously published (*Zhang et al., 2017*). Briefly, inclusion criteria were similar to phase III AA-302 trial (*Cunningham, 2019*) population, except allowing ECOG 2 performance status (PFD), prior exposure to enzalutamide, sipuleucel-T, and ketoconazole. Prior docetaxel was allowed if it was given during the castration-sensitive phase. Patients on opioids for cancer-related pain were excluded. Patients could be enrolled in the study after achieving 50% or more decline of their pre-abiraterone Prostate Specific Antigen (PSA) levels. Cohort size was designed to have sufficient statistical power to detect a 50% increase in TTP.

Each enrolled patient began on abiraterone (1000 mg by mouth daily) and prednisone (5 mg by mouth twice daily) until achieving a >50% decline in their baseline levels of PSA pre-abiraterone.

Upon achieving this decline, abiraterone therapy was suspended. Tumor regression or stability was confirmed by radiographic measurements.

Patients were monitored every 4–6 weeks with a lab (Complete Blood Count (CBC), Comprehensive Metabilic Panel (COMP), Lactic Dehydrogenase (LDH), and PSA) and clinic visit. Serum testosterone was not measured. Every 12 weeks, each patient received a bone scan, and a computed tomography (CT) of the abdomen and pelvis. Abiraterone plus prednisone were reinitiated when a patient's PSA increased to or above the pre-abiraterone PSA baseline. Abiraterone therapy was stopped again after the patient's PSA declined to >50% of his baseline PSA. Each successive peak of PSA when abiraterone therapy was reinstated defined a complete cycle of adaptive therapy.

For patients who did not undergo surgical castration, GnRH analog treatment was continued to maintain castration levels of serum testosterone. Patients who did not achieve a 50% decline of their baseline PSA after restarting abiraterone remained in study until they developed radiographic progression based on Prostate Cancer Working Group 2 criteria. Patients who developed radiographic progression while off abiraterone would restart abiraterone and remain on abiraterone until a partial response was noted in the repeat bone scan, and abdominal and pelvic CT. These subjects were then allowed to stop abiraterone and reenter the adaptive therapy cycles. Patients were followed until they developed radiographic progression or ECOG performance status deterioration while on abiraterone, whichever came first.

## SOC cohort

Sixteen patients who were treated with continuous abiraterone as the SOC and met the eligibility criteria for our adaptive therapy were identified through chart review of mCRPC patients treated at the Moffitt Cancer Center during the time of the study enrollment. Thus, all patients fulfilled trial eligibility requirements (including a >50% drop in PSA) and chose SOC treatment. Specifically, all patients in this group had a >50% decline in PSA following initial administration of abiraterone (Appendix 1) and a prior therapy history that met eligibility requirements for the adaptive trial.

## Mathematical models used in trial design and analysis

Our original mathematical model (*Zhang et al., 2017*) divided the prostate cancer subpopulations based on their interactions with testosterone: T+ cells require extrinsic androgen for survival and proliferation, TP cells require androgen for survival and proliferation but upregulated CYP17a1 (*Mostaghel et al., 2011*) allow them to produce testosterone generating an autocrine loop, T-cells are androgen independent. Note the potential coupling of TP and T+ cells as the testosterone produced by the TP cells represent a 'common good' or 'public good' (*Johnstone and Rodrigues, 2016*) that can be used by the T+ cells. In evolutionary terms, this coupling results in the T+ cells acting as 'cheaters' (*Ghoul et al., 2014*) because they use the testosterone produced by the TP cells but do not incur the fitness cost of producing it. Here, for our post-trial analysis, we combine the T+ and TP cells as both being sensitive to abiraterone. This simplifies the model into a sensitive population (T+ and TP) and a population resistant to abiraterone (T-). The model is available in GitHub (*Cunningham, 2022*).

As in our original report (*Zhang et al., 2017*), we use Lotka–Volterra (LV) competition equations to model the interactions between the three (prior model) and two (current analysis) cell types ($x_j$ based on parameters for intrinsic growth rates, $r_i$, carrying capacities, $K_i$, and the matrix of competition coefficients, $a_{ij}$.

$$\frac{dx_i}{dt} = r_i x_i \left( 1 - \frac{\sum_{j=1}^{3} a_{ij} x_j}{K_i} \right) \tag{1}$$

Since patients are included in the study only if their PSA declined by at least 50% after initial application of abiraterone, the sensitive cells must be more prevalent than the resistant cells. Based on census methodologies (*Pfister, 1995*) and the steep drop in PSA with therapy, we can conclude that the sensitive cells are fitter than the T- cells prior to treatment. We assume that each phenotype produces roughly equal amounts of PSA, which may introduce error if this assumption is substantially violated. This assumption does allow us to consider PSA as a direct estimator of the total number of cancer cells.

**Table 1.** Demographic and prior treatment history in each cohort.
The study was conducted before abiraterone, enzalutamide, or apalutamide was approved for treating castration-sensitive prostate cancer. Sipuleucel-T was the only treatment given before abiraterone for metastatic castrate-resistant prostate cancer (mCRPC) in the control and adaptive abiraterone cohorts.

|  | Control (Pfister, 1995) | Adaptive abiraterone (Zhang et al., 2017) |
|---|---|---|
| Age/mean [range] | 68 [57–76] | 67 [50–79] |
| History of androgen deprivation therapy for M0 prostate cancer | 5 | 7 |
| <12 months of androgen deprivation therapy prior to abiraterone for mCRPC | 3 | 3 |
| Sipuleucel-T prior to abiraterone | 5 | 6 |
| Gleason score/median [range] | 7 [7, 10] | 8 [6, 10] |
| Pre-abiraterone PSA/mean [range] | 36.52 [2.71, 93.4] | 29.7 [1.46, 109.4] |
| Lymph node metastases only | 1 | 1 |
| Bone, with or without lymph node metastases | 14 | 15 |
| Lung or soft tissue metastases | 1 | 1 |

Each competition coefficient ($a_{ij}$) standardizes the competitive effect of an individual of type $j$ on the per capita growth rate of type $i$ in units of type $i$. In general, the value of the competition coefficient reflects the relative fitness of the populations. All $a_{ii} = 1$. If $a_{ij} > 1$, then inter-type competition is greater than intra-type; and vice versa if $a_{ij} < 1$.

As in our original model, we let abiraterone therapy reduce the carrying capacities of the TP and T+ cells, with no effect on T-. We assume that TP cells are either killed or quiescent during abiraterone treatment. Since abiraterone inhibits the production of testosterone by the TP cells, the T+ 'cheater' population will have no source of testosterone and they too decline with abiraterone, rendering both sensitive to abiraterone both directly and indirectly.

## Statistics

To compare patient characteristics between the trial and SOC cohorts, we used Kruskal–Wallis nonparametric one-way ANOVAs (done with SYSTAT13). To compare progression-free survivorship between the two cohorts. we used the Mantel logrank test (done with SYSTAT13).

## Results

### Cohort analysis

Seventeen evaluable patients were enrolled between June 2015 and January 2019. Tumor stage, initial Gleason Scores, and pretreatment PSA values were not significant between both trial and SOC cohorts (*Table 1*): Gleason scores: Kruskal–Wallis test statistic of 0.088 with p=0.767 based on a chi-square distribution, df = 1. Pretreatment PSA levels: Kruskal–Wallis test statistic of 0.157 with p=0.692 based on a chi-square distribution, df = 1. All patients fulfilled trial eligibility so that pretreatment history was identical.

This study was conducted before abiraterone was approved in the castration-sensitive setting. None of the patients enrolled in the adaptive therapy trial or included in the historical control had received new hormonal agent (abiraterone, enzalutamide, or apalutamide) or docetaxel in the castration-sensitive setting. Abiraterone was the frontline therapy for mCRPC for most patients in each group.

Given patients enrolled in the study had more frequent PSA checks than the historical control arm, more patients in the study cohort had >50% PSA reduction within a month. The percentage of patients who had more than 50% PSA reduction within 2 months was similar: 15/17 (88%) vs. 12/16 (75%).

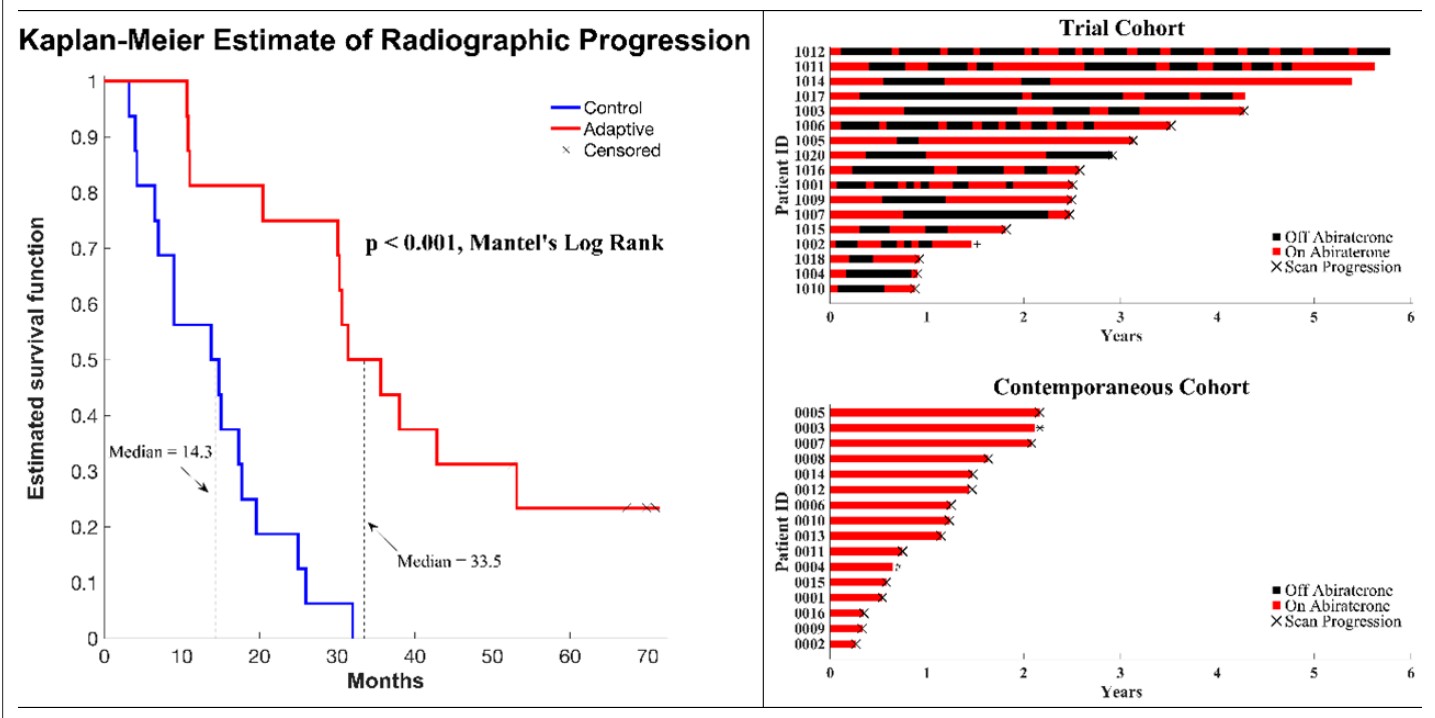

**Figure 1.** Clinical status of patients in the clinical trial at cutoff date of 01/01/2022.
(Left) Kaplan–Meier curve for time to radiographic progression in the adaptive therapy (n = 17) compared to continuous therapy (n = 16) cohort. Four patients in the adaptive arm remain in the trial with no evidence of progression (time on trial ranging from 52 to 69 months). (Right) Swimmer plot showing times on and off therapy, and clinical outcomes for each patient from both cohorts.

In a preliminary report, we found that the median radiographic TTP could not be less than 27 months (*Zhang et al., 2019*). Here, consistent with this and model predictions, evolution-based application of abiraterone significantly improved (p<0.001) median TTP (33.5 months; *Figure 1*) and median overall survival (OS; 58.5 months; hazard ratio, 0.41, 95% confidence interval [CI], 0.20–0.83, p<0.001) in the adaptive group compared to the 14.3 months median TTP and 31.3 months median OS in the contemporaneous group. Median radiographic TTP in the contemporaneous SOC group was slightly greater than the most recently reported outcomes (*Raju et al., 2021*) (13.0 months) perhaps reflecting selection criteria in which only patients with >50% decline of PSA after initial treatment were included in this cohort.

All 16 patients in the SOC group have progressed and all have died at the time of this report. Seven patients in the adaptive therapy remain alive and four patients remain on study without imaging progression (range 53–70 months) as of the date of submission. Patients in the adaptive therapy group received an average abiraterone dosing rate (mg drug/patient/unit time) of 54% compared to SOC. That is, on average, the trial patients were not receiving abiraterone during 46% of their time on trial.

## Mathematical analysis

Mathematically based analysis of the trial proceeded in two steps. First, longitudinal trial data allowed key model parameters, including growth rate, pretreatment ratio of sensitive and resistant cell populations, and the relative fitness of each population. Second, computer simulations of the model with updated parameters were performed on each patient in both cohorts to estimate intratumoral evolutionary dynamics that led to the observed outcomes.

## Converting longitudinal trial data to parameter estimates

Because ADT was continued during abiraterone therapy, we assumed that the T+ cell proliferation was linked exclusively to androgen production by the TP cells. That is, loss of TP cells would reduce intratumoral androgen concentrations necessitating a decline in the T+ population. Their linked fates allow us to reduce *Equation 1* to a two-species model with T- cells as the resistant population ($x_R$) and

the TP and T+ cells as a combined sensitive population ($x_S$). Population sizes correspond to overall tumor burden. We arbitrarily set the carrying capacity to 10,000 (its scalable). and we assume all cell types have the same carrying capacity. *Equation 1* becomes

$$S = \beta_{SC} r_S x_S \left( 1 - \frac{x_S + \alpha_{SR} x_R}{10000 - 9000} \right) \tag{2}$$

$$R = \beta_{SC} r_R x_R \left( 1 - \frac{x_R + \alpha_{RS} x_S}{10000} \right) \tag{3}$$

where $r_S$ and $r_r$ are the population growth rates (units of per day), $\beta_{SC}$ is a cell-type-independent scaling factor, and $\Lambda$ (value of 1 during abiraterone treatment and 0 during drug holidays) is the effect of abiraterone on the carrying capacity of the sensitive cells. Carrying capacity is generally set by limits to growth such as nutrients and space. Thus, we assume that under no therapy all cell types have the same need and utilization of nutrients and space, and we assume lack of testosterone induces a 90% drop in nutrient and space use efficiency. Finally, in the absence of abiraterone, any competitive advantage of the sensitive cells will manifest through either a higher growth rate, $r$ (only meaningful during transient dynamics away from carrying capacity), or a larger competitive effect of sensitive cells on resistant cells than vice versa (the most salient from a cost of resistance): $\alpha_{RS} > \alpha_{SR}$.

We used a two-step process for parameter estimation. First, for each patient (dropping those with insufficient pre-therapy data), we used the initial rate of increase of PSA to estimate the growth rate of sensitive cells, $r_S$. We then used the average of these patient-specific estimates of $r_S$ as the patient-wide value for subsequent parameter estimation in both the trial and continuous therapy cohorts. This assumes that prior to therapy, resistant cells represent a small fraction of extant cancer cells. To estimate the growth rate of resistant cells, we used the increase in PSA levels following disease progression in the continuous therapy cohort under the assumption that with disease progression the cancer cell population is predominately resistant. We then used the average of these patient-specific estimates of $r_R$ as the patient-wide parameter value for both trial and continuous therapy cohorts (Appendix 1).

The first step provided patient-wide estimates of $r_S$ and $r_R$ that were then used as fixed values for the second step of parameter estimation. In the second step, we used constrained nonlinear multi-variable optimization to estimate the scaling factor, $\beta_{SC}$ , the competitive effect of sensitive cells on resistant ones, $\alpha_{RS}$, and the initial population sizes of sensitive and resistant populations, $x_S(0)$ and $x_R(0)$ (Appendix 2). For these estimates, we made $\beta_{SC}$ and $\alpha_{RS}$ patient-wide, and $x_S(0)$ and $x_R(0)$ patient-specific. We set the competitive effect of resistant cells on sensitive cells to $\alpha_{RS} = 1$, the same as the intraspecific competition coefficients.

With these assumptions, we accomplish several things. We prevent overfitting with too many parameters by using the two-step estimation process and limiting the number of patient-specific parameters. By letting the initial sizes of sensitive and resistant cells be patient-specific, we allow for the high variability among patients in their initial response to therapy and subsequent disease dynamics. The efficacy of adaptive therapy depends on the presence of a cost of resistance, which in our model will manifest as the ratio of competition coefficients $\alpha_{RS}/\alpha_{SR} > 1$. It may be that the effect of resistant cells on sensitive cells is less than 1, but by assuming that $\alpha_{SR} = 1$ we expect the estimate for $\alpha_{RS} > 1$, and we have one less parameter to estimate.

Comparing patient PSA to the best model fit in the 32 patients (Appendix 3) allows several observations. The two steps of analyses resulted in estimates for the five parameters. The fits are generally tight, but there are exceptions in the relatively poor fit to the adaptive therapy patients 1004 and 1007, and the continuous therapy patient C002. This suggests that patient-wide model parameters (growth rates, scaling factor on growth rates, and competition coefficients) may vary in these patients. Alternatively, the serum PSA concentration may scale to population size differently in these patients. Relaxing any of these assumptions and refitting these three patients with more patient-specific parameters does substantially improve the model fit, but at the cost of having to do the same for all patients and overfitting.

Simulations producing a best model fit for longitudinal trial data in 33 patients (Appendix 3) were dependent on just six parameters.

The sensitive cell population had a significantly higher mean growth rate (0.0156 per day [population increase of 1.56% per day]) than that for resistant cells (0.0091 per day [increase of 0.91% per day]; $p < 0.05$; Appendix 1). Both values are well within the range observed in clinical cancers (*He*

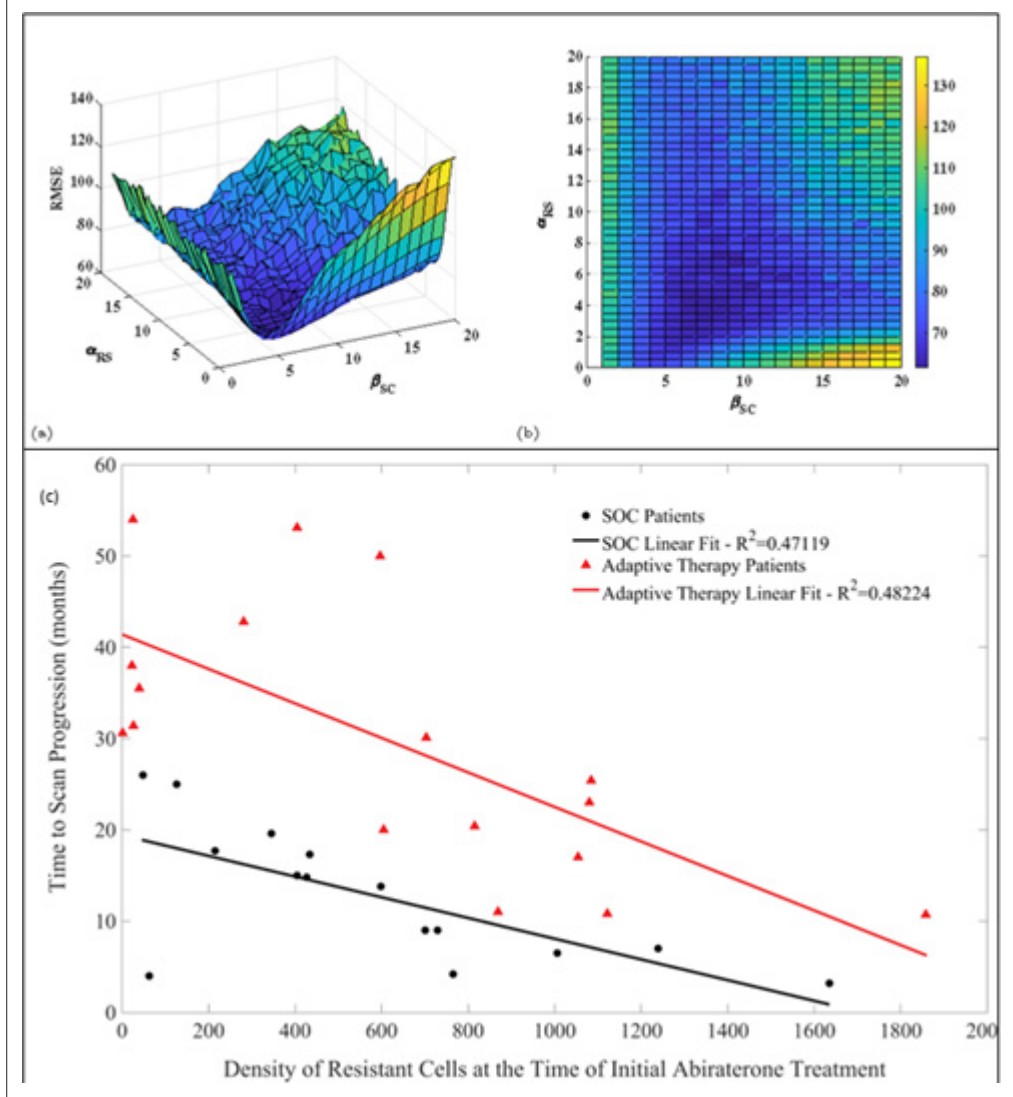

**Figure 2.** Estimates of key parameters (**a**, **b**) and relationship between time to radiographic progression (TTP) and initial population fraction of cancer cells resistant to abiraterone (**c**). Parameters $\alpha_{RS}$ (competition coefficient of sensitive on resistant cells) and $\beta_{SC}$ (growth rate scaling factor) were estimated by a constrained nonlinear multivariable optimization minimizing the least-squares difference between the output of the model and the actual patient data over the entire cohort. The global minimum occurred at $\alpha_{RS}$ = 6 and $\beta_{SC}$ = 8. (**c**) TTP declines with the estimated pretreatment fraction of resistant cells for both adaptive therapy and continuous therapy cohorts. Adaptive therapy is superior to continuous therapy. No adaptive therapy patient lies below the regression line for continuous therapy, and no continuous therapy patient lies above the regression line for adaptive therapy.

*et al., 2020*), and the difference is consistent with our theoretical premise that resistance incurs a cost that decreases fitness and proliferation when therapy is not applied. Mathematical estimates of pretreatment fractions of sensitive and resistant cells correlated with subsequent radiographic TTP in both cohorts (*Figure 2*, *Figure 3*), and TTP was greater in the adaptive cohort for every level of the pretreatment-resistant population.

Estimated values for the remaining patient-wide parameters yielded $\beta_{SC}$ = 8 and $\alpha_{RS} \approx 6$. For perspective, the ratio of the competition coefficient of a dominant species over a nondominant species in nature ranges from slightly above 1 to over 100 (*Inouye, 1999*). In our pre-trial model (*Zhang et al., 2017*), we used a ratio of $\alpha_{RS}/\alpha_{SR} \approx 2$. We now see that this was too conservative, and the actual ratio of 6 has clinical implications (*Figure 4*). If $\alpha_{RS}/\alpha_{SR} = 2$, the resistant population ($x_R$) increases as the sensitive population ($x_S$) declines but, following treatment cessation, remains constant. As a result of

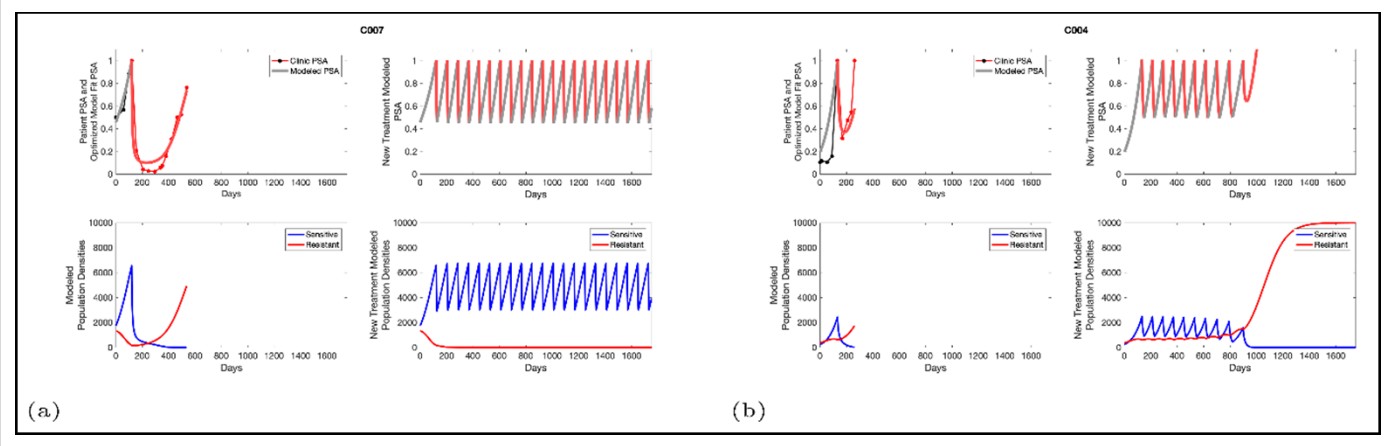

**Figure 3.** Based on estimated parameter values, retrospective analyses show that adaptive therapy could have provided disease control for patients on continuous therapy. For patients C007 (**a**) and C004 (**b**), the pretreatment fraction of resistant cancer cells was estimated as 0.3 and 0.25, respectively, and time to radiographic progression was 526 and 128 days, respectively.

this increase then plateau sequence, our original model simulations predicted the resistant population will inevitably become predominate, leading to progression after 2–20 cycles. However, with the retrospective and empirically derived $\alpha_{RS}/\alpha_{SR} = 6$, simulations (*Figure 4*) show the increasing sensitive population ($x_S$) after treatment cessation causes a decrease in the size of the resistant population ($x_R$)

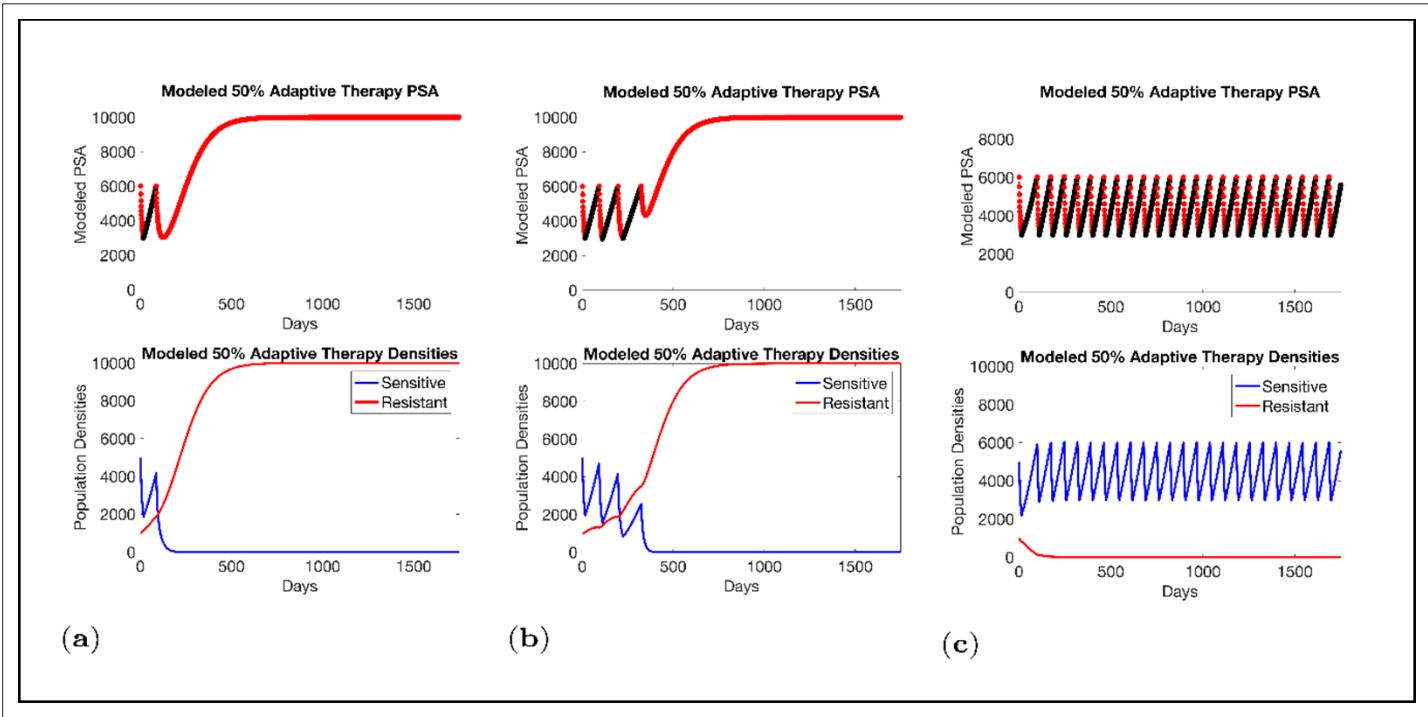

**Figure 4.** Sensitivity of resistant cell population to the value of the competition coefficient. In the top panels, we show the ideal cycling of the PSA treatment in which treatment is stopped immediately upon reaching 50% of the pretreatment value and resumed immediately upon reaching that value. In the lower panel, we show computer simulations of changes in the treatment-sensitive (blue) and treatment-resistant populations (red). Treatment dynamics are sensitive to the value of the competition coefficient ($\alpha_{RS}$), which is dependent on the fitness differences of the sensitive and resistant populations in the absence of treatment. In panel (**a**) we assume $\alpha_{RS} = 0.8$ and increase in $x_S$ does not decrease the population $x_R$ and adaptive therapy fails. In panel (**b**), $\alpha_{RS} = 2$, the increase in $x_S$ during treatment holidays slows the growth of $x_R$ and delays treatment failure. In panel (**c**) the estimated $\alpha_{RS}$RS 6 results in a *negative* growth rate in $x_R$ during proliferation of $x_S$. Over 3–4 cycles, the $x_R$ population approaches 0. This allows the cycling treatment to maintain tumor control indefinitely. Note that, however, this represents an ideal setting and does not account for other dynamics (see below) that may result in loss of control.

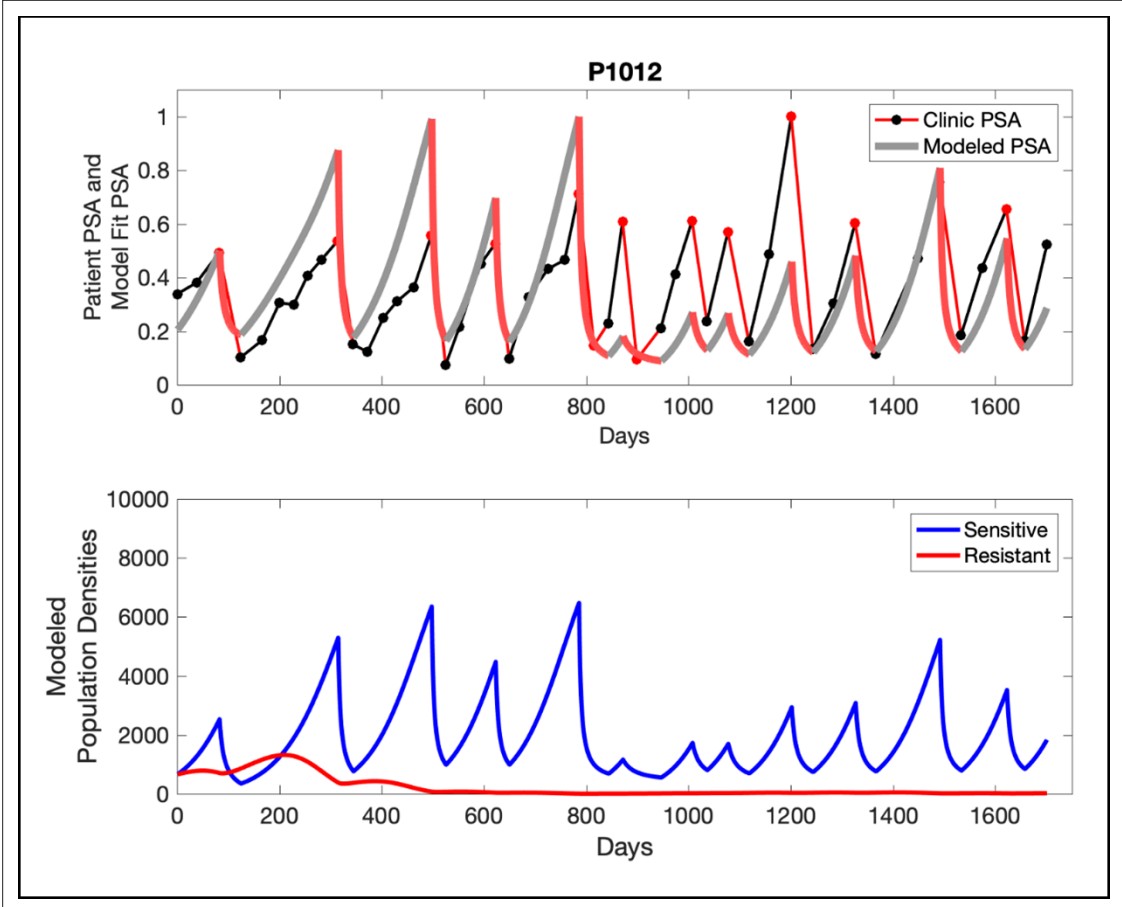

**Figure 5.** In the upper panel, the dotted line indicates actual PSA measurements. The red line represents the model fit for the data. In the lower panel, computer simulations estimating the sizes of the treatment-sensitive (blue) and resistant (red) populations over time are demonstrated. Simulations suggest that optimal timing resulted in the elimination of the resistant population in adaptive therapy patients with prolonged survival. Patient 1012 in the adaptive therapy cohort with enduring control (>1800 days). Model simulations suggest that the sequence of 2–4 treatment cycles caused the resistant population to reduce to near extinction permitting a stable cycling regime in which only abiraterone-sensitive cells are present.

such that, after 3–4 consecutive optimal treatment cycles, the resistant population can approach 0. This means that, in theory, cycling can persist indefinitely. Or, after using adaptive therapy to create persistent cycles, it may be possible to be therapeutically more aggressive to achieve cure.

The potential for achieving the actual or near extinction of the resistant cancer cells after 3–4 cycles of adaptive therapy may have occurred for the four trial patients who after >5 years of adaptive therapy remain on a stable cycling regime (*Figure 5*). If there is potential for permanent control, why was tumor progression observed in most members of the adaptive therapy cohort? Computer simulations suggest that they were overtreated. While protocol design required monthly PSA levels, radiographic studies were limited to 3-month intervals. As demonstrated in *Figure 6*, the protocol requirement for radiographic confirmation of response resulted in multiple weeks and even months in which patients remained on treatment even when the PSA was <10% of pretreatment values. Simulations demonstrate this excessive reduction of the sensitive population leads to the proliferation and predominance of the resistant population.

Finally, the models allowed us to explore whether the protocol to remove therapy at 50% PSA decline was optimal. In *Figure 7*, we demonstrate modeling results showing stopping therapy after a 20% PSA decline improved outcomes while stopping after an 80% decline produced a more rapid loss of control. This again demonstrates that the counterintuitive principle of adaptive therapy as more aggressive therapy, by reducing the size of the sensitive population, tends to accelerate growth of the resistant cells.

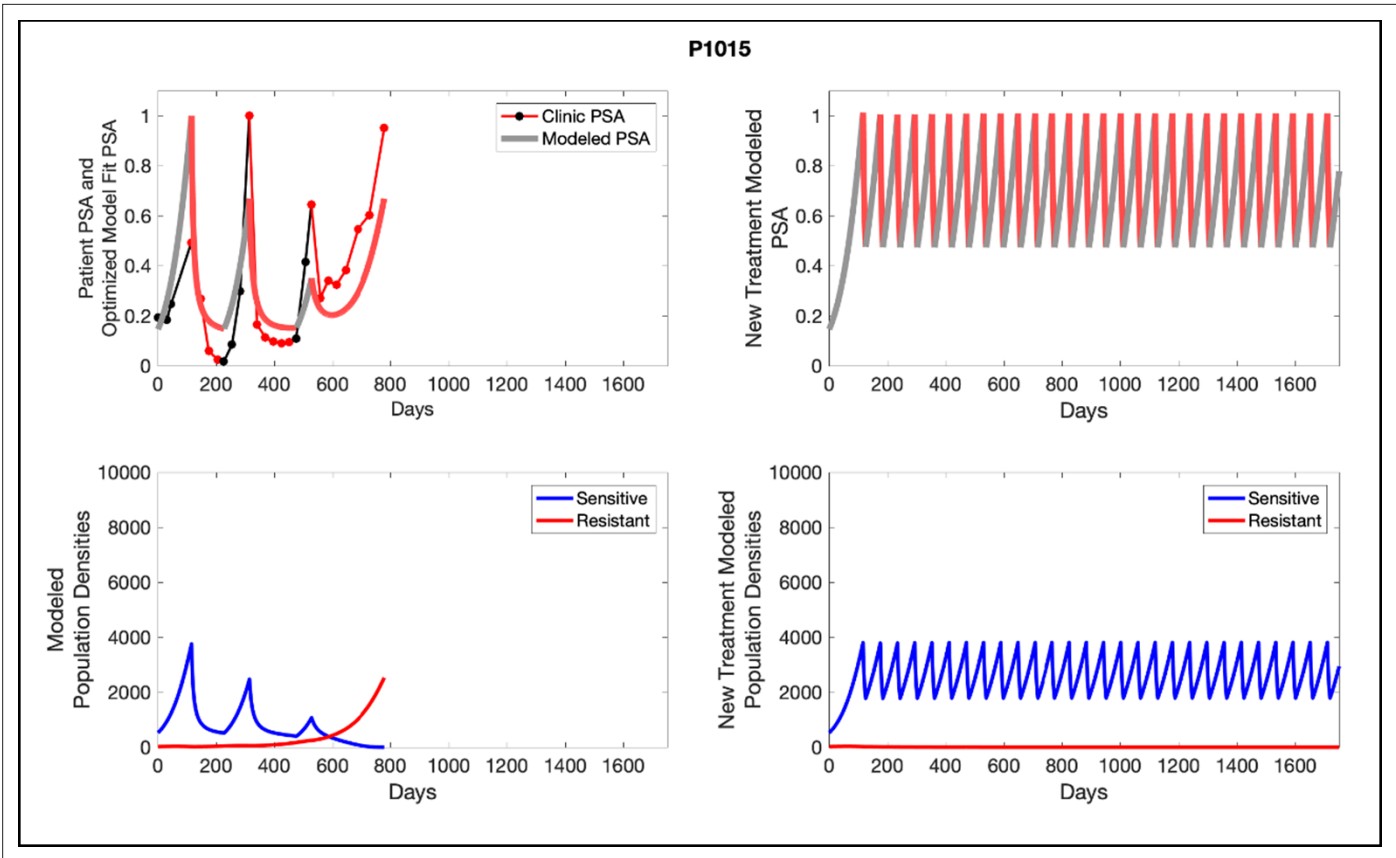

**Figure 6.** Left panels show actual patient PSA data (dotted line) and computer simulation curve fits (red line in upper panel) and estimated sizes (lower panel) of treatment-sensitive (blue) and resistant (red) populations during treatment. Right panel represents a computer simulation in which treatment is withdrawn immediately upon reaching the 50% threshold and restarted immediately upon returning to the pretreatment value. Left panel: simulations suggest that unintended, yet excessive, reduction of the sensitive population led to the proliferation and dominance of the resistant population. Right panel: optimizing the timing of withdrawing therapy immediately upon reaching the 50% pretreatment PSA threshold, thereby preventing overtreatment, allows maximal suppression of the resistant population and consistent long-term control in adaptive therapy patients. Of note, allowing the PSA to increase above pretreatment value had no negative consequences because it generally caused a further decline in the resistant population (data not shown).

## Discussion

There are multiple strategies for including evolutionary principles in cancer therapy. In general, maximum benefit is obtained by maintaining the largest possible population of treatment-sensitive cells, thus allowing them, through their greater fitness in the absence of treatment, to minimize or even reduce proliferation of resistant cells. Consistent with these general principles, computer simulations estimated that the best outcomes were obtained when treatment reduction of sensitive cells was minimized (*Figure 7*). Thus, for example, computer simulations suggested stopping treatment when the PSA reached only 80% of the pretreatment value (i.e., just a 20% reduction) compared to the 50% threshold used in this study (*Cunningham et al., 2018*; *Cunningham et al., 2020*). Furthermore, in preclinical experiments, a 'dose-adjustment' strategy in which the tumor was maintained at a stable volume by continuous adjustment of the treatment dose achieved the longest tumor control (*Enriquez-Navas et al., 2016*). Here, however, we opted to use the 50% threshold as compromise between optimal control and concerns about compliance and cost (the other methods, e.g., require more frequent testing and clinic visits).

Nevertheless, our results find that cycling of sensitive cells, depending on key intratumoral evolutionary parameters, can maintain control of resistant cells often for prolonged time periods. More broadly, we present a conceptual model for trial design in which the treatment protocol is linked to predictions from a mathematical model. Here, analyses of trial results, in addition to traditional cohort outcomes, includes mathematical curve fitting of longitudinal data from individual patients

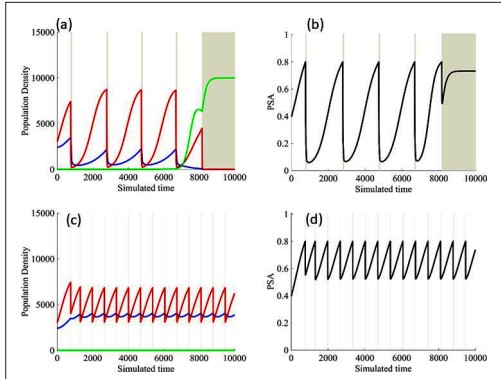

**Figure 7.** Results from adaptive therapy protocols in which a PSA drop of 80% is required before drug holiday (**a**) and (**b**), and in which only a 20% drop is required before drug holiday (**c**) and (**d**). The red is TP cells that are directly affected by administration of abiraterone, blue is the T+ cells, and green is the therapy-resistant T- cells. The regions where the background is highlighted are the times at which abiraterone is being administered.

to estimate key model parameters. Subsequently, computer simulations using the updated model can be applied to each patient to estimate the tumor evolutionary dynamics that led to the observed outcomes. Finally, computer simulations can examine alternative treatment strategies that would have produced better outcomes. Ultimately, further refinements of this approach will be necessary for patient-specific optimization of therapy. For example, here we must assume that key parameters for tumor eco-evolutionary dynamics are identical within the cohort. This is almost certainly incorrect, but patient-specific parameterization will require a new generation of clinical biomarkers. Thus, ideally, every cancer patient should have a unique mathematical model that is continuously updated throughout the treatment arc – similar to, for example, the models and computer simulations used to track storms.

While this approach seemed successful in this pilot trial, it clearly must be validated in multiple other trials with larger study cohorts. Furthermore, variable mechanisms of resistance can give rise to dynamics other than those observed in this relatively small cohort. In this study, for example, the patients received three different treatments (ADT, steroids, and abiraterone) but only abiraterone dosing was modulated. Thus, for example, intermittent dosing of steroids instead of or in addition to abiraterone is a potentially successful alternative strategy (*Fenioux et al., 2019*). We note that these more complex strategies can and should be explored mathematically to identify optimal trial design. Lastly, adaptive dosing in prostate cancer treatment is enabled by a serum biomarker (PSA) that is a generally accurate metric of changing tumor burden within a patient. Other cancers that lack a serum biomarker will require clinical decisions to be made based on estimates of tumor volumes from imaging. This strategy has been used in animal experiments (*Enriquez-Navas et al., 2016*) but does add concerns regarding accuracy and cost in a clinical setting.

Nevertheless, acknowledging the above caveats, we demonstrate that integration of evolution-based mathematical models into trial design significantly increased TTP in abiraterone therapy for mCRPC. Patients did not receive abiraterone, on average, during 54% of the trial period, thus reducing potential toxicity and expense. While we did not use quality-of-life metrics to estimate these benefits, an economic analysis of the trial found an average cost reduction of $70,000 per patient per year (*Mason et al., 2019*). The cohort size in this pilot study is relatively small, but we note that the increase in TTP was highly statistically significant (p<0.001) compared to a contemporaneous cohort and historical data (*Ryan et al., 2015*; *Ryan et al., 2013*; *Ryan et al., 2010*). Furthermore, as noted above, the use of mathematical models in trial design and analysis expands information that can be obtained from even small cohorts.

Finally, our 'After Action Analysis (*Stanková et al., 2019*)' using the updated parameter estimates from the longitudinal trial data predicted that every SOC patient would have benefited from the adaptive application of abiraterone, and no members of the adaptive cohort would have benefited from SOC dosing. Computer simulations also identified important flaws in the trial protocol. Because PSA sampling occurred at monthly intervals and imaging as well as physician appointment occurred at 3-to-4-month intervals, the decision to end or restart therapy often occurred weeks or months after the PSA value had crossed the necessary threshold. Somewhat counterintuitively, simulations demonstrated that delay in restarting abiraterone as the PSA increased had little clinical effect but delays in withdrawing treatment often resulted in excessive reduction of the sensitive tumor population that significantly reduced TTP. That is, by waiting too long for therapy withdrawal, the sensitive population was reduced to below levels that could effectively suppress proliferation of the resistant

population. In fact, computer simulations demonstrate that optimal timing of abiraterone withdrawal over 3–4 cycles could substantially reduce the resistant population and further increase the TTP. This is supported by computer simulations of the intratumoral evolutionary dynamics in four members of the adaptive therapy cohort who remain on stably cycling after >5 years. Thus, future plans for adaptive therapy trials in prostate cancer include more rapid withdrawal of therapy when PSA crosses the 50% threshold and more extensive monitoring of intratumoral evolution using serum biomarkers, including testosterone as well as circulating DNA for AR amplification, AR mutations, and CYP17a expression. Finally, we note that the models focus on prostate cancer interactions with testosterone, and, thus, any therapy related to androgen receptors and androgen production can be modeled using this approach. Furthermore, any cancer treatment with cytotoxic effects that induce evolution of resistance can be addressed using these methods.

## Acknowledgements

We thank Dr. Robert Gillies for his helpful comment and critiques, the patients who participated in the trial, and the trial coordinators and clinic staff who facilitated the complicated protocol. This work was supported by the James S McDonnell Foundation grant, 'Cancer therapy: Perturbing a complex adaptive system,' grants from the Jacobson Foundation, NIH/National Cancer Institute (NCI) R01CA170595, Application of Evolutionary Principles to Maintain Cancer Control (PQ21), and NIH/NCI U54CA143970-05 (Physical Science Oncology Network (PSON)) 'Cancer as a complex adaptive system.' This work has also been supported in part by the Clinical Trials Core Facility at the H Lee Moffitt Cancer Center and Research Institute, an NCI-designated Comprehensive Cancer Center (P30-CA076292), and the European Union's Horizon 2020 research and innovation program under the Marie Sklodowska-Curie grant agreement no. 690817.

## Additional information

### Competing interests

Jingsong Zhang: JZ received consultancy fees from AstraZeneca, Bayer, Dendreon, Myoant Sciences and Pfizer, and honoraria from AstraZeneca, Dendreon and Sanofi. The author has no other competing interests to declare. The other authors declare that no competing interests exist.

### Funding

| Funder | Grant reference number | Author |
| --- | --- | --- |
| National Cancer Institute | R01CA170595 | Robert Gatenby |
| National Cancer Institute | U54CA143970-05 | Robert Gatenby |
| Moffitt Cancer Center | | Jingsong Zhang<br>Jessica Cunningham<br>Joel Brown<br>Robert Gatenby |
| Horizon 2020 | | Joel Brown |

The funders had no role in study design, data collection and interpretation, or the decision to submit the work for publication.

### Author contributions

Jingsong Zhang, Conceptualization, Data curation, Investigation, Methodology, Project administration, Writing – original draft, Writing – review and editing; Jessica Cunningham, Conceptualization, Formal analysis, Software, Writing – original draft, Writing – review and editing; Joel Brown, Conceptualization, Data curation, Formal analysis, Supervision, Writing – original draft, Writing – review and editing; Robert Gatenby, Conceptualization, Formal analysis, Funding acquisition, Methodology, Writing – original draft, Writing – review and editing

### Author ORCIDs

Robert Gatenby http://orcid.org/0000-0002-1621-1510

## Ethics
Clinical trial registration NCT02415621.

Human subjects: The study obtained IRB approval and is available in clinicaltrials.gov (NCT02415621). Detailed informed consent was obtained from all subjects. All data presented are anonymized.

## Decision letter and Author response
Decision letter https://doi.org/10.7554/eLife.76284.sa1
Author response https://doi.org/10.7554/eLife.76284.sa2

---

## Additional files

### Supplementary files
• MDAR checklist

### Data availability
The data generated in this study are available within the article and the PSA graphs in Appendix 4, 5, and 6. The code used for analyzing, parameter fitting and simulations of the Lotka-Volterra model is available on GitHub (https://github.com/cunninghamjj/Evolution-based-mathematical-models-significantly-prolong-response-to-Abiraterone-in-mCRPC, copy archived at swh:1:rev:ffa835ce8f-4252d92a8c97f0e7324a1b6f87727b). The source data file containing anonymized PSA data from trial patients is available in both Matlab and .xlsx format at https://github.com/cunninghamjj/Evolution-based-mathematical-models-significantly-prolong-response-to-Abiraterone-in-mCRPC/blob/main/data/TrialPatientData.xlsx.

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

# Appendix 1

## Estimating growth rates from patient data

We estimated the growth rates (units of per day) of the sensitive and resistant cancer cell populations using exponential fits to PSA measurements from the patients enrolled in the trial and the contemporaneous cohort. To estimate the resistant cell population growth rate (*Appendix 1—table 1*), we used time series of increasing PSA levels after disease progression from patients in the contemporaneous cohort. At the point of disease progression, we assume that the tumor population is comprised of mostly abiraterone-resistant cells. The patients of this cohort provided 11 good examples for estimating the resistant cell population growth rate. *Appendix 1—figure 1* provides examples from two patients of these data and analyses.

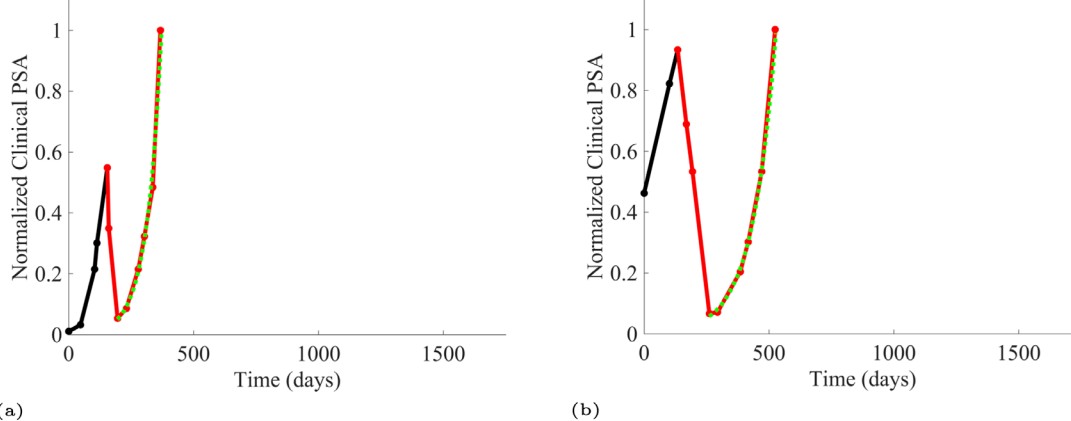

(a)  (b)

**Appendix 1—figure 1.** PSA time-series data for patients C005 (**a**) and C013 (**b**). Here, the PSA dynamics are highlighted in red, showing that abiraterone is being administered. The green dotted line shows the fit used to estimate the growth rate of the resistant cell population after progression through abiraterone. As abiraterone is still being administered and cell populations are still growing, we assume that sensitive cells have been eliminated and only abiraterone-resistant cells remain.

**Appendix 1—table 1.** Estimates of sensitive, resistant growth rates from patient data in both the contemporaneous and adaptive therapy cohorts.

| Sensitive cells | | Resistant cells | |
|---|---|---|---|
| Patient identifier | Extracted growth rate | Patient identifier | Extracted growth rate |
| C010 | 0.0071 | C013 | 0.0109 |
| C014 | 0.0142 | C012 | 0.0025 |
| C012 | 0.0075 | C010 | 0.0046 |
| C011 | 0.0123 | C005 | 0.0173 |
| C009 | 0.0107 | C008 | 0.0047 |
| C007 | 0.0062 | C007 | 0.0111 |
| C005 | 0.0214 | C006 | 0.0130 |
| C004 | 0.0196 | C005 | 0.0173 |
| C003 | 0.0100 | C004 | 0.0124 |
| C001 | 0.0062 | C002 | 0.0031 |
| P1018 | 0.0189 | C001 | 0.0022 |
| P1017 | 0.0068 | | |
| P1016 | 0.0106 | | |
| P1016 | 0.0146 | | |

*Appendix 1—table 1 Continued on next page*

*Appendix 1—table 1 Continued*

| Sensitive cells | | Resistant cells | |
| --- | --- | --- | --- |
| **Patient identifier** | **Extracted growth rate** | **Patient identifier** | **Extracted growth rate** |
| P1015 | 0.0419 | | |
| P1014 | 0.0109 | | |
| P1012 | 0.0059 | | |
| P1012 | 0.0076 | | |
| P1012 | 0.0091 | | |
| P1012 | 0.0160 | | |
| P1012 | 0.0100 | | |
| P1012 | 0.0170 | | |
| P1011 | 0.0191 | | |
| P1011 | 0.0191 | | |
| P1011 | 0.0304 | | |
| P1007 | 0.0118 | | |
| P1006 | 0.0216 | | |
| P1004 | 0.0071 | | |
| P1003 | 0.0124 | | |
| P1003 | 0.0116 | | |
| P1003 | 0.0105 | | |
| P1002 | 0.0245 | | |
| P1001 | 0.0446 | | |
| P1001 | 0.0317 | | |

To estimate the population growth rate of abiraterone-sensitive cells (*Appendix 1—table 1*), we used time series of increasing PSA levels prior to any treatment with abiraterone. Prior to treatment, we expect that virtually all cancer cells are abiraterone-sensitive cells. Ten patients from the contemporaneous cohort provided sufficient data. *Appendix 1—figure 2* provides examples from two patients of these data and analyses.

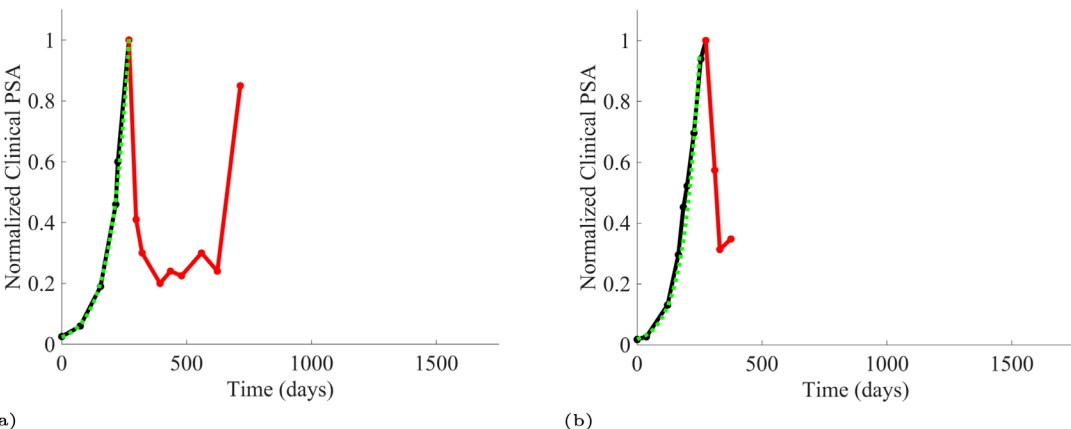

(a) (b)

**Appendix 1—figure 2.** PSA time-series data for patients C014 (**a**) and C011 (**b**). Here, the PSA dynamics are highlighted in red, showing that abiraterone is being administered. The green dotted line shows the fit used to estimate the growth rate of the sensitive cell population before any treatment with abiraterone. As abiraterone has not yet been introduced, this growth is assumed to be completely from the abiraterone-sensitive cells.

Additionally, from patients on the adaptive therapy trial, we estimated the growth rates of sensitive cells using increases in PSA values during periods of the therapy cycle when patients were not receiving abiraterone. During such periods, we assume that growth comes predominantly from the abiraterone-sensitive population. Thirteen adaptive therapy patients were used to estimate the growth rates of sensitive cells. A given patient could provide from 1 to 6 estimates based on the number of therapy cycles. *Appendix 1—figure 3* provides examples for two such patients.

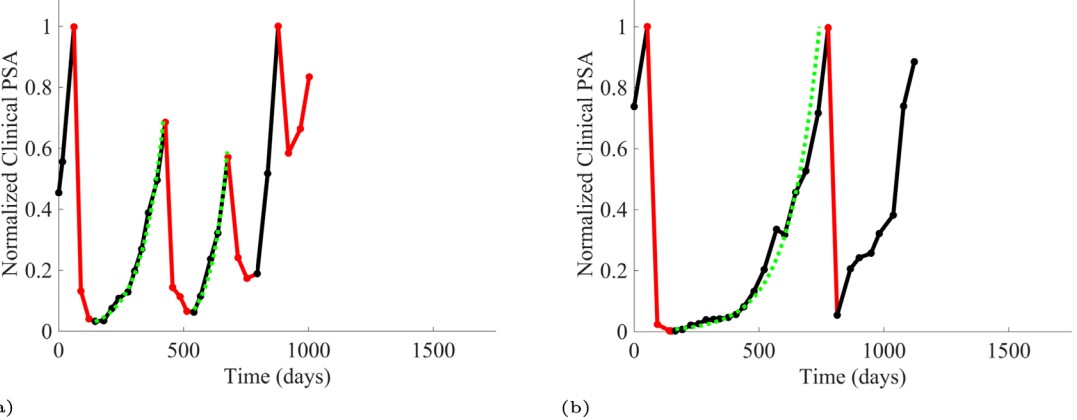

(a)                                                                                     (b)

**Appendix 1—figure 3.** PSA time-series data for patients P1016 (**a**) and P1017 (**b**). PSA dynamics highlighted in red and black show periods of abiraterone therapy on and off, respectively. Green dotted lines show exponential fits to periods when abiraterone was off. These provide estimates for the growth rates of the sensitive cell population during the off treatment period of an adaptive therapy cycle. We obtained two estimated values for P1016 and one for P1017. When off abiraterone, we assume that the growth in tumor burdens comes predominantly from the abiraterone-sensitive cells.

Using estimates shown in *Appendix 1—table 1*, sensitive cells had a significantly higher mean growth rate (0.0156) than that for resistant cells (0.0091), $t_x = 2.11$, p<0.05. For fitting patient data, we fixed the growth rates of sensitive and resistant cancer cells to $r_S = 0.0156$ and $r_R = 0.0091$. Because the actual PSA growth rates may be dampened by limits to growth, these estimates for the cancer cells' intrinsic growth rates may be underestimates. Higher intrinsic growth rates may better reflect the dynamics of the Lotka–Volterra competition model. Hence, for fitting the L-V model to the patients' data we included a multiplier $\beta_{SC}$ as a scaling factor for the PSA dynamics.

Note that P1002, while not used in the analysis of the trial due to noncompliance, is used for mathematical analysis as data for multiple adaptive cycles prior to noncompliance is available. This brings the total number of adaptive therapy patients used in parameter estimation to 17. Furthermore, historical PSA measurements are unavailable for one patient used in analysis of the trial, bringing the total number of contemporaneous patients used in parameter estimation to 15.

# Appendix 2

## Parameter optimization

With the growth rates set to $r_S = 0.0156$ and $r_R = 0.0091$ for all patients, there remain just four parameters for fitting the patient data (*Appendix 2—table 1*). The parameter optimization seeks to estimate values for the ecological scaling of the growth rates $\beta_{SC}$, the competitive effect of sensitive cells on resistant cells $\alpha_{RS}$, and the initial tumor composition of sensitive $x_S(0)$ and resistant $x_R(0)$ cells. In what follows, we set the competitive effect of resistant cells on sensitive cells to $\alpha_{SR} = 1$ which may be an overestimate if sensitive cells are weak competitors, but which provides another fixed parameter for all patients. Furthermore, like the growth rates, we shall make across-patient estimates of $\beta_{SC}$ and $\alpha_{RS}$. In this way, we assume strong convergent and parallel evolution of prostate cancer between patients. We will let the initial population sizes of sensitive and resistant cancer cells be patient specific.

**Appendix 2—table 1.** Table of parameters and definitions to be optimized using constrained optimization.

| Parameter | Definition |
| --- | --- |
| $x_S(0)$ | Abundance of sensitive cells at the time of initial treatment |
| $x_R(0)$ | Abundance of resistant cells at the time of initial treatment |
| $\alpha_{RS}$ | Competitive effect of sensitive cells on resistant cells |
| $\beta_{SC}$ | Ecological scale factor |

We implemented constrained nonlinear multivariable optimization using the MATLAB optimization toolbox (fmincon) in order to find the combination of variables presented in *Appendix 2—table 1* that minimized the cumulative least-squares difference between the output of the model and the actual patient data over the entire cohort. Patient PSA and model PSA are normalized to the maximum PSA value for that patient or simulation. In this way, all PSA values of the patient data and the modeled data were $\in [0, 1]$. The optimization minimizes the mean square error between the scaled PSA of the model and the scaled PSA of the patient data. Extra weight is given to the data points when a change in treatment occurs where using a weighting term of $w = 5$ at these points and $w = 1$ at all other points. Constraints on each of the four variables are presented in *Appendix 2—table 2*.

**Appendix 2—table 2.** Table of constraints on parameters to be optimized using constrained optimization.

| Parameter | Constraint |
| --- | --- |
| $x_S(0)$ | $\in [1, 10000]$ |
| $x_R(0)$ | $\in [1, 10000]$ |
| $\alpha_{RS}$ | $\in [0, 20]$ |
| $\beta_{SC}$ | $\in [0, 20]$ |

## Optimized parameters

The minimal mean square error was found with the optimized parameters as follows: $\alpha_{RS} = 6$ and $\beta_{SC} = 8$. The patient-specific estimates for the initial population sizes of sensitive and resistant cells with $\alpha_{RS} = 6$ and $\beta_{SC} = 8$ are shown in *Appendix 2—table 3*.

**Appendix 2—table 3.** Optimized parameters for each patient resulting from nonlinear constrained optimization.

| Patient | $x_S(0)$ | $x_R(0)$ | $\alpha_{RS}$ | $\beta_{SC}$ |
| --- | --- | --- | --- | --- |
| P1001 | 86.74 | 1.00 | 6 | 8 |

*Appendix 2—table 3 Continued on next page*

*Appendix 2—table 3 Continued*

| Patient | $x_S(0)$ | $x_R(0)$ | $\alpha_{RS}$ | $\beta_{SC}$ |
|---|---|---|---|---|
| P1002 | 165.99 | 24.67 | 6 | 8 |
| P1003 | 10000.00 | 124.98 | 6 | 8 |
| P1004 | 19.90 | 400.57 | 6 | 8 |
| P1005 | 802.20 | 12.31 | 6 | 8 |
| P1006 | 102.28 | 174.59 | 6 | 8 |
| P1007 | 253.07 | 149.12 | 6 | 8 |
| P1009 | 20.41 | 23.61 | 6 | 8 |
| P1010 | 196.09 | 1001.19 | 6 | 8 |
| P1011 | 29.19 | 6.31 | 6 | 8 |
| P1012 | 567.60 | 663.13 | 6 | 8 |
| P1014 | 1640.76 | 273.61 | 6 | 8 |
| P1015 | 126.13 | 6.38 | 6 | 8 |
| P1016 | 4589.21 | 4078.11 | 6 | 8 |
| P1017 | 6035.16 | 4060.25 | 6 | 8 |
| P1018 | 1495.72 | 318.18 | 6 | 8 |
| P1020 | 2061.34 | 477.07 | 6 | 8 |
| C001 | 111.70 | 19.46 | 6 | 8 |
| C002 | 1867.68 | 1.00 | 6 | 8 |
| C003 | 61.96 | 5.81 | 6 | 8 |
| C004 | 110.57 | 156.36 | 6 | 8 |
| C005 | 66.07 | 127.48 | 6 | 8 |
| C006 | 1226.45 | 25.31 | 6 | 8 |
| C007 | 1740.08 | 1325.03 | 6 | 8 |
| C008 | 4362.58 | 24.08 | 6 | 8 |
| C009 | 148.30 | 936.54 | 6 | 8 |
| C010 | 273.95 | 798.61 | 6 | 8 |
| C011 | 19.85 | 129.22 | 6 | 8 |
| C012 | 2148.71 | 301.47 | 6 | 8 |
| C013 | 165.20 | 31.48 | 6 | 8 |
| C014 | 13.57 | 5.09 | 6 | 8 |
| C015 | 1211.43 | 483.69 | 6 | 8 |

In the following analysis where we consider how each patient might have fared under different treatment protocols, we keep the patient-specific parameters as given in *Appendix 2—table 3*. This includes the patients that showed relatively poorer fits. We feel this maintains a conservative approach to the subsequent exploration of treatment options with the caveat that additional error propagation might occur insofar as other model parameters may indeed be patient specific.

These initial conditions allowed for the extraction of the tumor composition at the time abiraterone was first administered clinically to each patient, which was not always at t = 0. The relative fraction of the resistant to sensitive cells at the time of initial treatment is compared to the TTP (months) reported for each patient (*Appendix 2—table 4*).

**Appendix 2—table 4.** Tumor composition of sensitive and resistant populations at the time of initial abiraterone therapy from the optimized model fits to patient data.
The calculated percentage of resistant to sensitive cells alongside the clinical time to progression (TTP) for each patient is also shown.

| Patient | $x_S(t_{ABI})$ | $x_R(t_{ABI})$ | $\frac{x_r(t_{ABI})}{x_S(t_{ABI})}\%$ | TTP |
|---|---|---|---|---|
| P1001 | 2158.32 | 52.35 | 2.43 | 30.6 |
| P1003 | 5782.98 | 105.32 | 1.82 | 53.1 |
| P1004 | 3475.70 | 1621.81 | 46.66 | 11.0 |
| P1005 | 4407.49 | 7.77 | 0.18 | 38.0 |
| P1006 | 2483.02 | 603.80 | 24.32 | 42.8 |
| P1007 | 3503.41 | 203.61 | 5.81 | 30.1 |
| P1009 | 2830.43 | 156.89 | 5.54 | 17.0 |
| P1010 | 2407.93 | 1765.24 | 73.31 | 10.7 |
| P1011 | 3285.31 | 26.37 | 0.80 | 25.4 |
| P1012 | 2530.65 | 706.23 | 27.91 | 54.0 |
| P1014 | 4708.02 | 127.91 | 2.72 | 50.0 |
| P1015 | 3757.24 | 23.55 | 0.63 | 20.4 |
| P1016 | 6418.04 | 678.74 | 10.58 | 31.4 |
| P1017 | 7268.07 | 557.30 | 7.67 | 35.5 |
| P1018 | 1887.88 | 315.85 | 16.73 | 10.8 |
| P1020 | 3534.42 | 346.12 | 9.79 | 23.0 |
| C001 | 2250.42 | 136.14 | 6.05 | 9.0 |
| C002 | 1867.68 | 1.00 | 0.05 | 26.0 |
| C003 | 1976.15 | 32.96 | 1.67 | 15.0 |
| C004 | 2435.97 | 617.06 | 25.33 | 4.2 |
| C005 | 2696.58 | 600.39 | 22.26 | 7.0 |
| C006 | 1998.86 | 25.70 | 1.29 | 17.7 |
| C007 | 6552.52 | 166.29 | 2.54 | 17.3 |
| C008 | 4362.58 | 24.08 | 0.55 | 19.6 |
| C009 | 4545.92 | 787.31 | 17.32 | 6.5 |
| C010 | 4951.79 | 472.52 | 9.54 | 9.0 |
| C011 | 2953.60 | 772.31 | 26.15 | 4.0 |
| C012 | 7754.85 | 15.28 | 0.20 | 25.0 |
| C013 | 2113.11 | 72.78 | 3.44 | 13.8 |
| C014 | 2645.67 | 44.98 | 1.70 | 14.8 |
| C015 | 2356.24 | 683.69 | 29.02 | 3.2 |

## Appendix 3

### Optimized model fits

For each patient, the optimized model fits are shown with patient-specific parameters from *Appendix 2—table 3*.

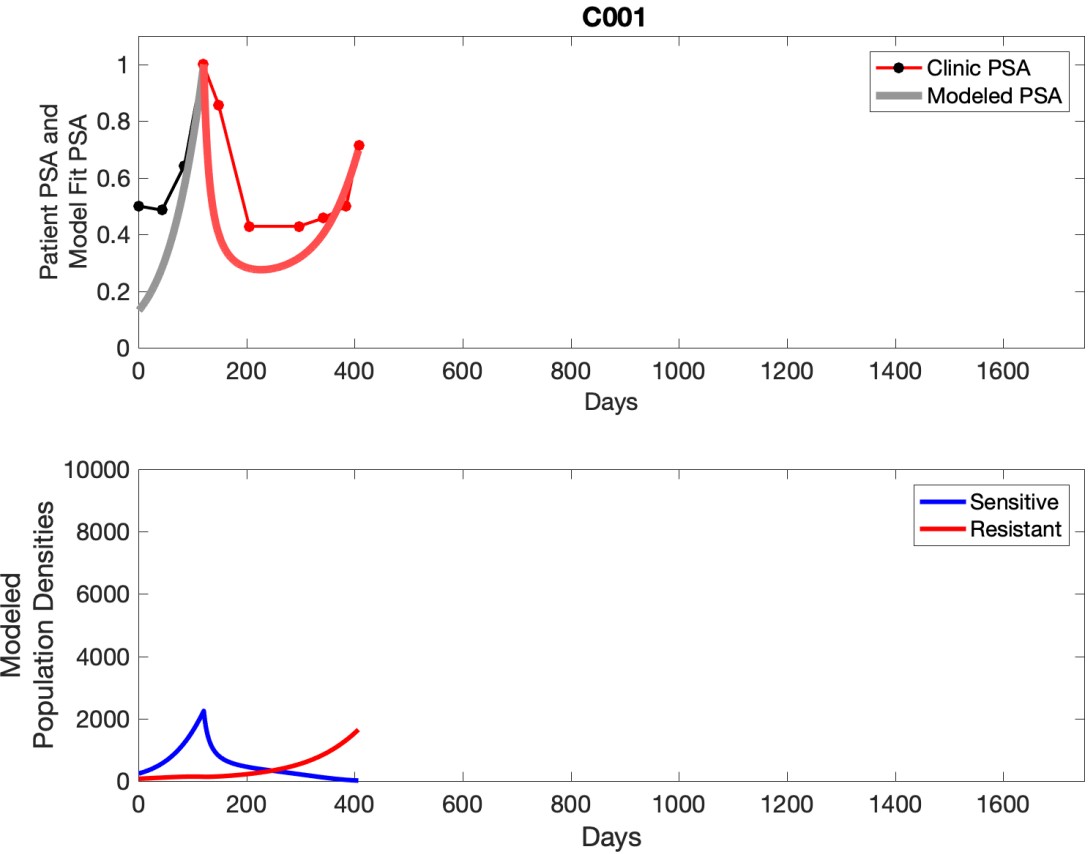

**Appendix 3—figure 1.** Observed PSA dynamics and model fit for the designated subject are shown in the top panel, and model simulations estimating changes in sensitive and resistant cell populations are in the lower panel.

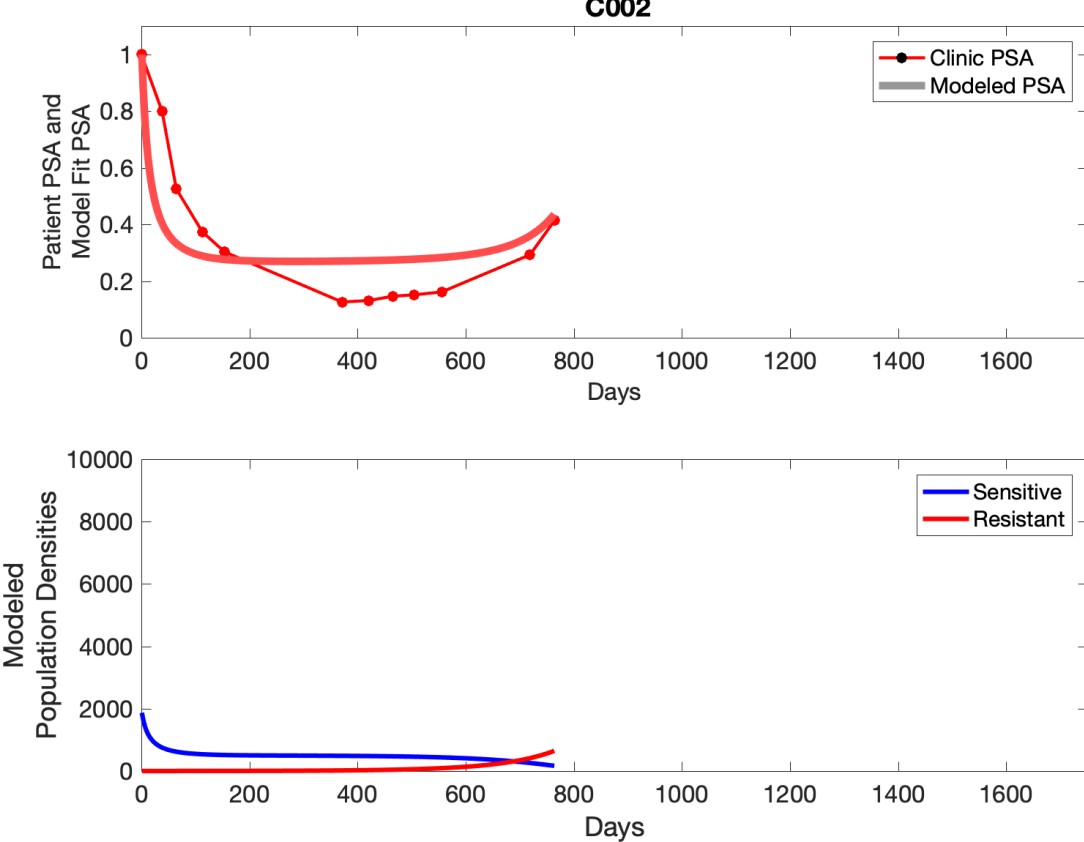

**Appendix 3—figure 2.** Observed PSA dynamics and model fit for the designated subject are shown in the top panel, and model simulations estimating changes in sensitive and resistant cell populations are in the lower panel.

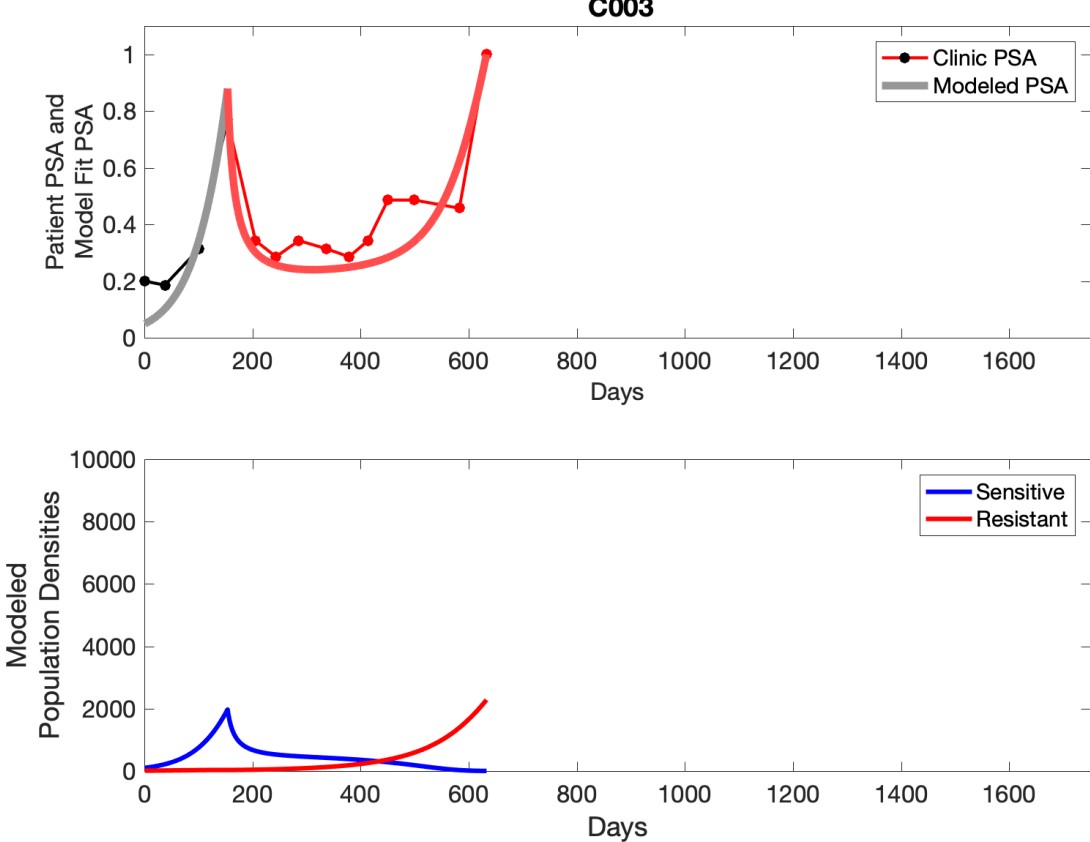

**Appendix 3—figure 3.** Observed PSA dynamics and model fit for the designated subject are shown in the top panel, and model simulations estimating changes in sensitive and resistant cell population are in the lower panel.

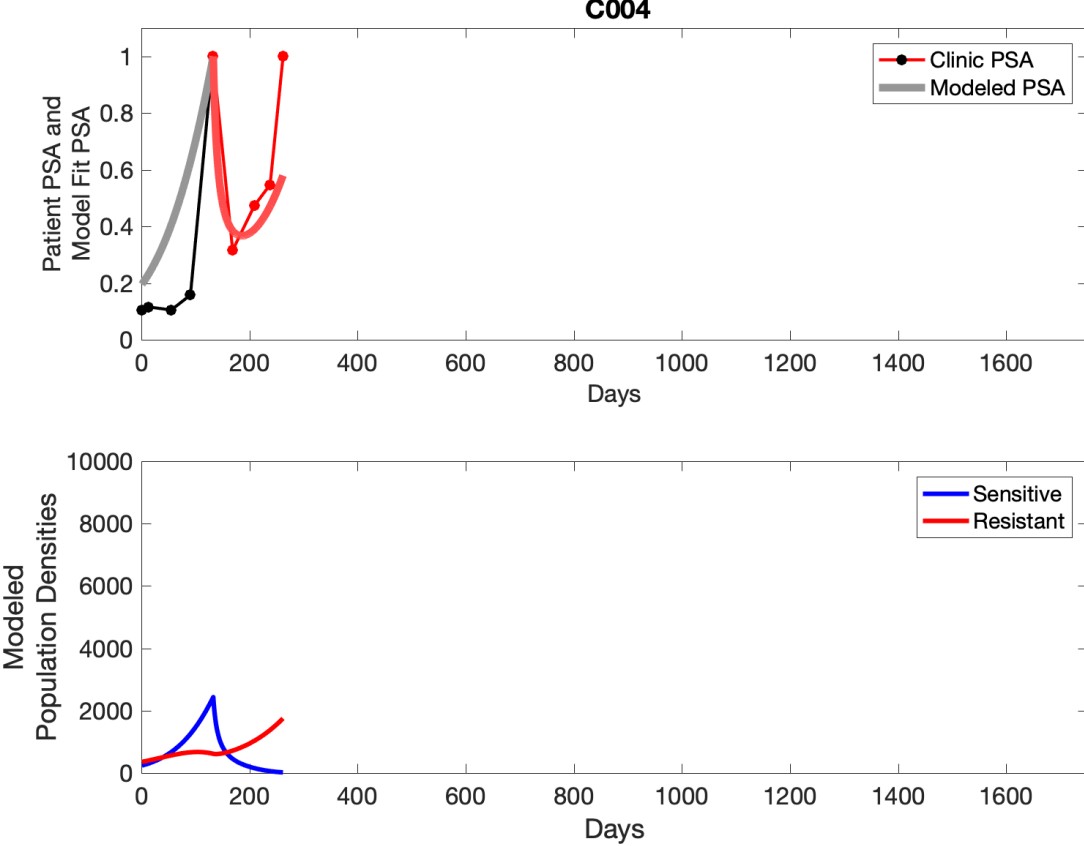

**Appendix 3—figure 4.** Observed PSA dynamics and model fit for the designated subject are shown in the top panel, and model simulations estimating changes in sensitive and resistant cell population are in the lower panel.

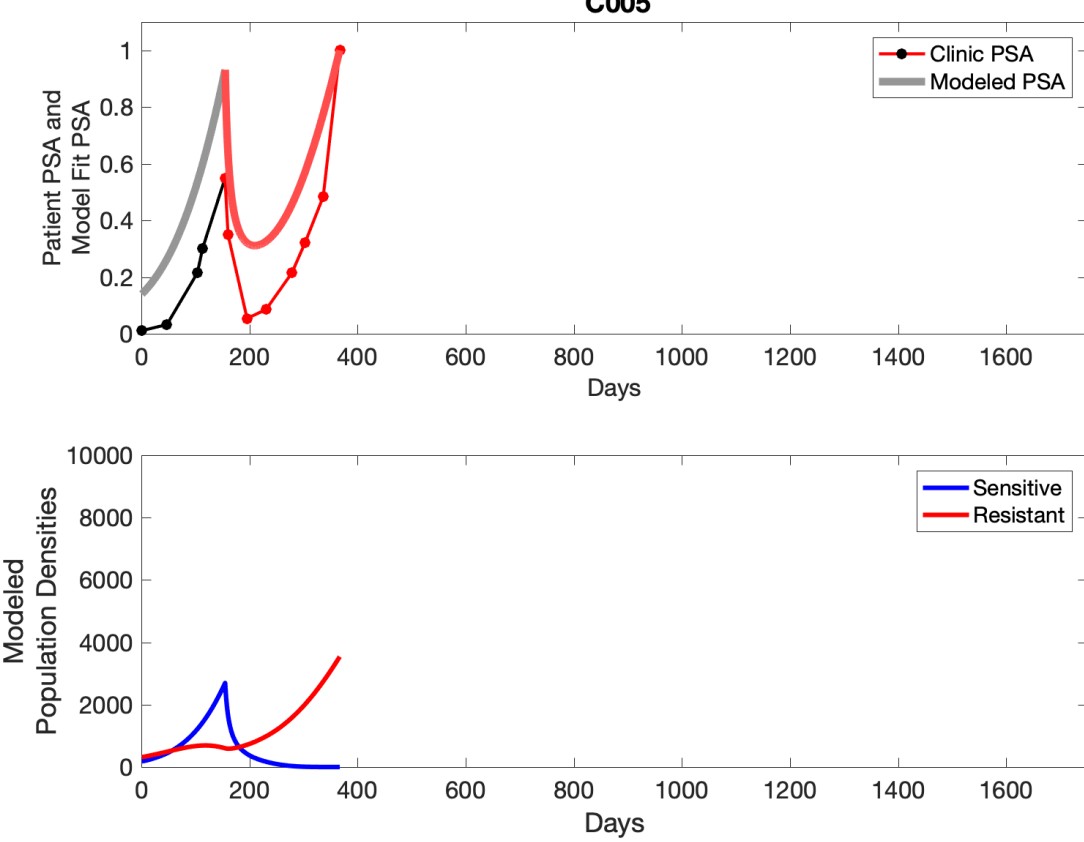

**Appendix 3—figure 5.** Observed PSA dynamics and model fit for the designated subject are shown in the top panel, and model simulations estimating changes in sensitive and resistant cell population are in the lower panel.

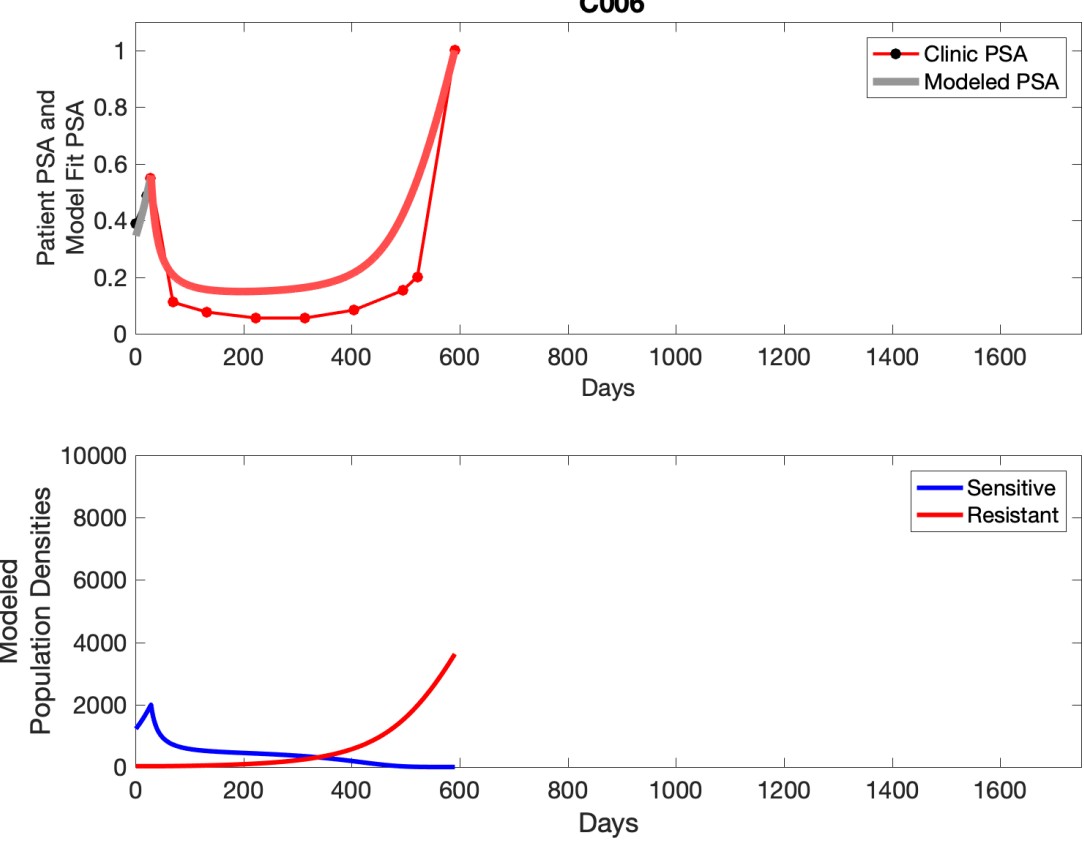

**Appendix 3—figure 6.** Observed PSA dynamics and model fit for the designated subject are shown in the top panel, and model simulations estimating changes in sensitive and resistant cell population are in the lower panel.

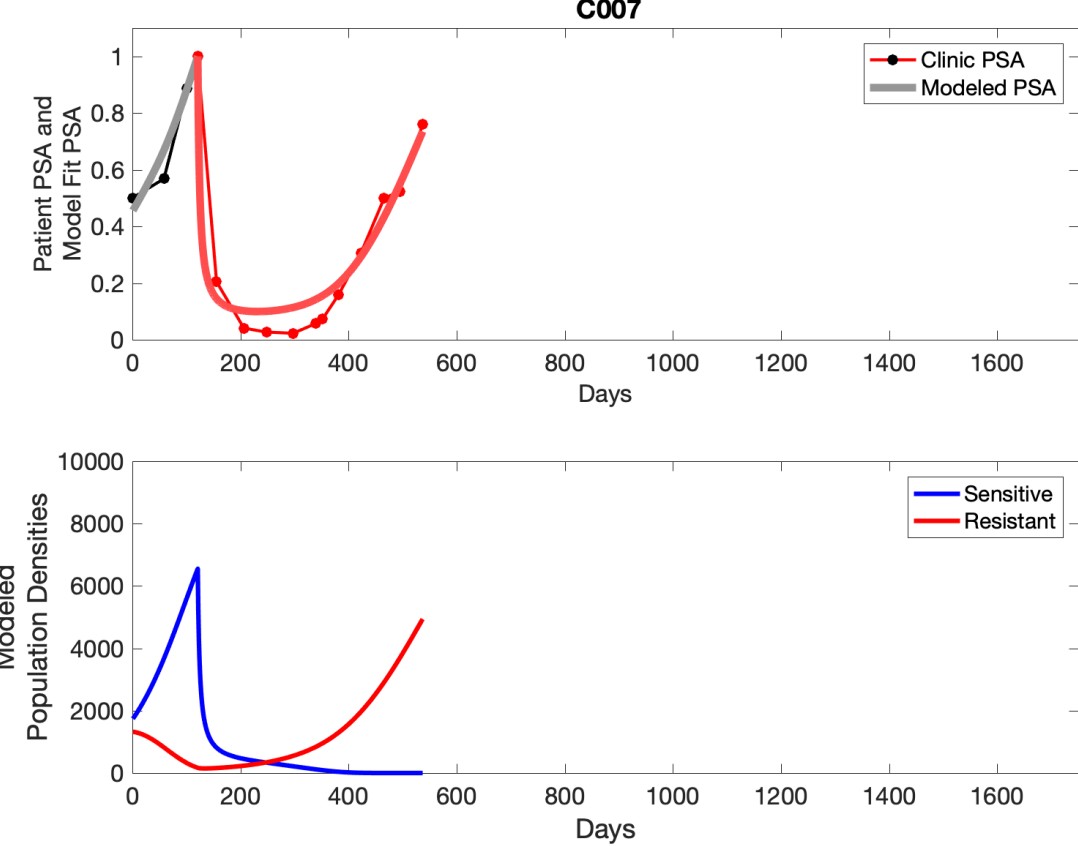

**Appendix 3—figure 7.** Observed PSA dynamics and model fit for the designated subject are shown in the top panel, and model simulations estimating changes in sensitive and resistant cell population are in the lower panel.

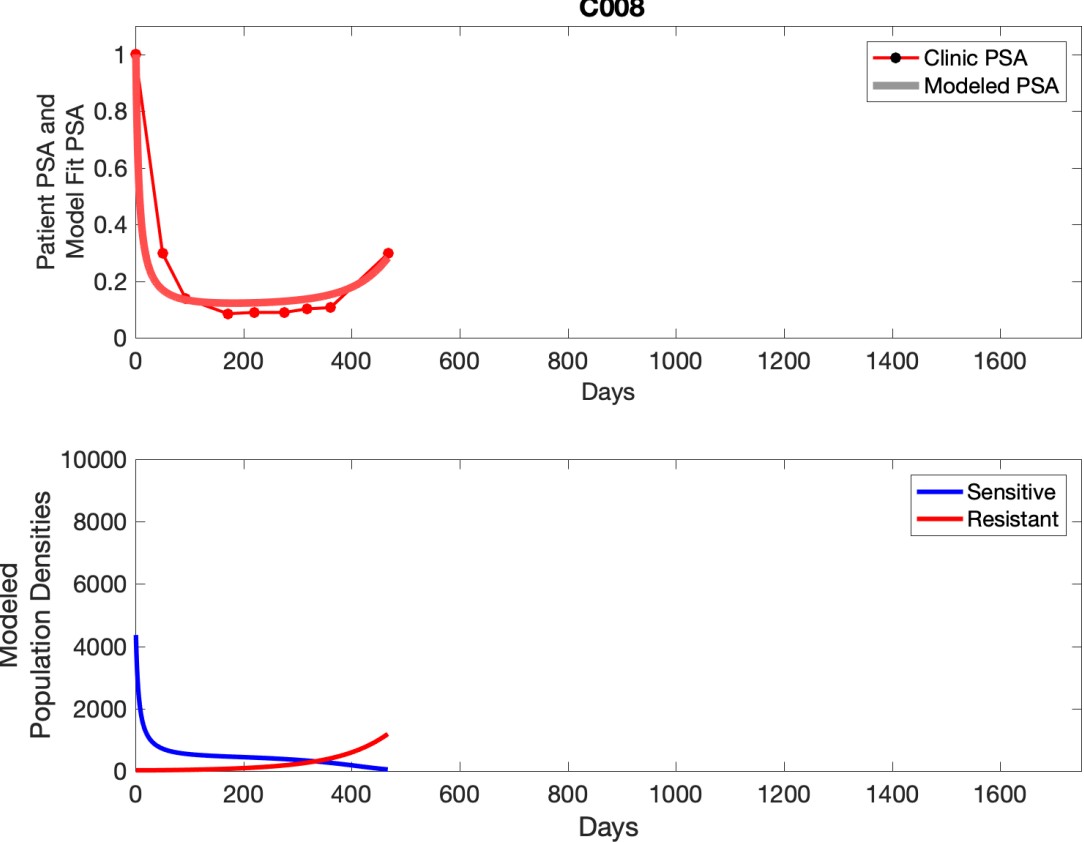

**Appendix 3—figure 8.** Observed PSA dynamics and model fit for the designated subject are shown in the top panel, and model simulations estimating changes in sensitive and resistant cell population are in the lower panel.

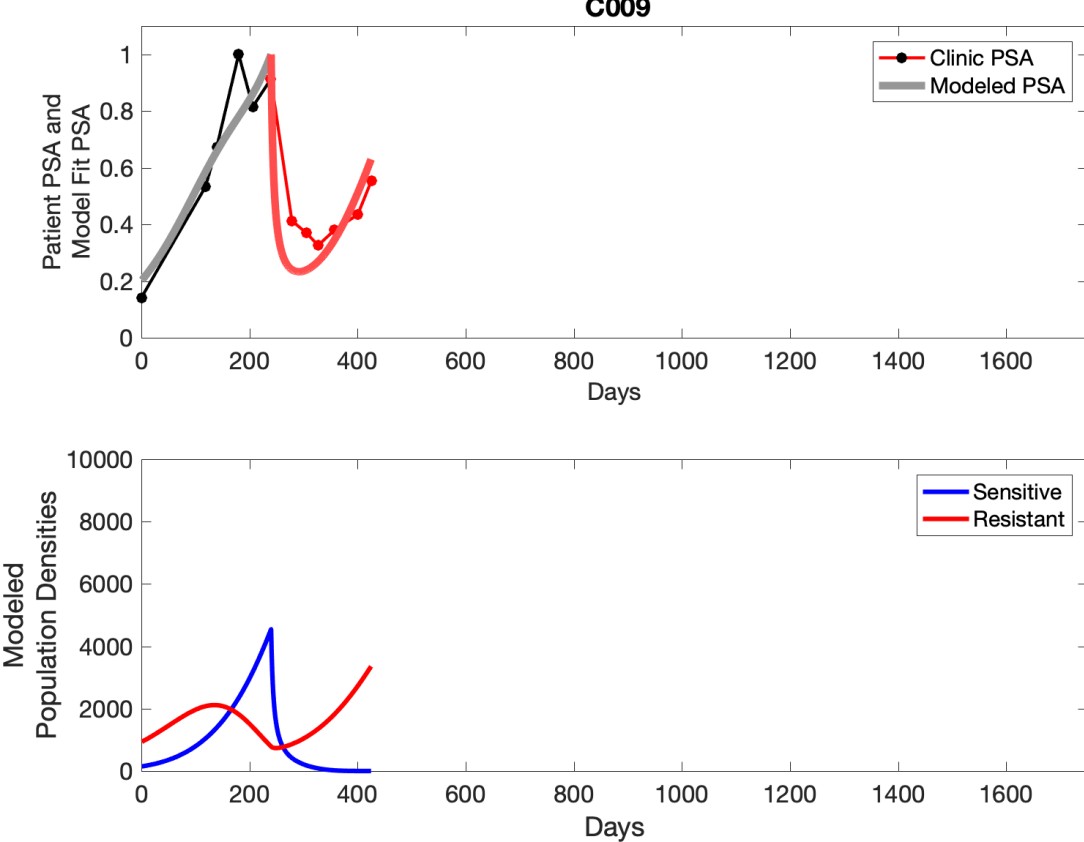

**Appendix 3—figure 9.** Observed PSA dynamics and model fit for the designated subject are shown in the top panel, and model simulations estimating changes in sensitive and resistant cell population are in the lower panel.

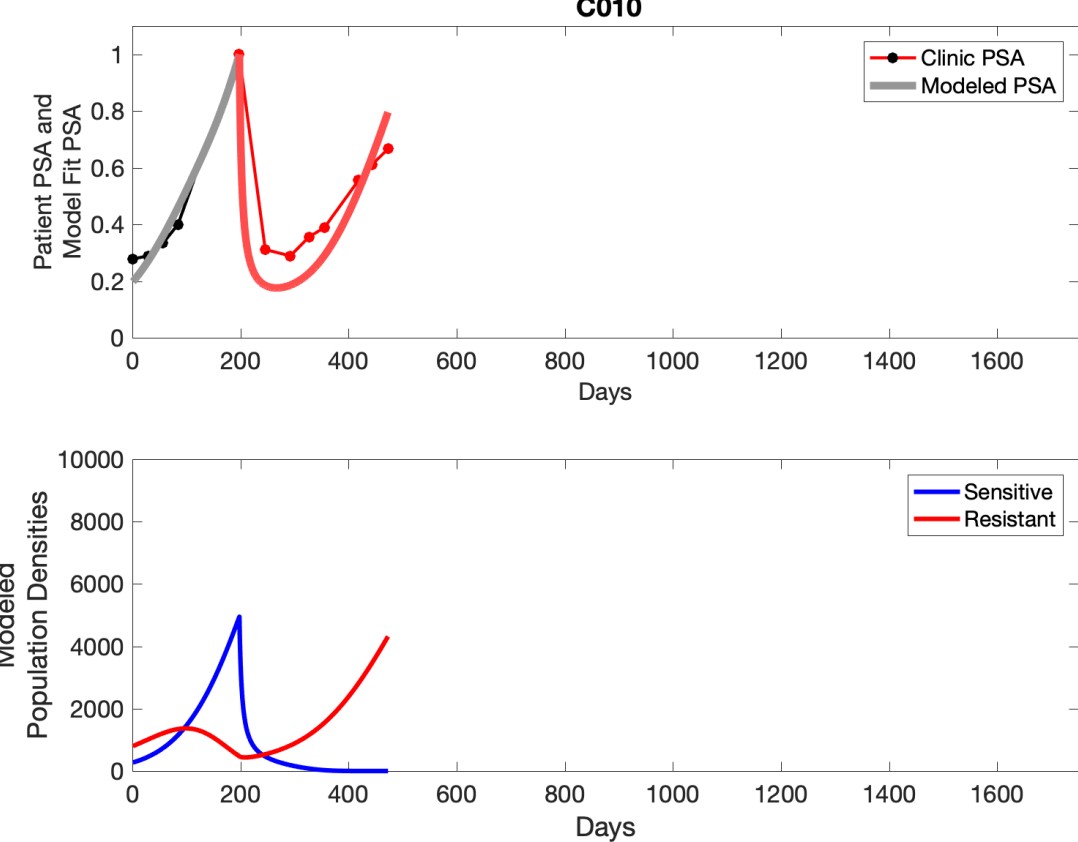

**Appendix 3—figure 10.** Observed PSA dynamics and model fit for the designated subject are shown in the top panel, and model simulations estimating changes in sensitive and resistant cell populations are in the lower panel.

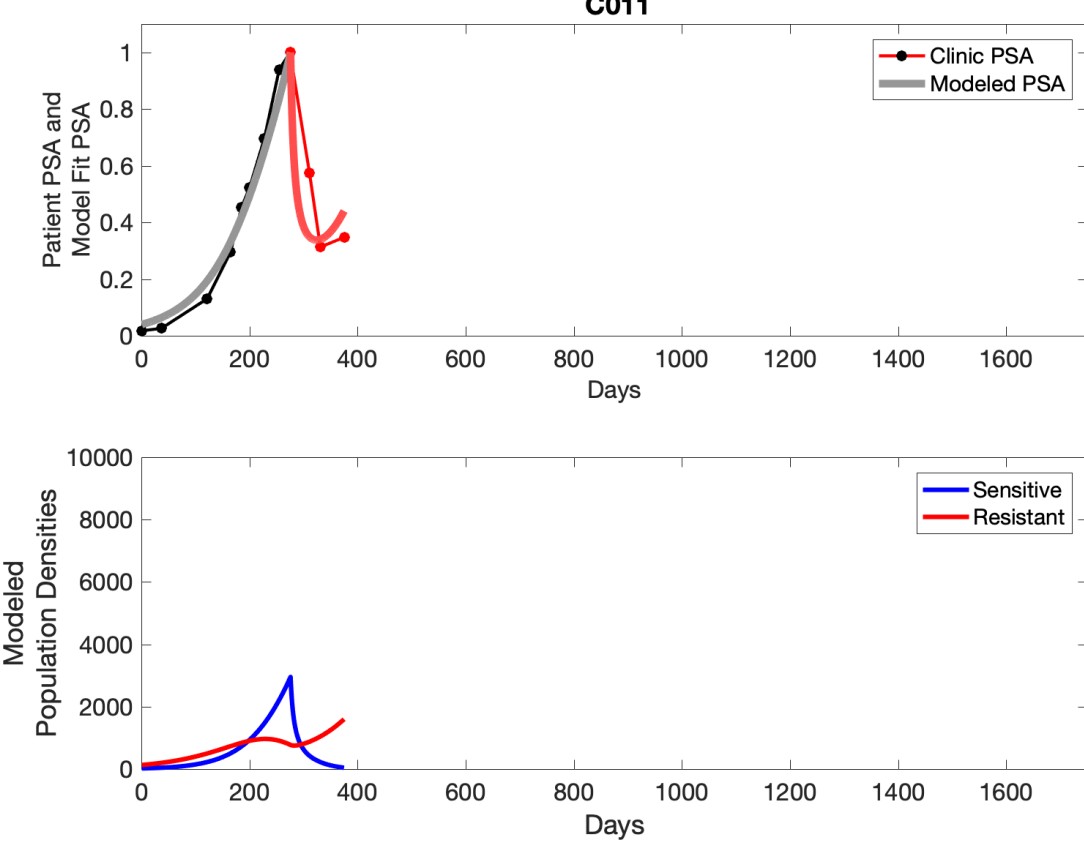

**Appendix 3—figure 11.** Observed PSA dynamics and model fit for the designated subject are shown in the top panel, and model simulations estimating changes in sensitive and resistant cell populations are in the lower panel.

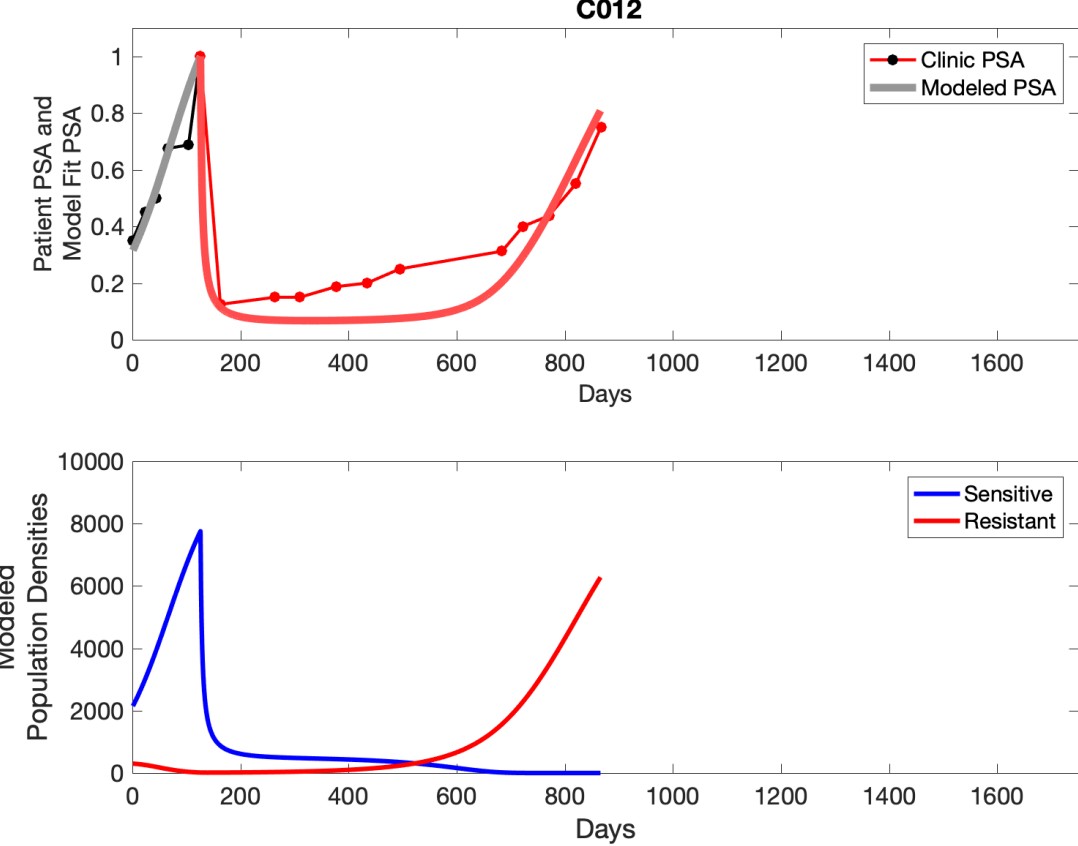

**Appendix 3—figure 12.** Observed PSA dynamics and model fit for the designated subject are shown in the top panel, and model simulations estimating changes in sensitive and resistant cell populations are in the lower panel.

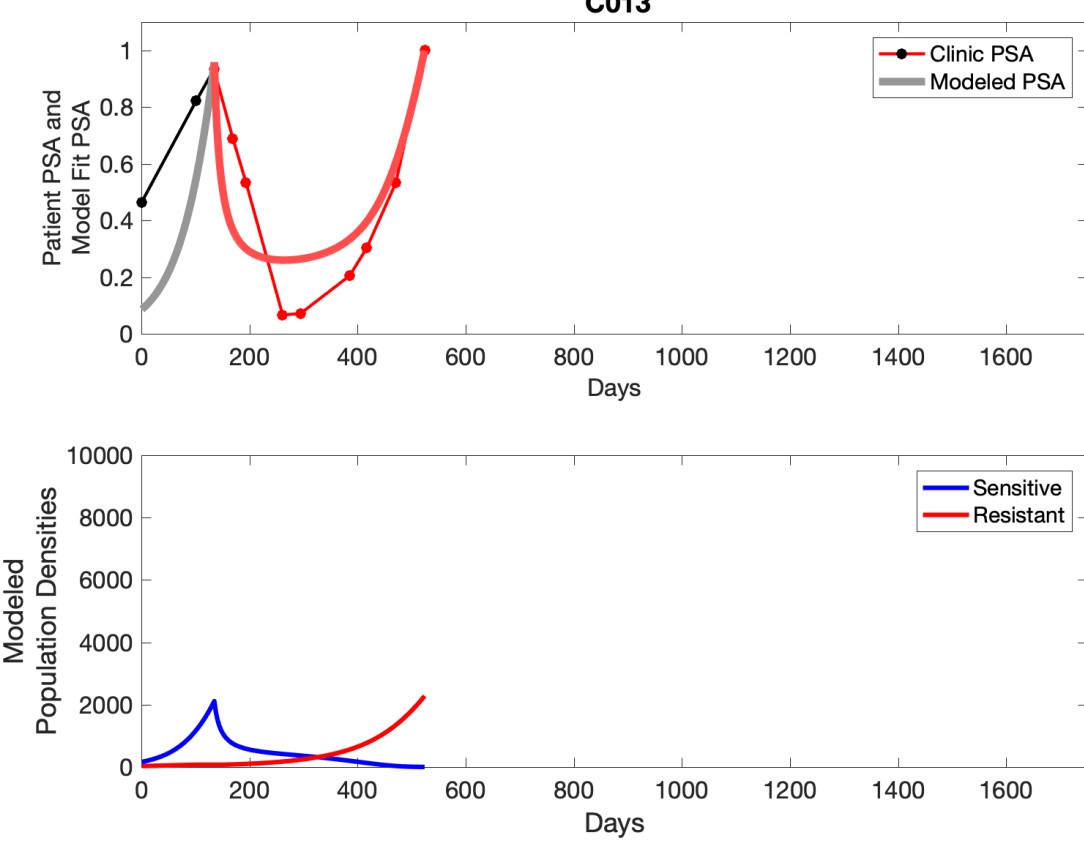

**Appendix 3—figure 13.** Observed PSA dynamics and model fit for the designated subject are shown in the top panel, and model simulations estimating changes in sensitive and resistant cell populations are in the lower panel.

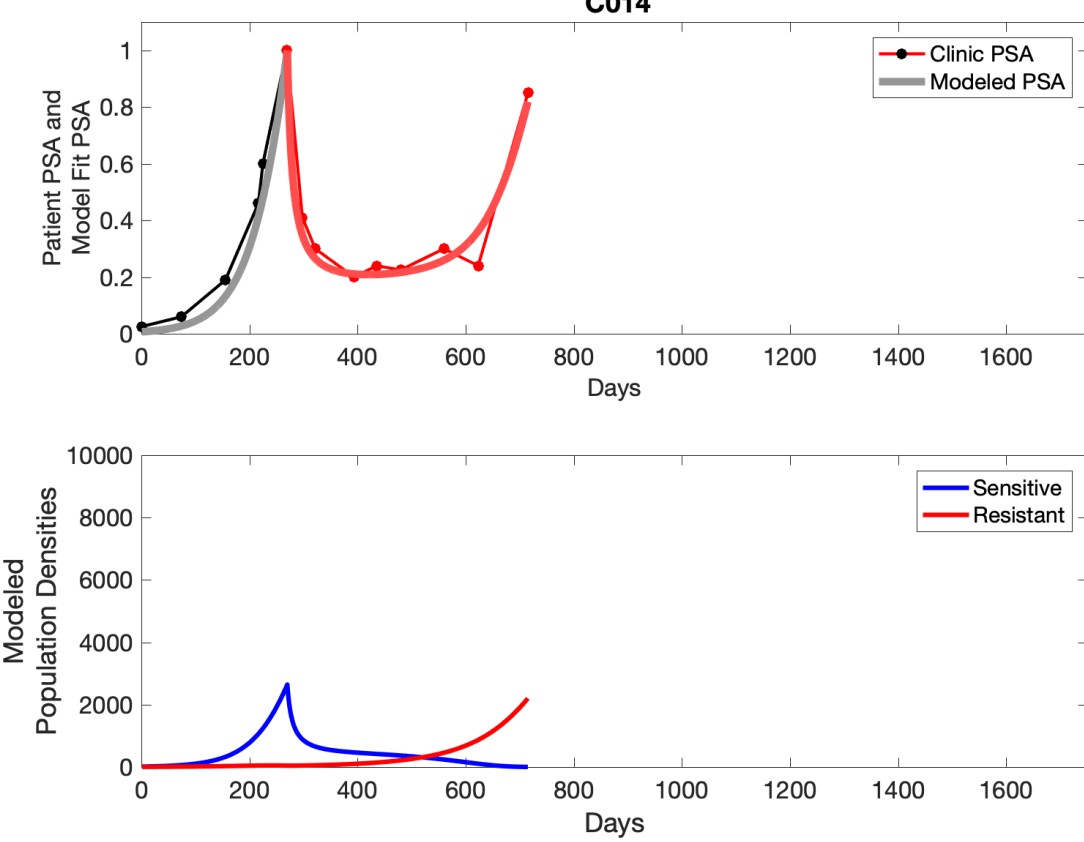

**Appendix 3—figure 14.** Observed PSA dynamics and model fit for the designated subject are shown in the top panel, and model simulations estimating changes in sensitive and resistant cell populations are in the lower panel.

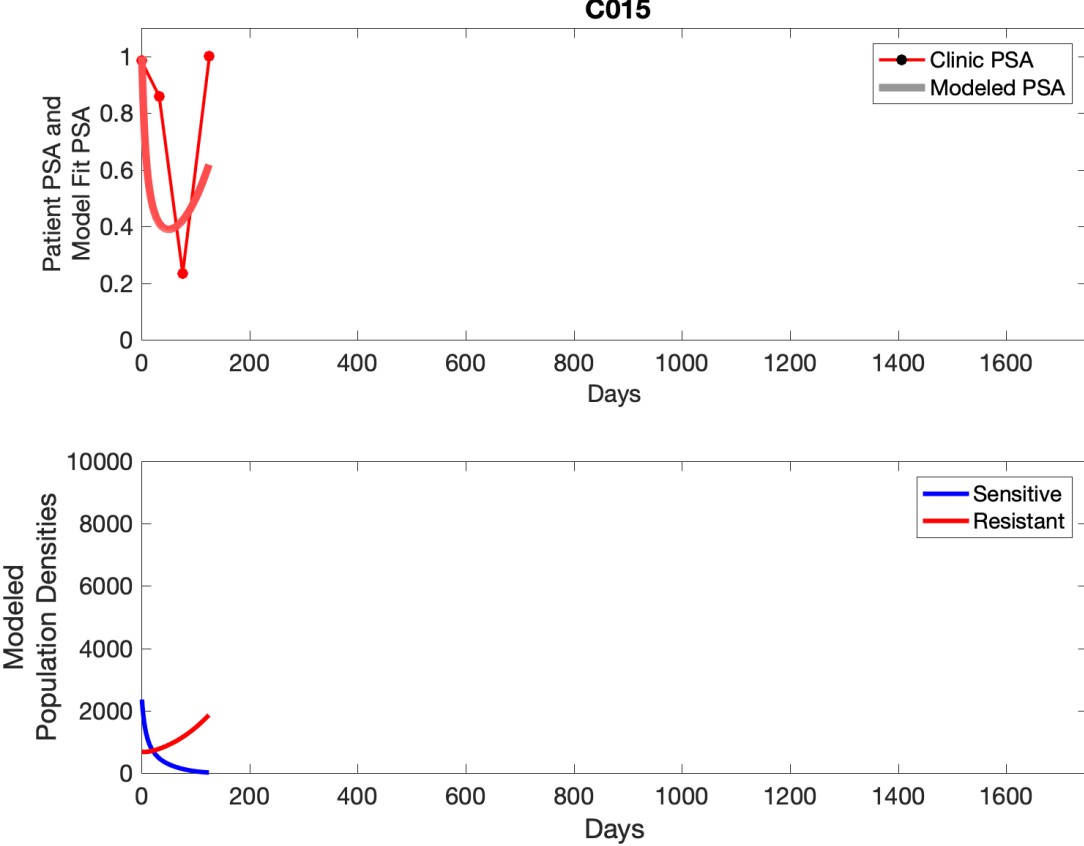

**Appendix 3—figure 15.** Observed PSA dynamics and model fit for the designated subject are shown in the top panel, and model simulations estimating changes in sensitive and resistant cell population are in the lower panel.

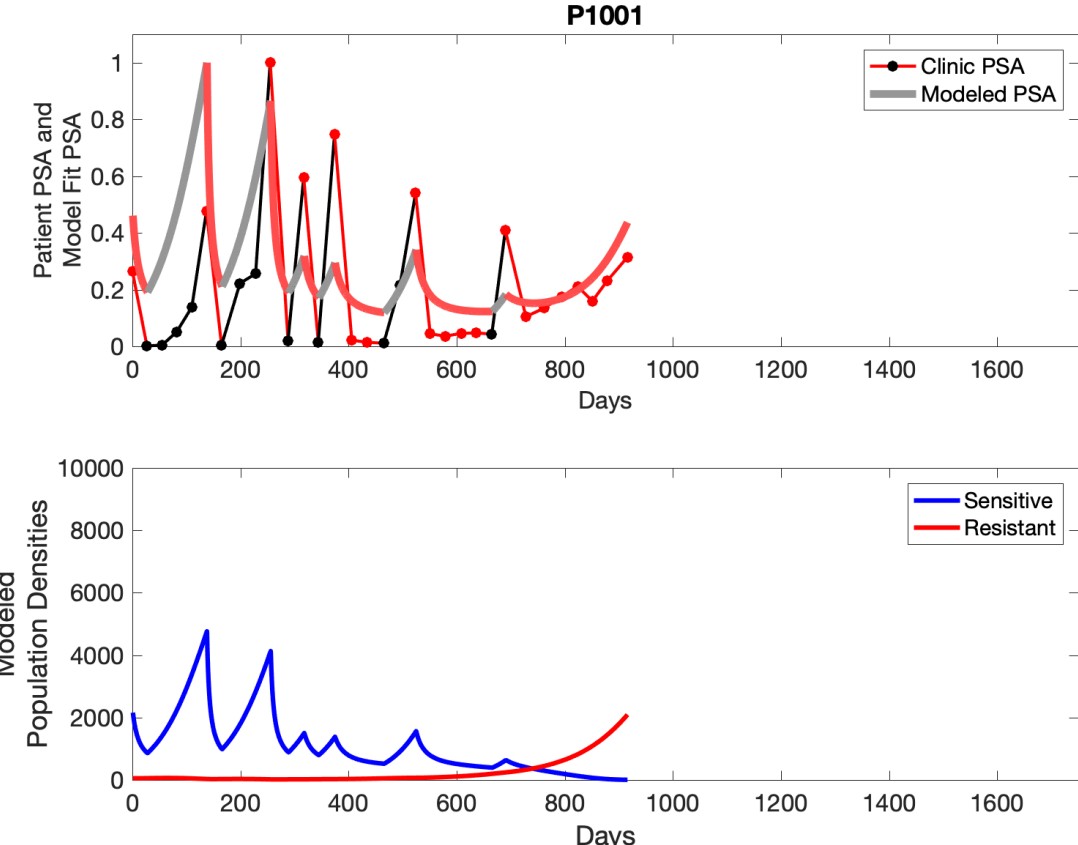

**Appendix 3—figure 16.** Observed PSA dynamics and model fit for the designated subject are shown in the top panel, and model simulations estimating changes in sensitive and resistant cell populations are in the lower panel.

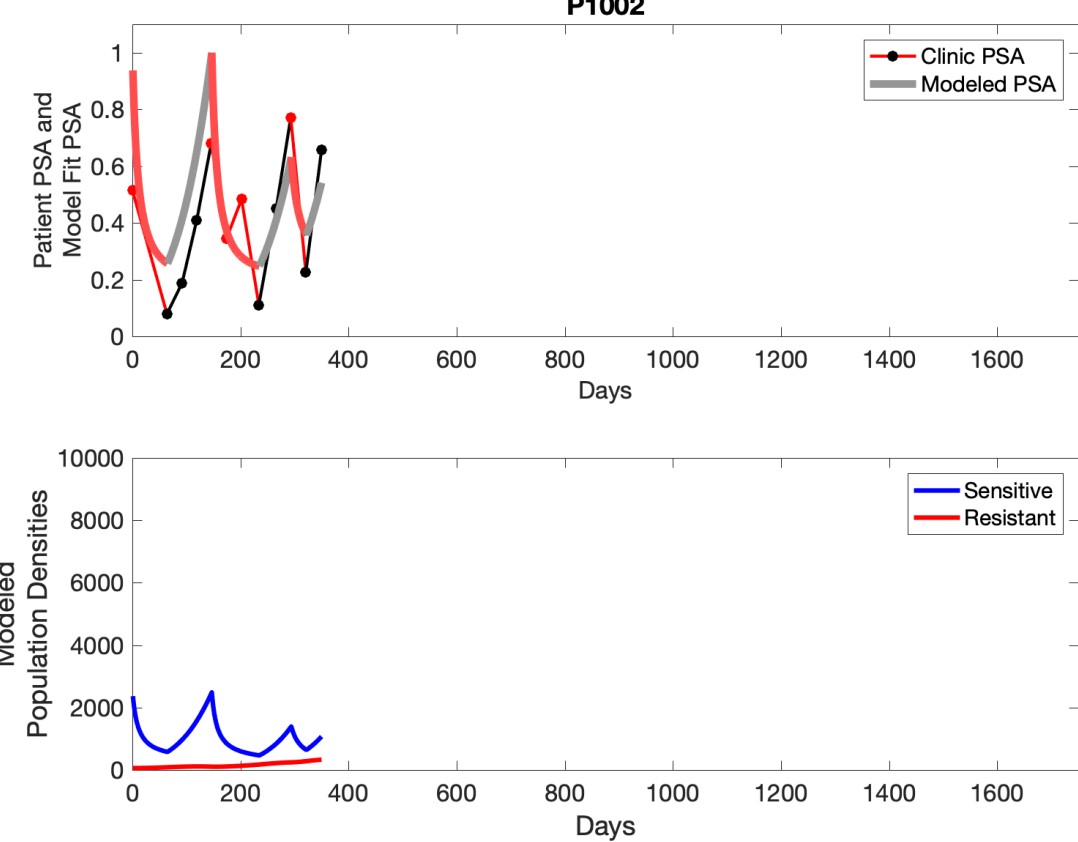

**Appendix 3—figure 17.** Observed PSA dynamics and model fit for the designated subject are shown in the top panel, and model simulations estimating changes in sensitive and resistant cell populations are in the lower panel.

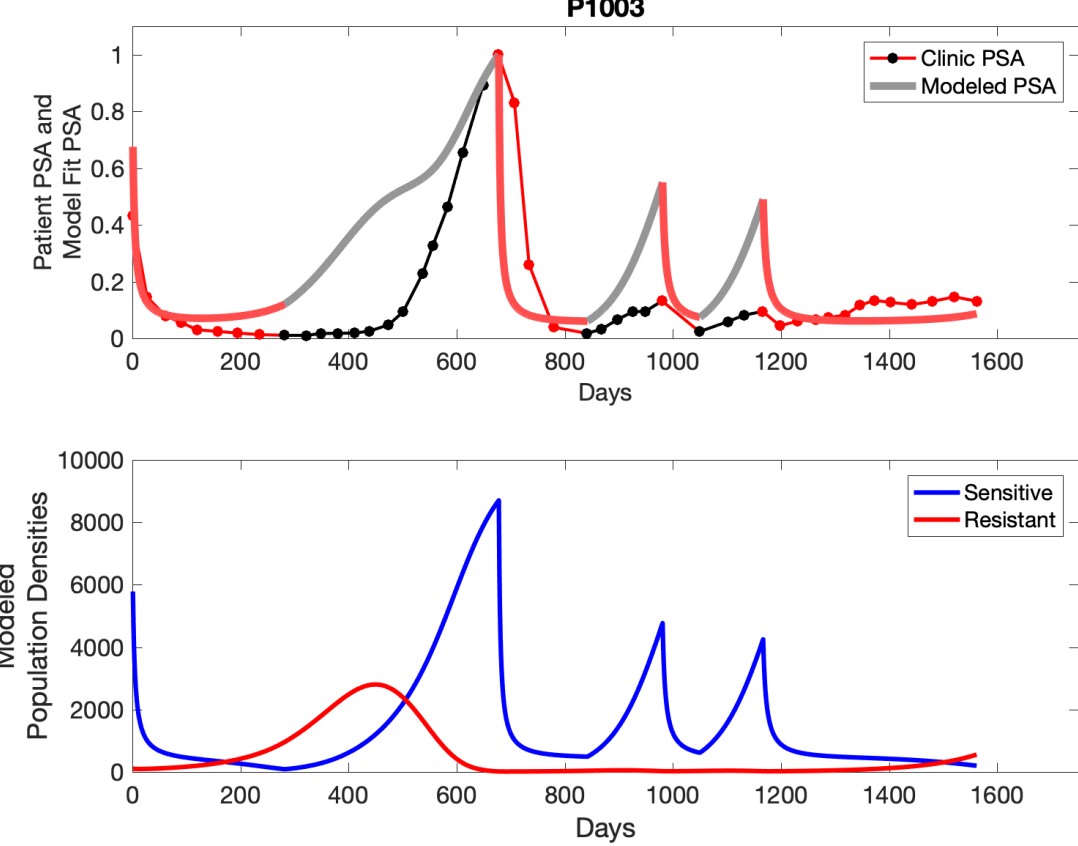

**Appendix 3—figure 18.** Observed PSA dynamics and model fit for the designated subject are shown in the top panel, and model simulations estimating changes in sensitive and resistant cell populations are in the lower panel.

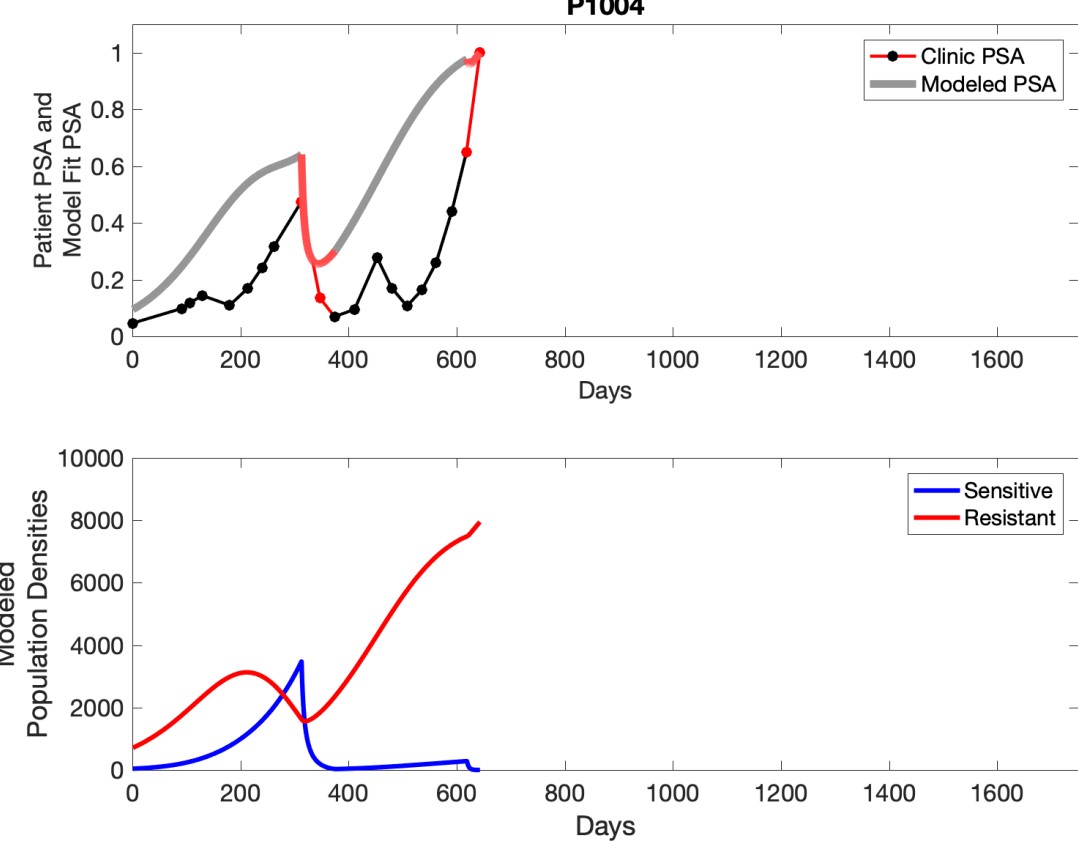

**Appendix 3—figure 19.** Observed PSA dynamics and model fit for the designated subject are shown in the top panel, and model simulations estimating changes in sensitive and resistant cell populations are in the lower panel.

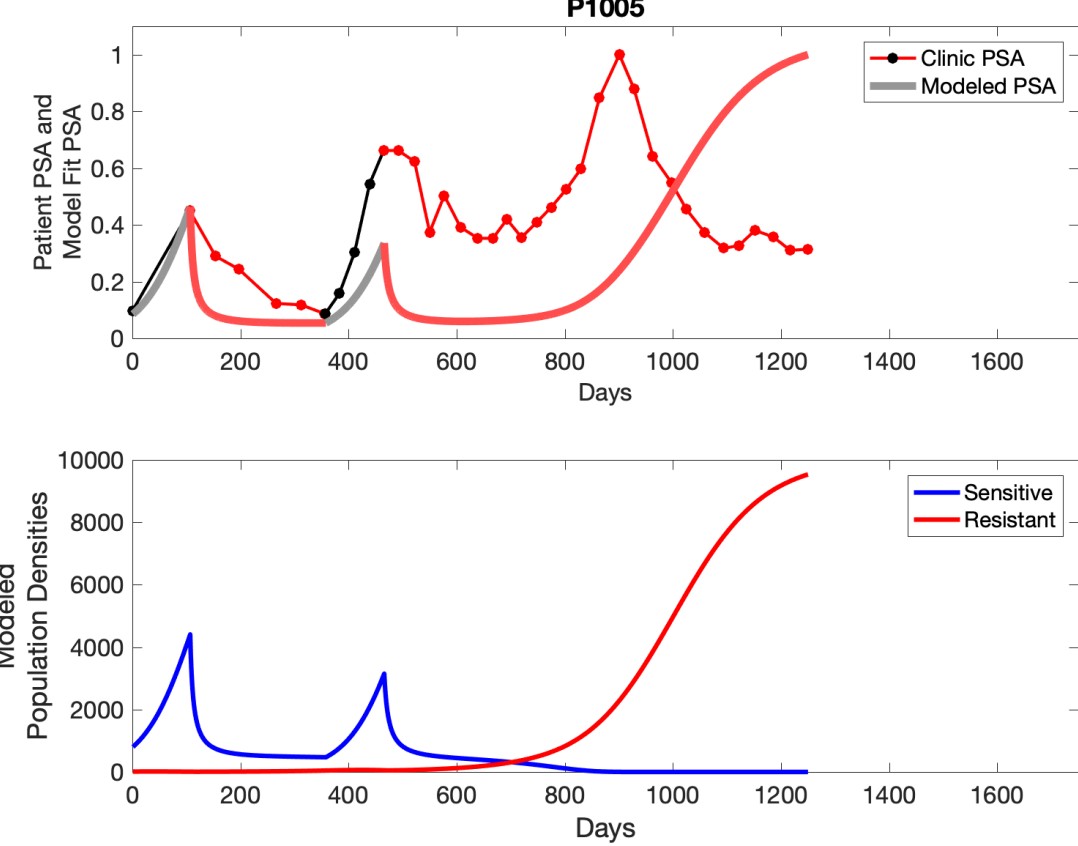

**Appendix 3—figure 20.** Observed PSA dynamics and model fit for the designated subject are shown in the top panel, and model simulations estimating changes in sensitive and resistant cell populations are in the lower panel.

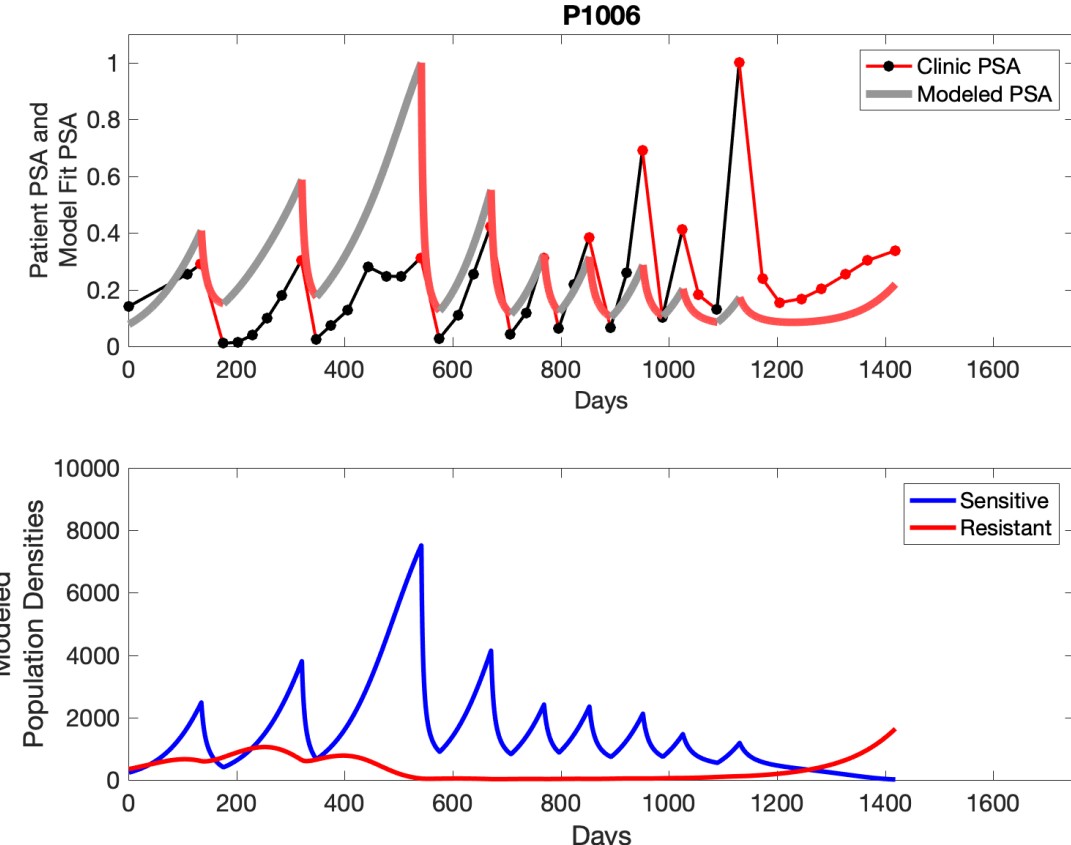

**Appendix 3—figure 21.** Observed PSA dynamics and model fit for the designated subject are shown in the top panel, and model simulations estimating changes in sensitive and resistant cell populations are in the lower panel.

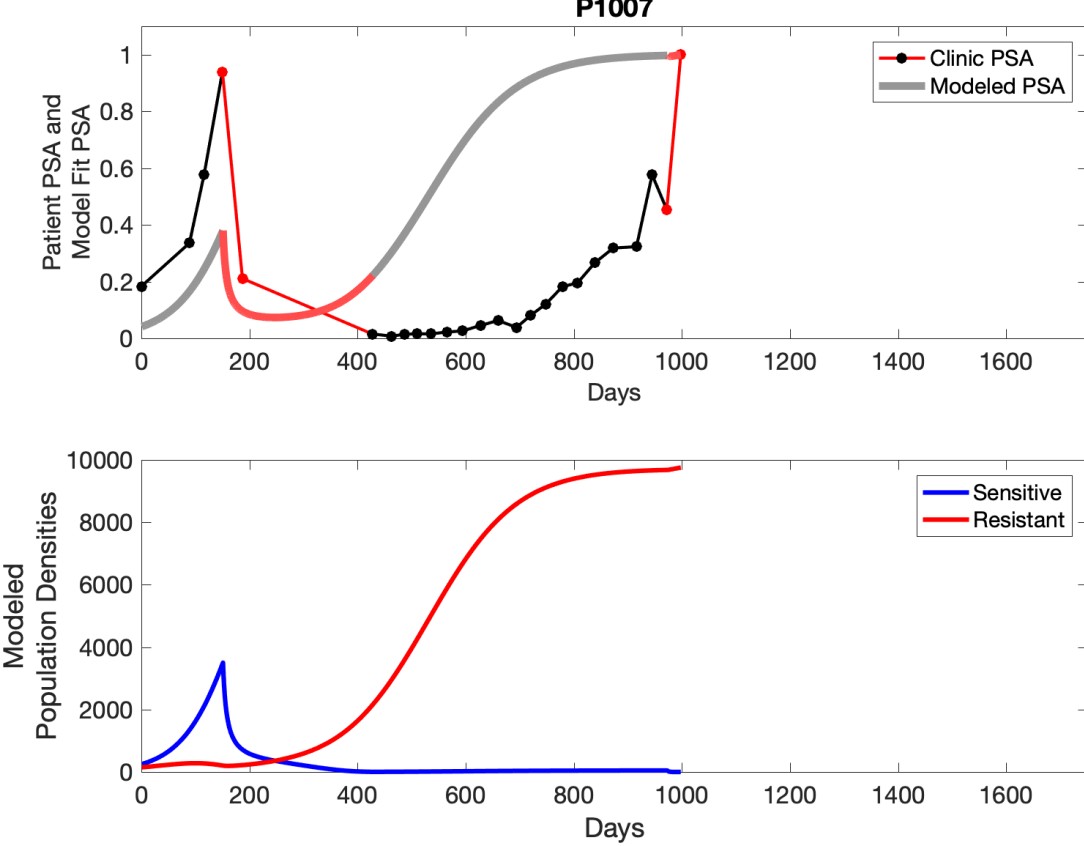

**Appendix 3—figure 22.** Observed PSA dynamics and model fit for the designated subject are shown in the top panel, and model simulations estimating changes in sensitive and resistant cell populations are in the lower panel.

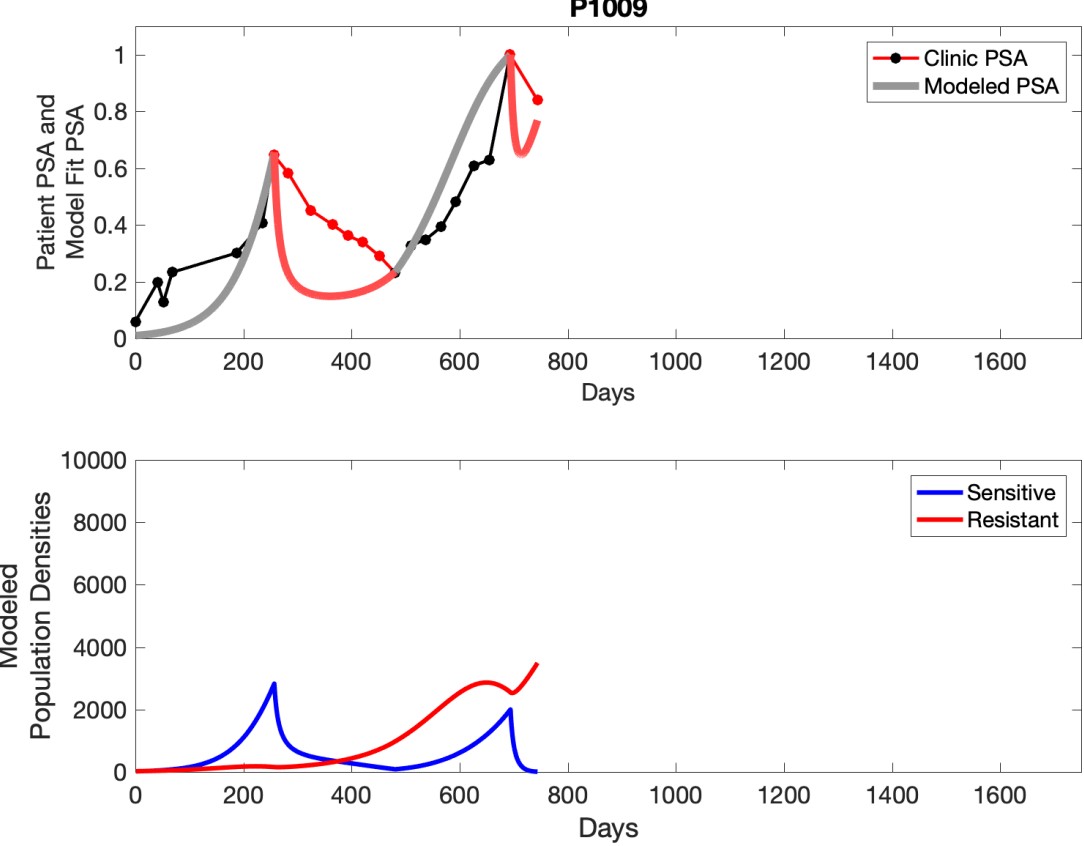

**Appendix 3—figure 23.** Observed PSA dynamics and model fit for the designated subject are shown in the top panel, and model simulations estimating changes in sensitive and resistant cell populations are in the lower panel.

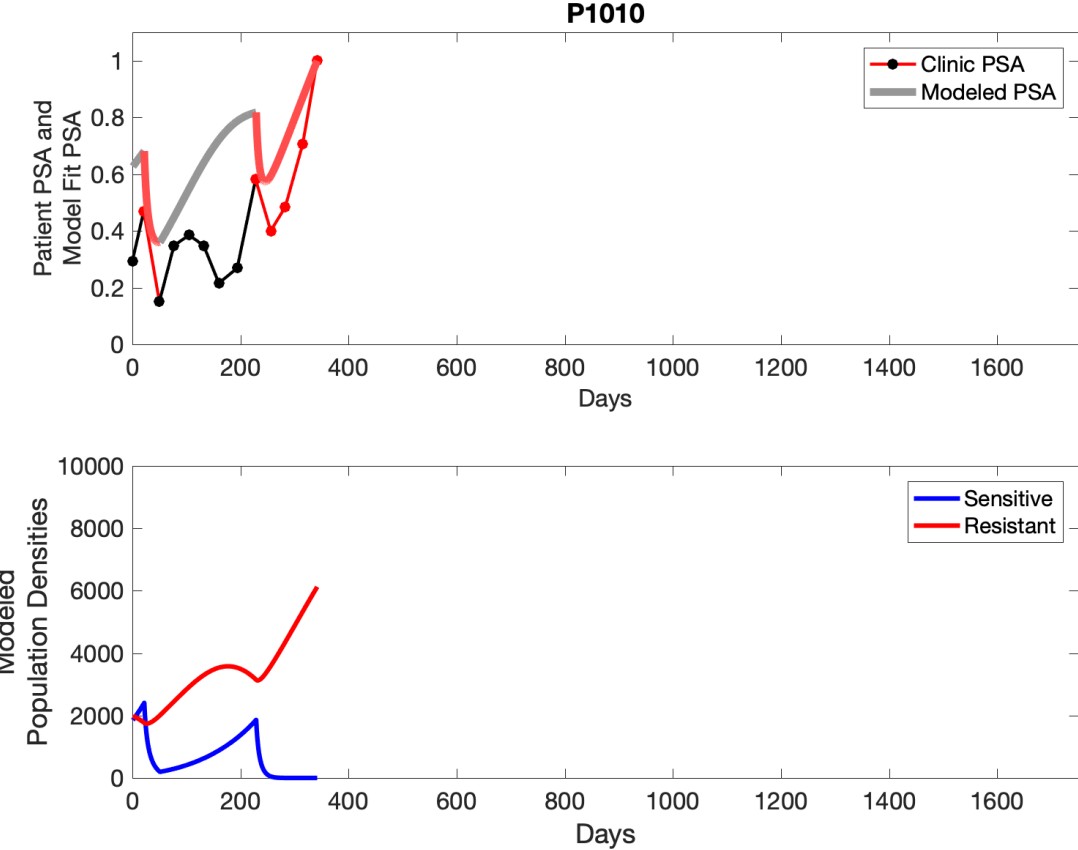

**Appendix 3—figure 24.** Observed PSA dynamics and model fit for the designated subject are shown in the top panel, and model simulations estimating changes in sensitive and resistant cell populations are in the lower panel.

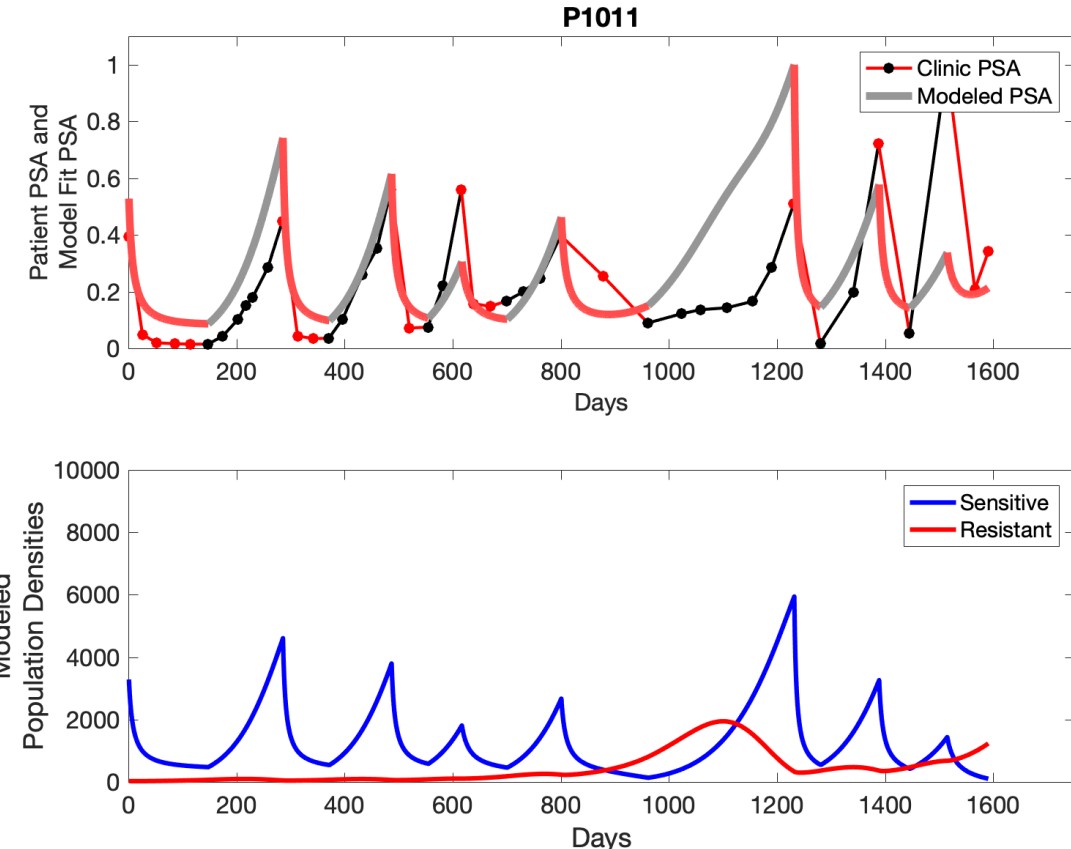

**Appendix 3—figure 25.** Observed PSA dynamics and model fit for the designated subject are shown in the top panel, and model simulations estimating changes in sensitive and resistant cell populations are in the lower panel.

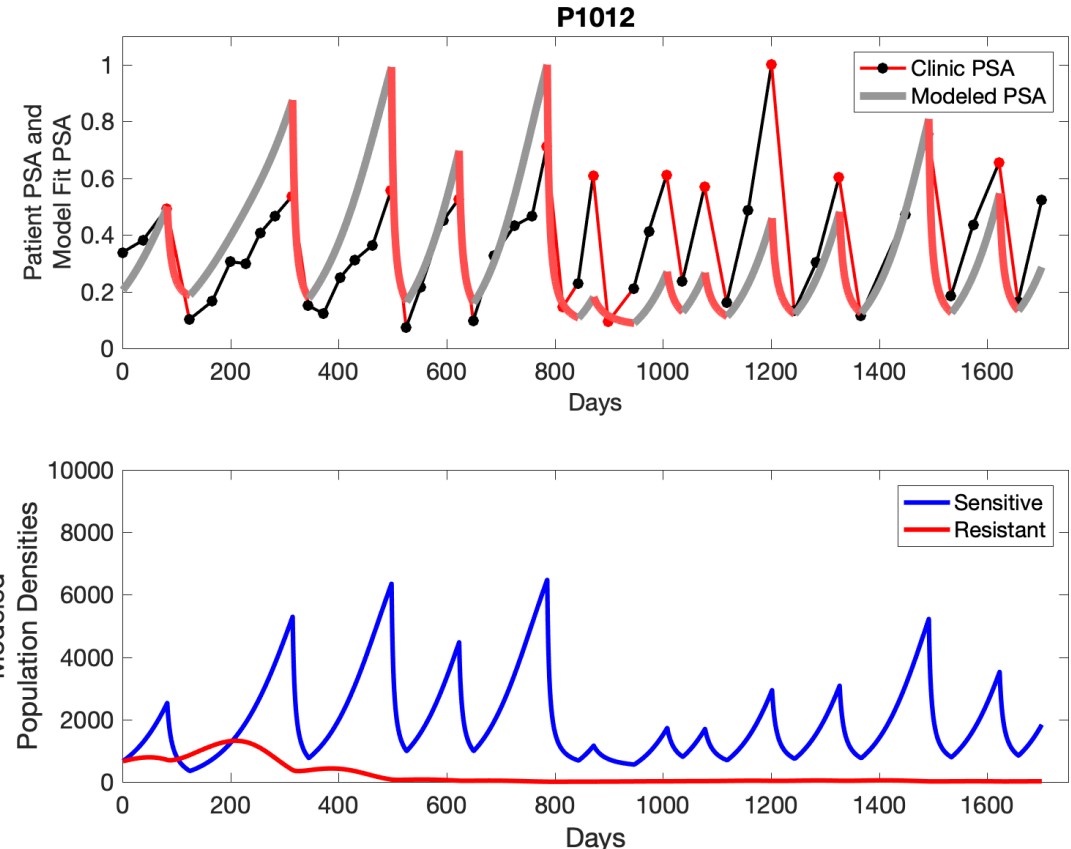

**Appendix 3—figure 26.** Observed PSA dynamics and model fit for the designated subject are shown in the top panel, and model simulations estimating changes in sensitive and resistant cell populations are in the lower panel.

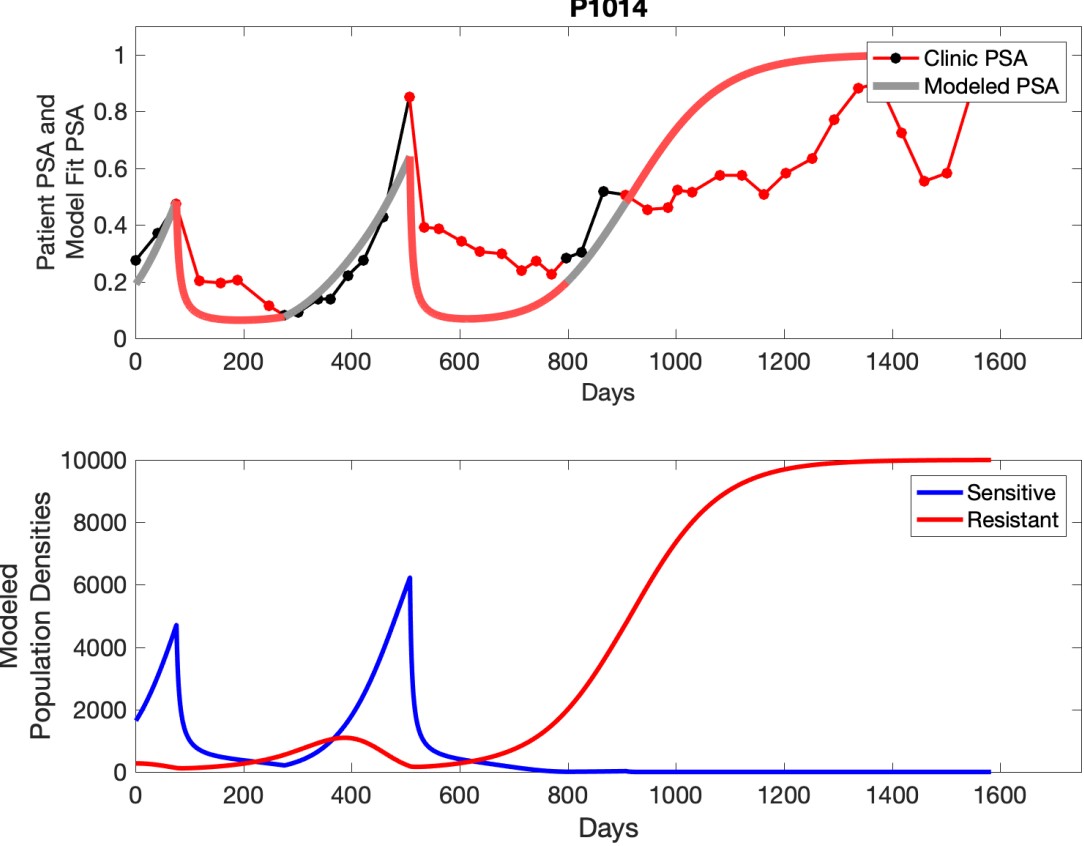

**Appendix 3—figure 27.** Observed PSA dynamics and model fit for the designated subject are shown in the top panel, and model simulations estimating changes in sensitive and resistant cell populations are in the lower panel.

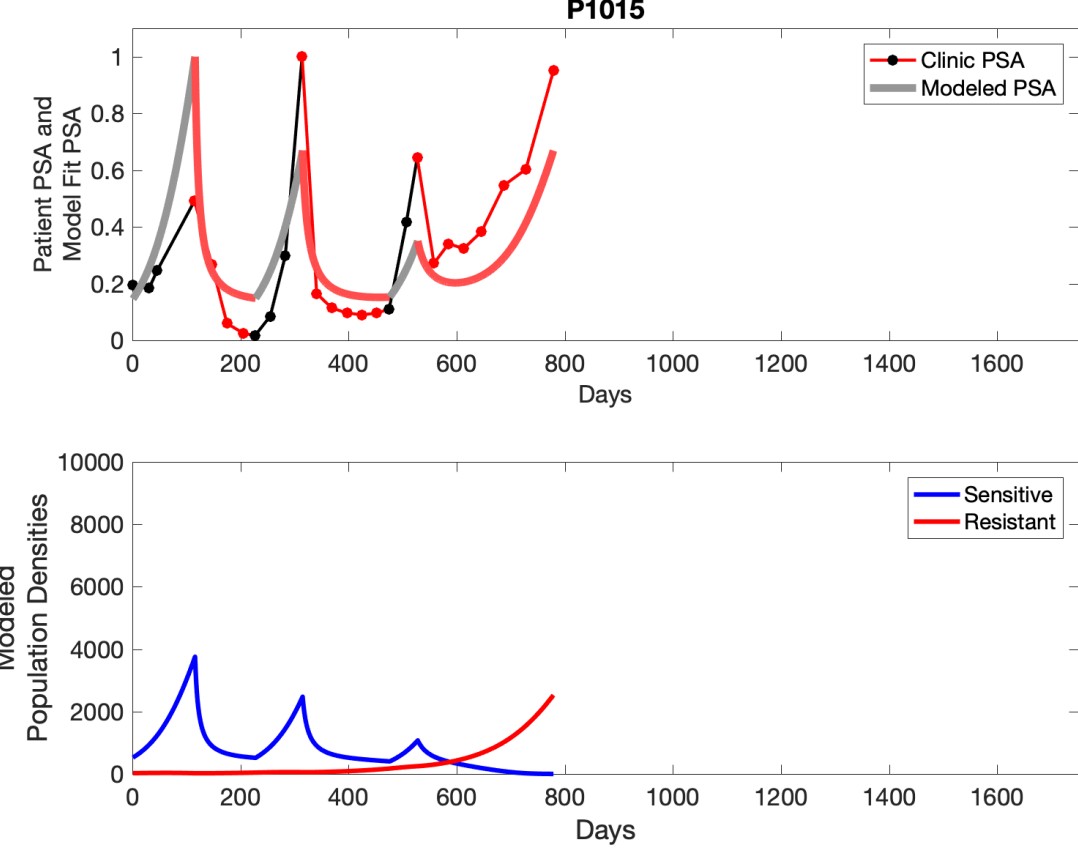

**Appendix 3—figure 28.** Observed PSA dynamics and model fit for the designated subject are shown in the top panel, and model simulations estimating changes in sensitive and resistant cell populations are in the lower panel.

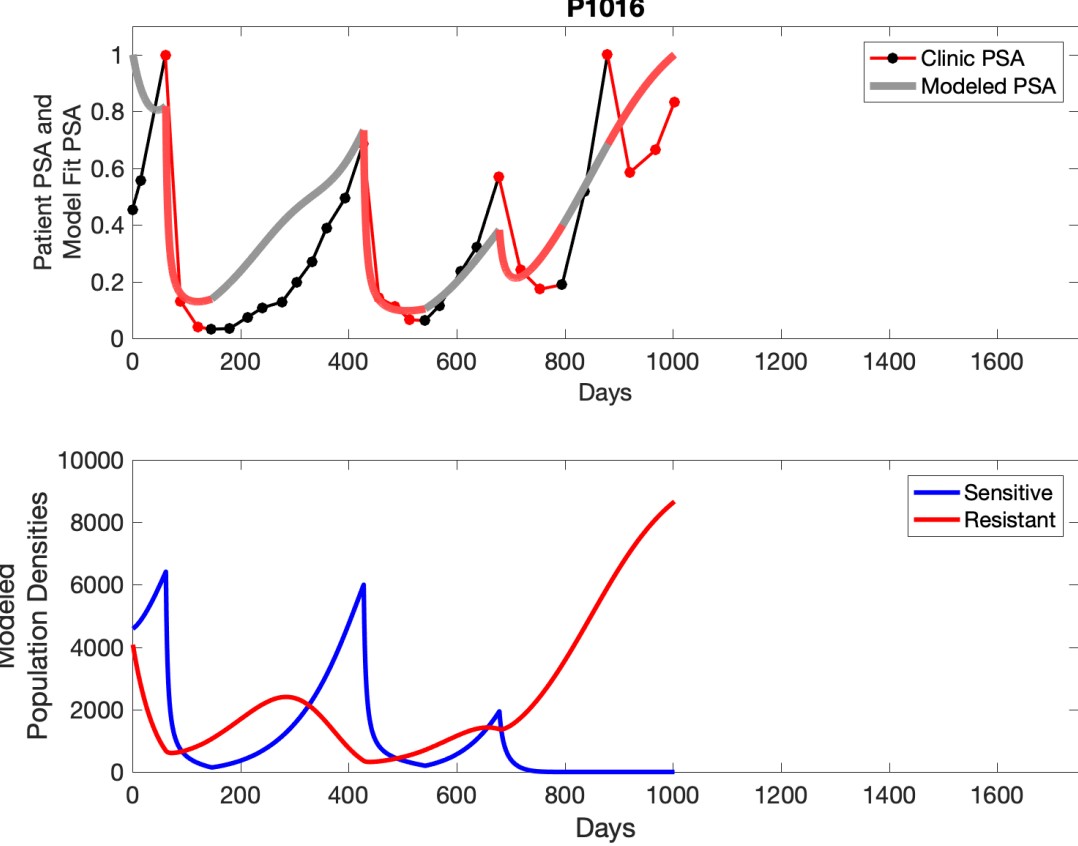

**Appendix 3—figure 29.** Observed PSA dynamics and model fit for the designated subject are shown in the top panel, and model simulations estimating changes in sensitive and resistant cell populations are in the lower panel.

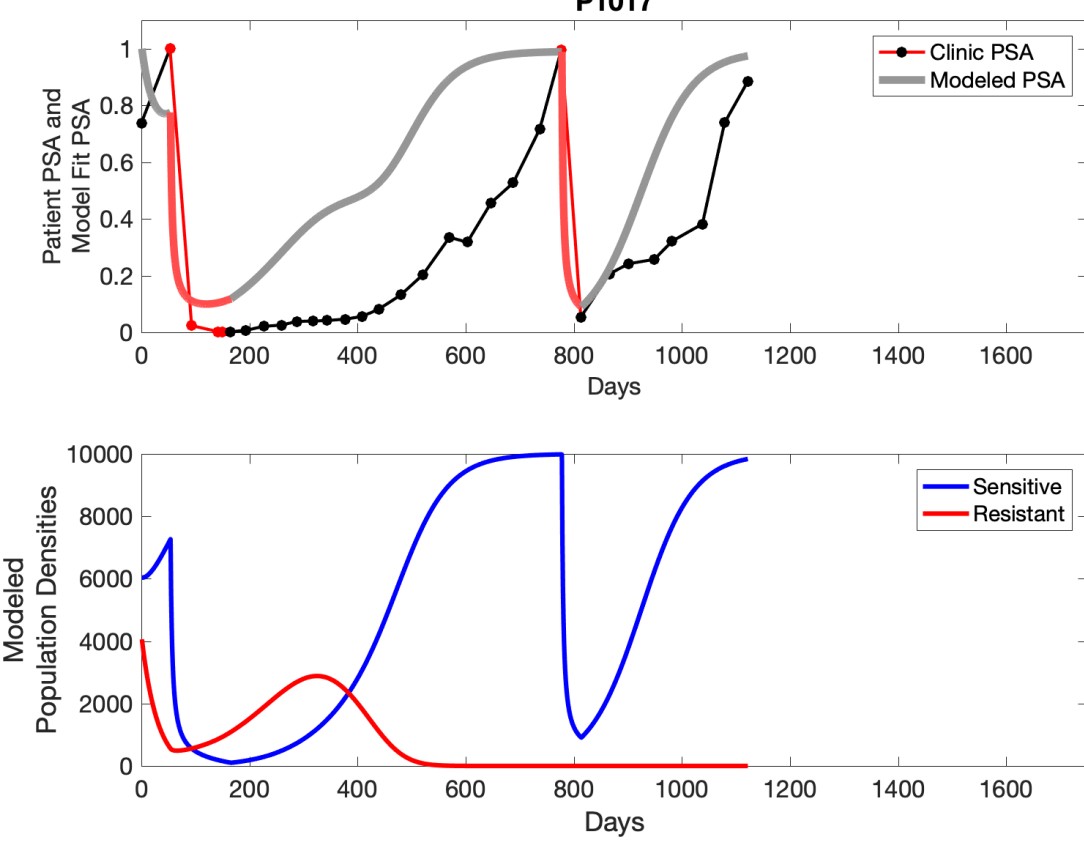

**Appendix 3—figure 30.** Observed PSA dynamics and model fit for the designated subject are shown in the top panel, and model simulations estimating changes in sensitive and resistant cell populations are in the lower panel.

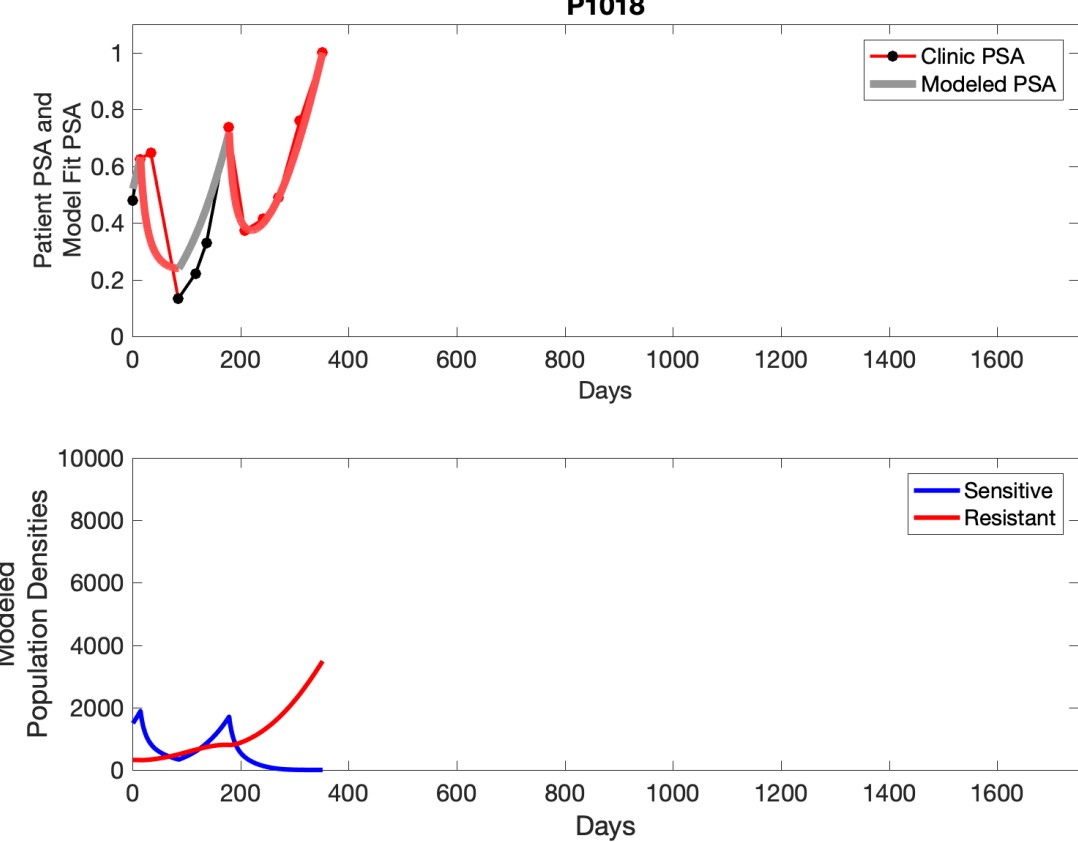

**Appendix 3—figure 31.** Observed PSA dynamics and model fit for the designated subject are shown in the top panel, and model simulations estimating changes in sensitive and resistant cell populations are in the lower panel.

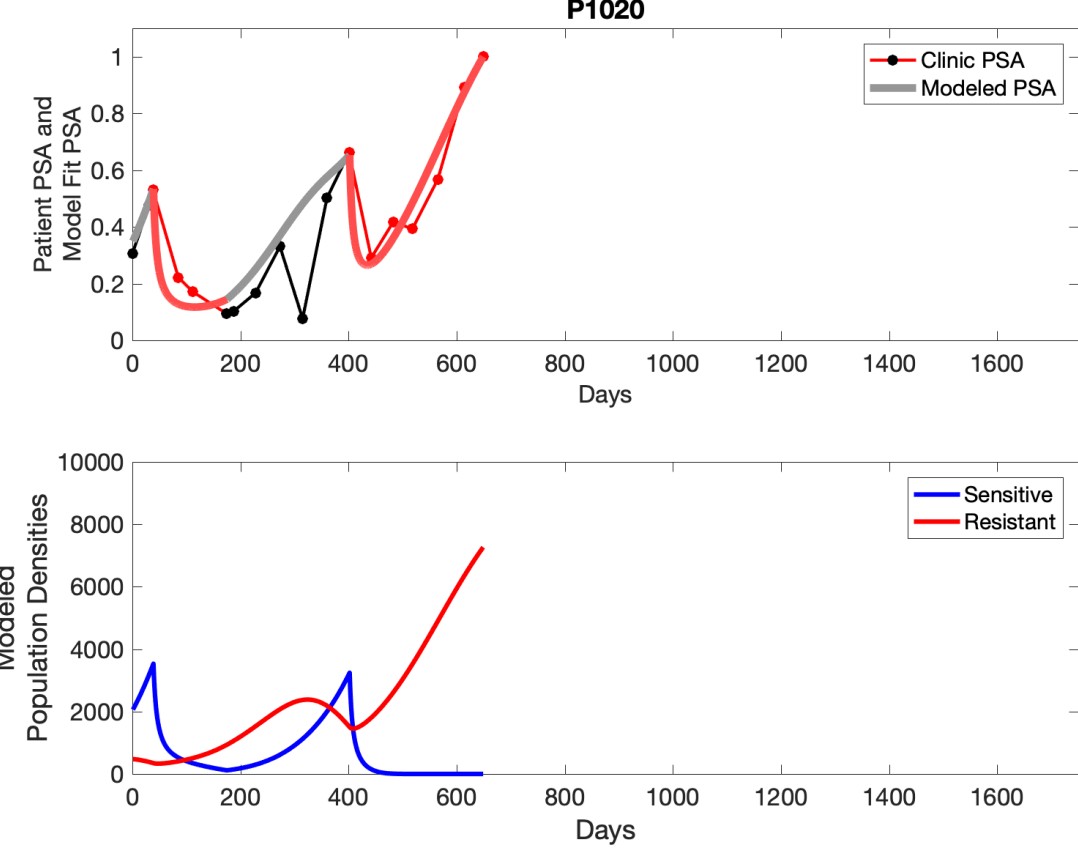

**Appendix 3—figure 32.** Observed PSA dynamics and model fit for the designated subject are shown in the top panel, and model simulations estimating changes in sensitive and resistant cell population are in the lower panel.

# Appendix 4

## Simulated idealized adaptive therapy

To evaluate an ideal 50% adaptive therapy protocol, we ran a simulation for each patient (both cohorts). In line with the trial, abiraterone was administered until there was a 50% drop in the patient's PSA at which point therapy ceased until PSA levels returned to the patient's initial value at which point therapy was resumed, and so on. If the PSA ceased to decline to 50% of the initial PSA, than abiraterone was continued indefinitely. For patients in the contemporaneous cohort, we can predict how they might have fared under adaptive therapy. For the patients that clinically received adaptive therapy, we can see how they would have fared if the therapy had been idealized to start and stop at the exact switch points. In reality, the start and stop to abiraterone often occurred at PSA levels higher than the initial level and lower than the 50% threshold, respectively. This treatment protocol was run for each patient using their patient-specific parameters from *Appendix 2—table 3*.

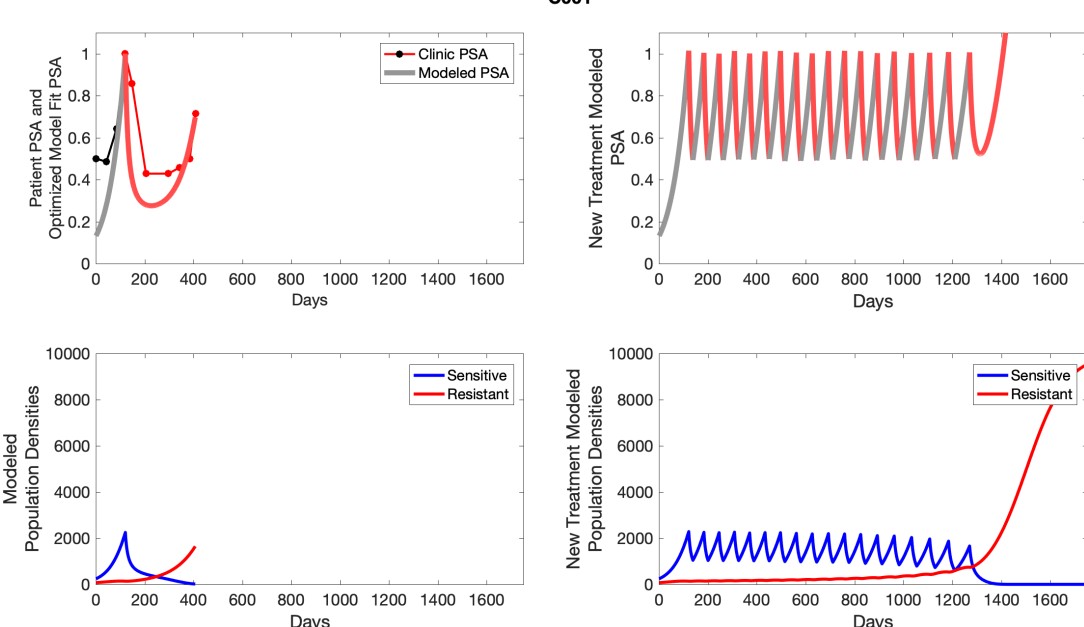

**Appendix 4—figure 1.** The left column shows actual outcome (upper panel) and computer simulations (lower panel) of the estimated population dynamics of the sensitive and resistant poulations for the indicated subject; the right column shows computer simulations based on an assumption that treatment was stopped immediately when the PSA fell below 0.5 of the pre-treatment and resumed immediately when PSA increased to the pretreatment value.

**C002**

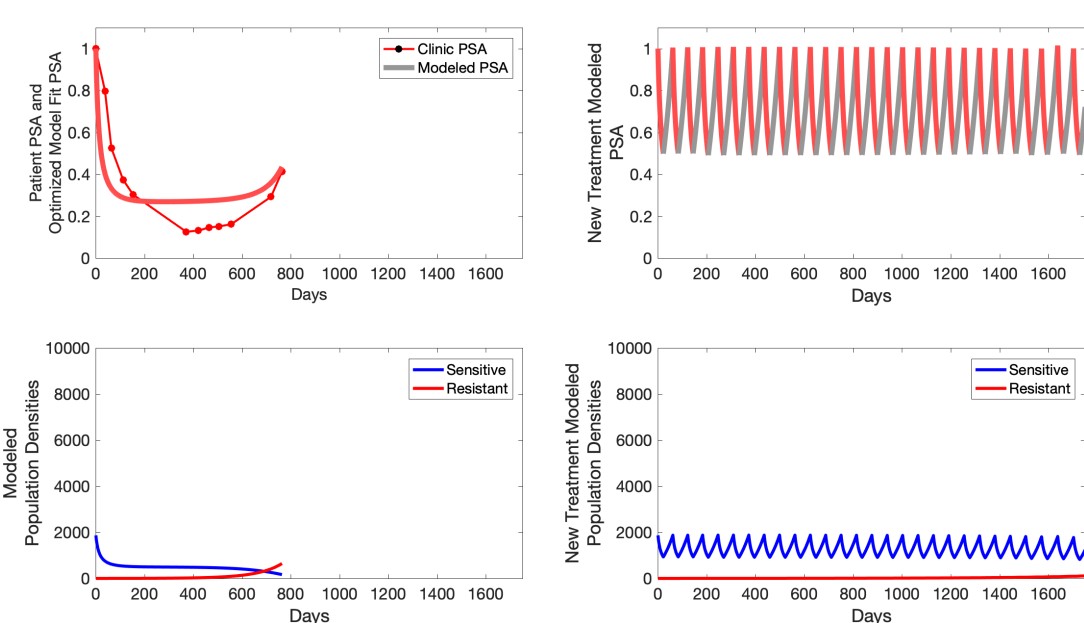

**Appendix 4—figure 2.** The left column shows actual outcome (upper panel) and computer simulations (lower panel) of the estimated population dynamics of the sensitive and resistant poulations for the indicated subject; the right column shows computer simulations based on an assumption that treatment was stopped immediately when the PSA fell below 0.5 of the pre-treatment and resumed immediately when PSA increased to the pretreatment value.

**C003**

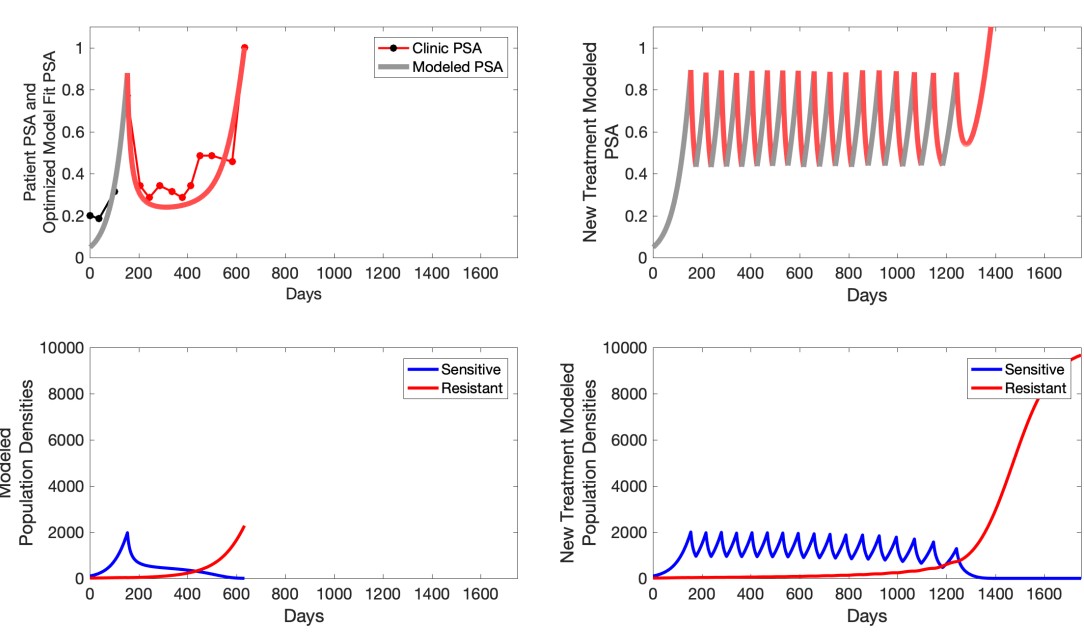

**Appendix 4—figure 3.** The left column shows actual outcome (upper panel) and computer simulations (lower panel) of the estimated population dynamics of the sensitive and resistant poulations for the indicated subject; the right column shows computer simulations based on an assumption that treatment was stopped immediately when the PSA fell below 0.5 of the pre-treatment and resumed immediately when PSA increased to the pretreatment value.

**C004**

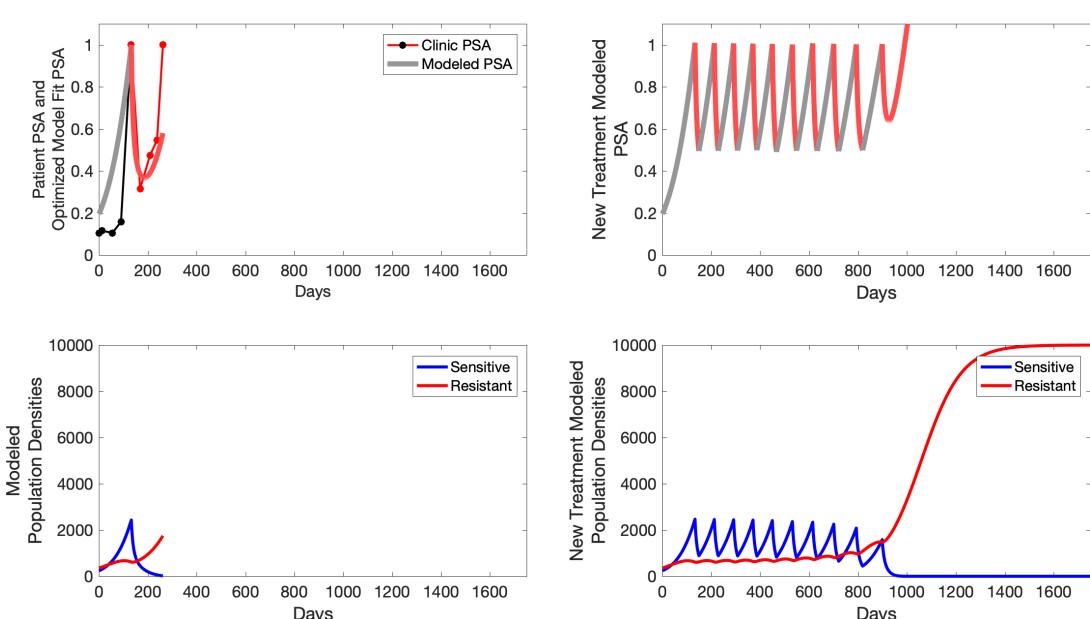

**Appendix 4—figure 4.** The left column shows actual outcome (upper panel) and computer simulations (lower panel) of the estimated population dynamics of the sensitive and resistant poulations for the indicated subject; the right column shows computer simulations based on an assumption that treatment was stopped immediately when the PSA fell below 0.5 of the pre-treatment and resumed immediately when PSA increased to the pretreatment value.

**C005**

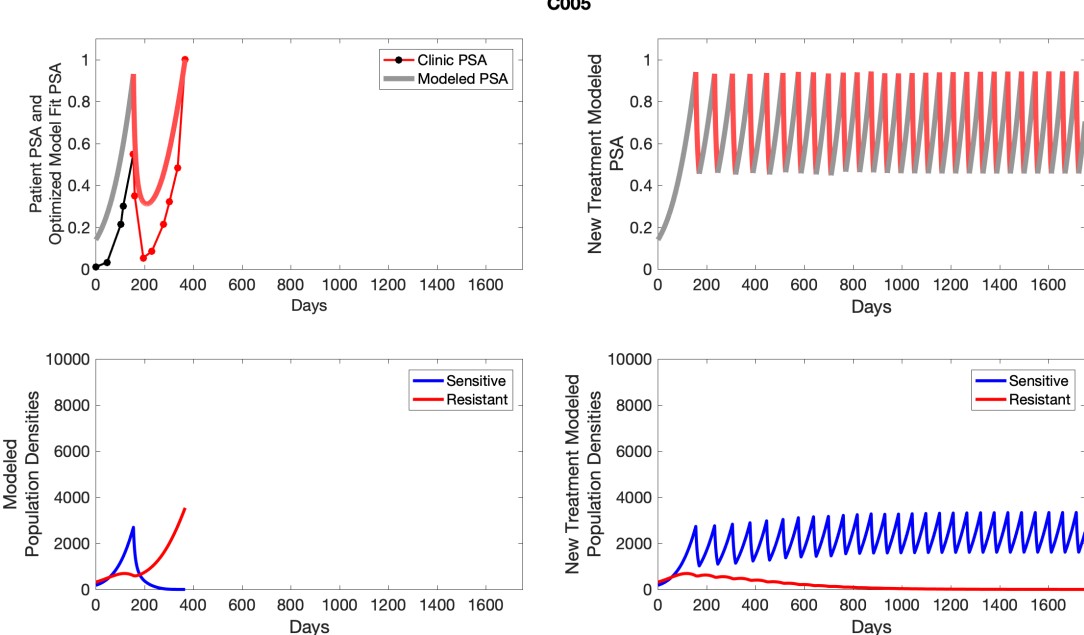

**Appendix 4—figure 5.** The left column shows actual outcome (upper panel) and computer simulations (lower panel) of the estimated population dynamics of the sensitive and resistant poulations for the indicated subject; the right column shows computer simulations based on an assumption that treatment was stopped immediately when the PSA fell below 0.5 of the pre-treatment and resumed immediately when PSA increased to the pretreatment value.

**C006**

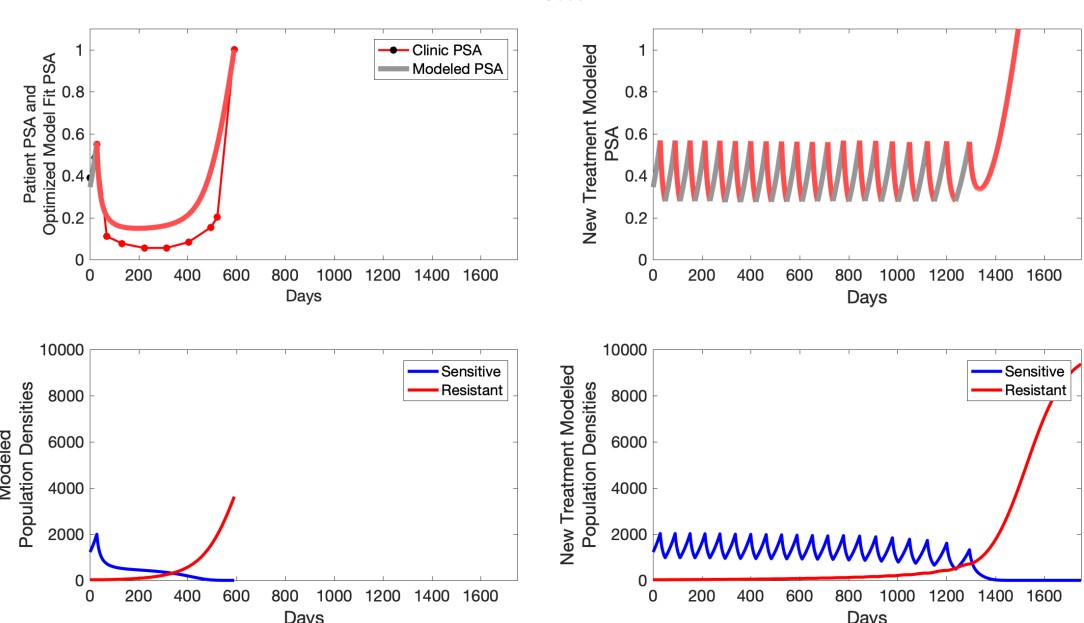

**Appendix 4—figure 6.** The left column shows actual outcome (upper panel) and computer simulations (lower panel) of the estimated population dynamics of the sensitive and resistant poulations for the indicated subject; the right column shows computer simulations based on an assumption that treatment was stopped immediately when the PSA fell below 0.5 of the pre-treatment and resumed immediately when PSA increased to the pretreatment value.

**C007**

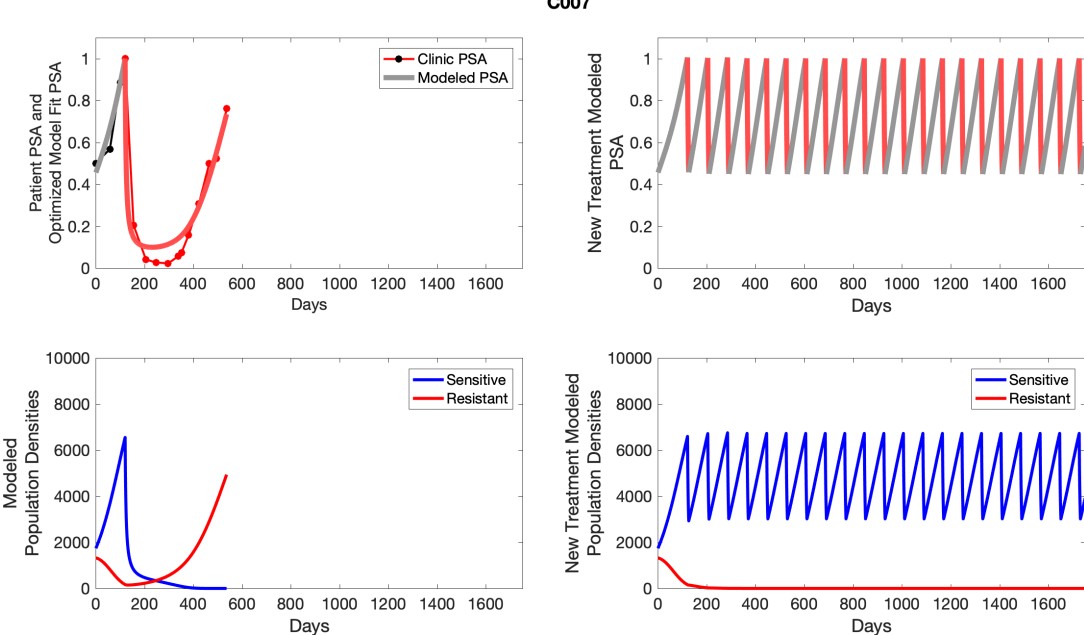

**Appendix 4—figure 7.** The left column shows actual outcome (upper panel) and computer simulations (lower panel) of the estimated population dynamics of the sensitive and resistant poulations for the indicated subject; the right column shows computer simulations based on an assumption that treatment was stopped immediately when the PSA fell below 0.5 of the pre-treatment and resumed immediately when PSA increased to the pretreatment value.

**C008**

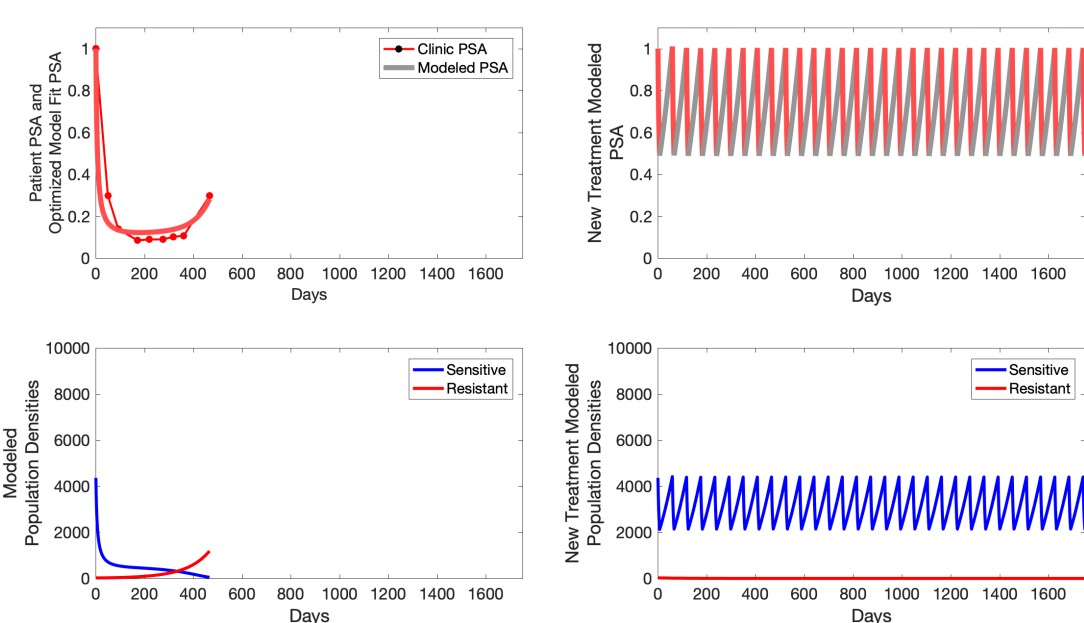

**Appendix 4—figure 8.** The left column shows actual outcome (upper panel) and computer simulations (lower panel) of the estimated population dynamics of the sensitive and resistant poulations for the indicated subject; the right column shows computer simulations based on an assumption that treatment was stopped immediately when the PSA fell below 0.5 of the pre-treatment and resumed immediately when PSA increased to the pretreatment value.

**C009**

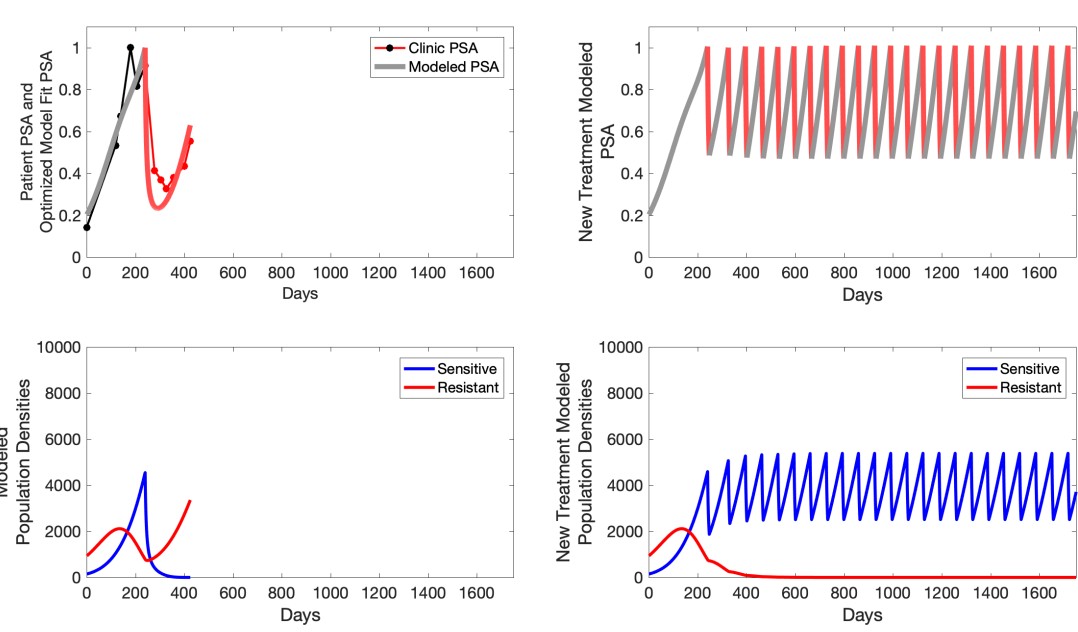

**Appendix 4—figure 9.** The left column shows actual outcome (upper panel) and computer simulations (lower panel) of the estimated population dynamics of the sensitive and resistant poulations for the indicated subject; the right column shows computer simulations based on an assumption that treatment was stopped immediately when the PSA fell below 0.5 of the pre-treatment and resumed immediately when PSA increased to the pretreatment value.

**C010**

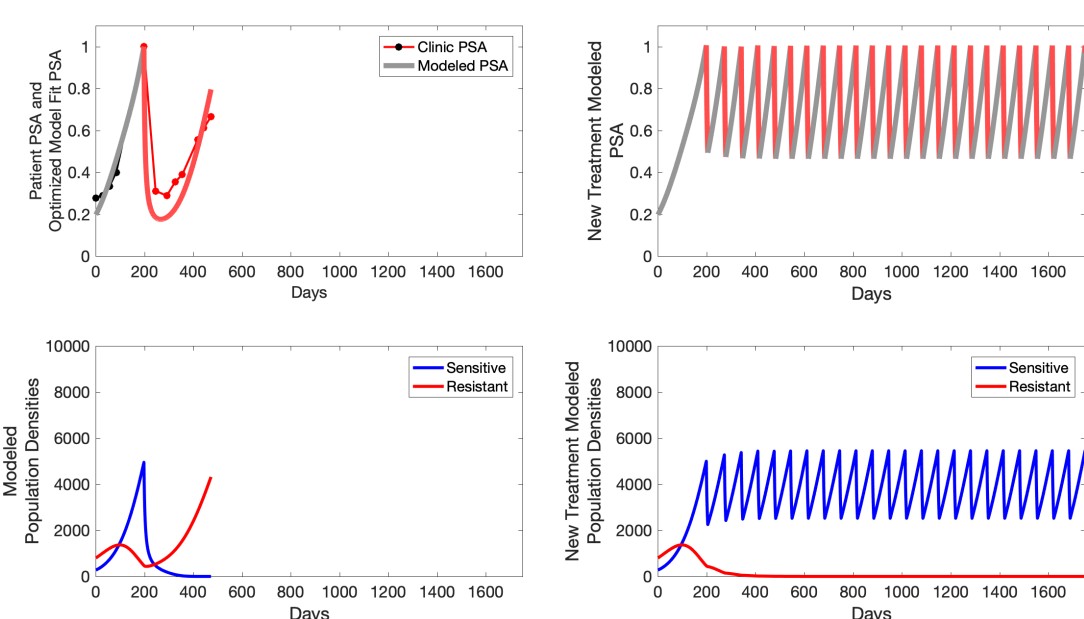

**Appendix 4—figure 10.** The left column shows actual outcome (upper panel) and computer simulations (lower panel) of the estimated population dynamics of the sensitive and resistant poulations for the indicated subject; the right column shows computer simulations based on an assumption that treatment was stopped immediately when the PSA fell below 0.5 of the pre-treatment and resumed immediately when PSA increased to the pretreatment value.

**C011**

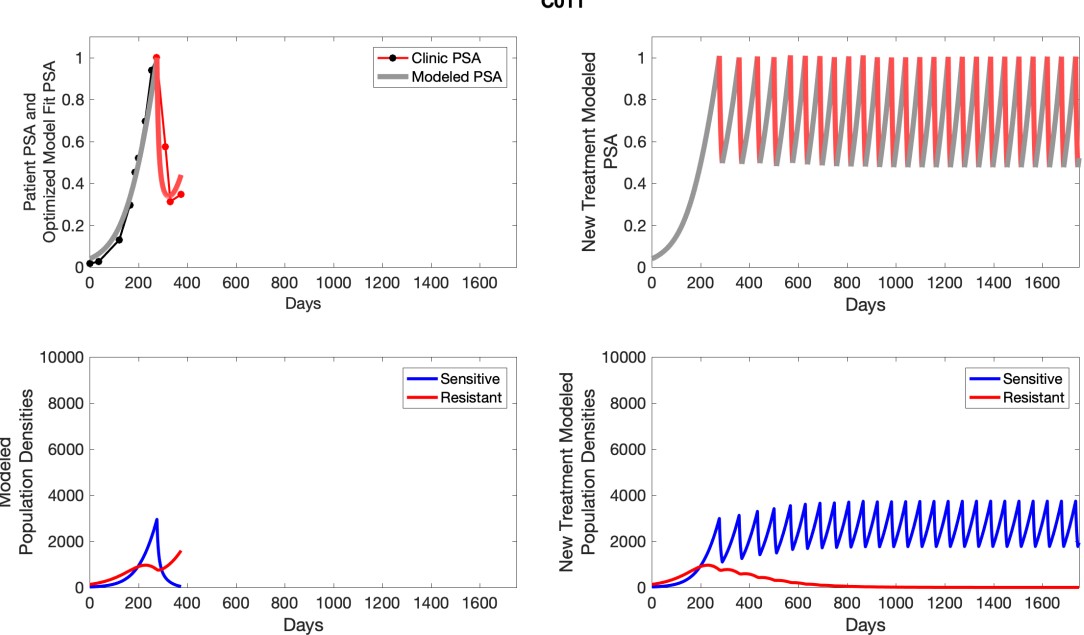

**Appendix 4—figure 11.** The left column shows actual outcome (upper panel) and computer simulations (lower panel) of the estimated population dynamics of the sensitive and resistant poulations for the indicated subject; the right column shows computer simulations based on an assumption that treatment was stopped immediately when the PSA fell below 0.5 of the pre-treatment and resumed immediately when PSA increased to the pretreatment value.

**C012**

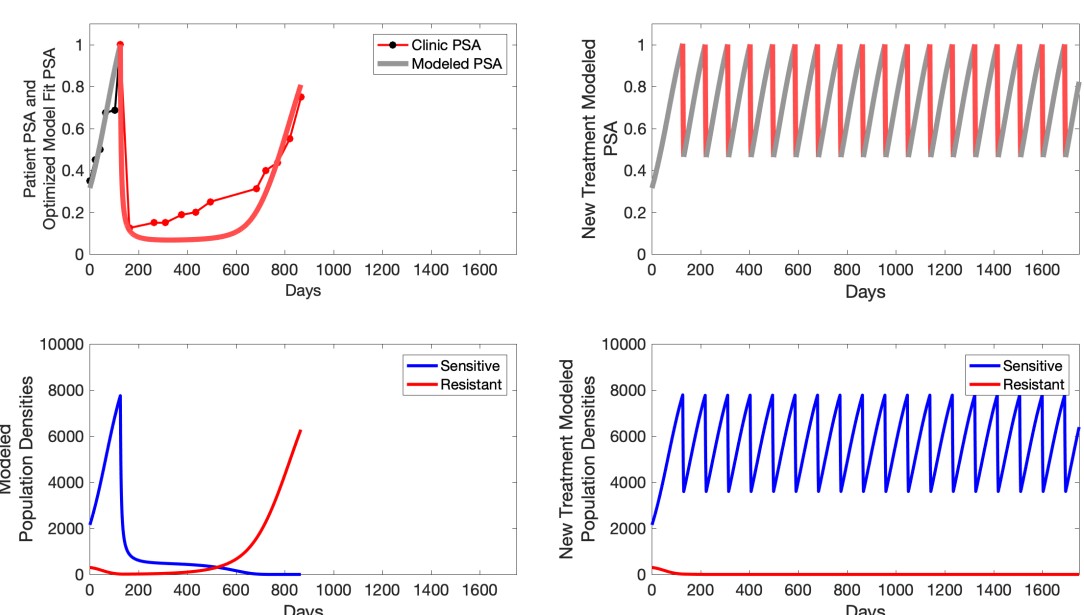

**Appendix 4—figure 12.** The left column shows actual outcome (upper panel) and computer simulations (lower panel) of the estimated population dynamics of the sensitive and resistant poulations for the indicated subject; the right column shows computer simulations based on an assumption that treatment was stopped immediately when the PSA fell below 0.5 of the pre-treatment and resumed immediately when PSA increased to the pretreatment value.

**C013**

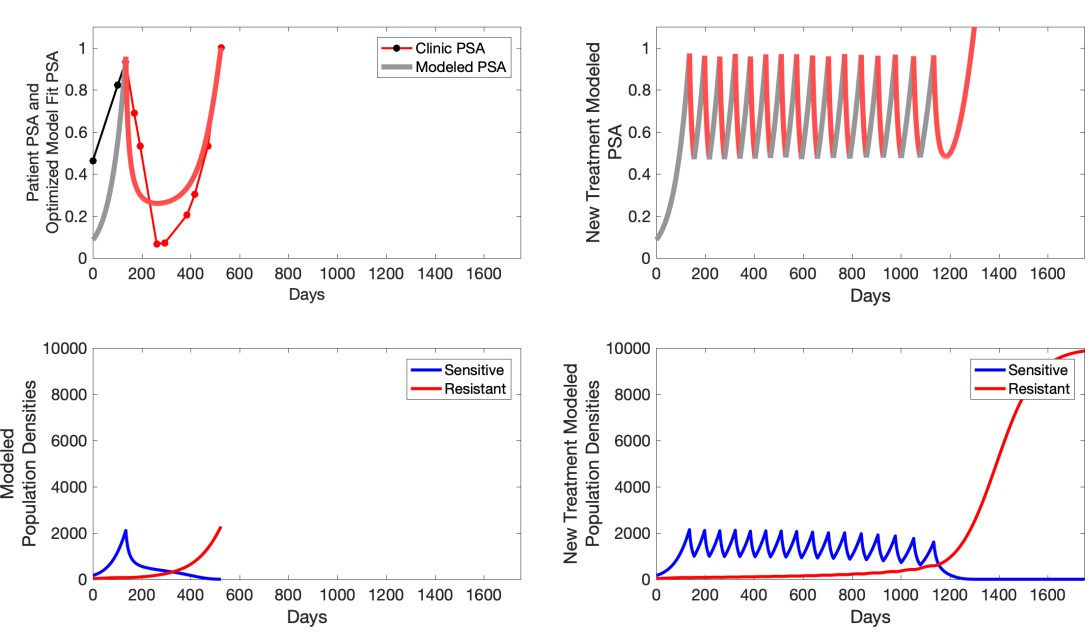

**Appendix 4—figure 13.** The left column shows actual outcome (upper panel) and computer simulations (lower panel) of the estimated population dynamics of the sensitive and resistant poulations for the indicated subject; the right column shows computer simulations based on an assumption that treatment was stopped immediately when the PSA fell below 0.5 of the pre-treatment and resumed immediately when PSA increased to the pretreatment value.

**C014**

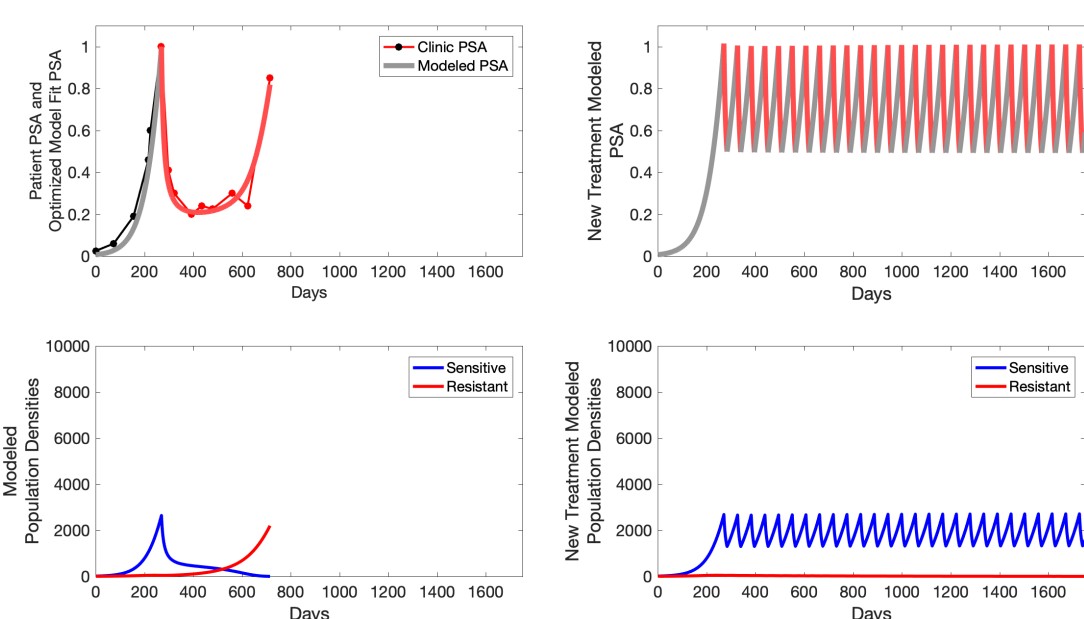

**Appendix 4—figure 14.** The left column shows actual outcome (upper panel) and computer simulations (lower panel) of the estimated population dynamics of the sensitive and resistant poulations for the indicated subject; the right column shows computer simulations based on an assumption that treatment was stopped immediately when the PSA fell below 0.5 of the pre-treatment and resumed immediately when PSA increased to the pretreatment value.

**C015**

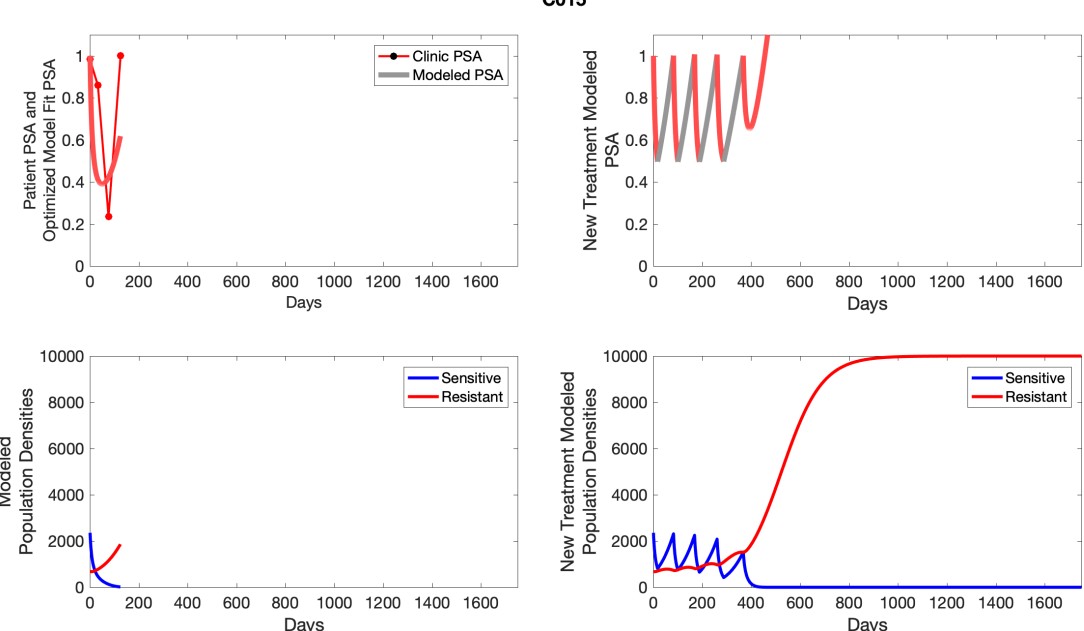

**Appendix 4—figure 15.** The left column shows actual outcome (upper panel) and computer simulations (lower panel) of the estimated population dynamics of the sensitive and resistant poulations for the indicated subject; the right column shows computer simulations based on an assumption that treatment was stopped immediately when the PSA fell below 0.5 of the pre-treatment and resumed immediately when PSA increased to the pretreatment value.

P1001

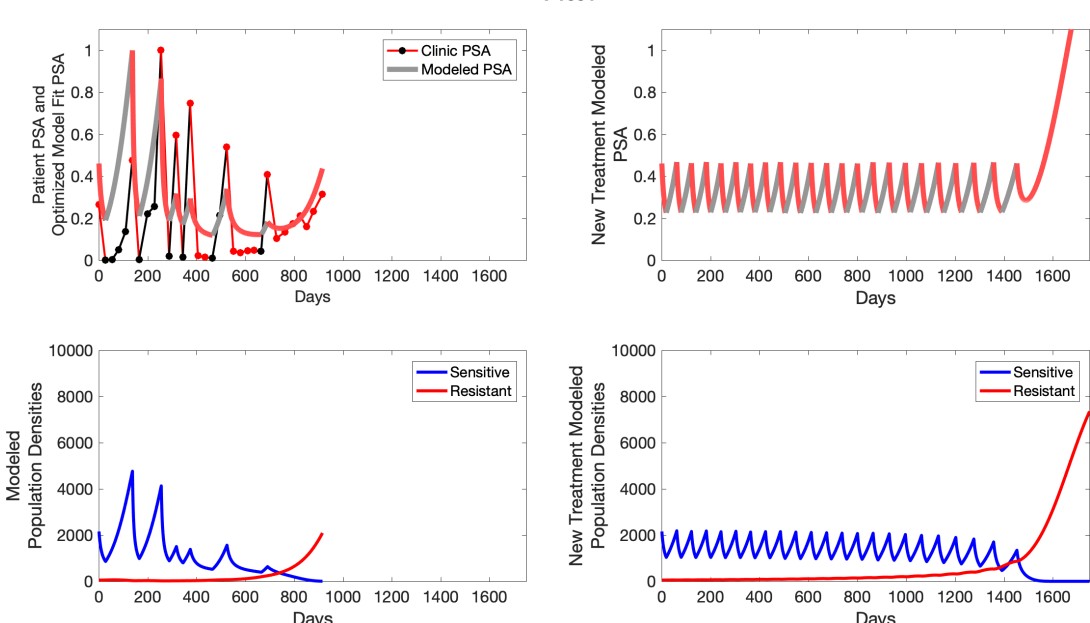

**Appendix 4—figure 16.** The left column shows actual outcome (upper panel) and computer simulations (lower panel) of the estimated population dynamics of the sensitive and resistant poulations for the indicated subject; the right column shows computer simulations based on an assumption that treatment was stopped immediately when the PSA fell below 0.5 of the pre-treatment and resumed immediately when PSA increased to the pretreatment value.

P1002

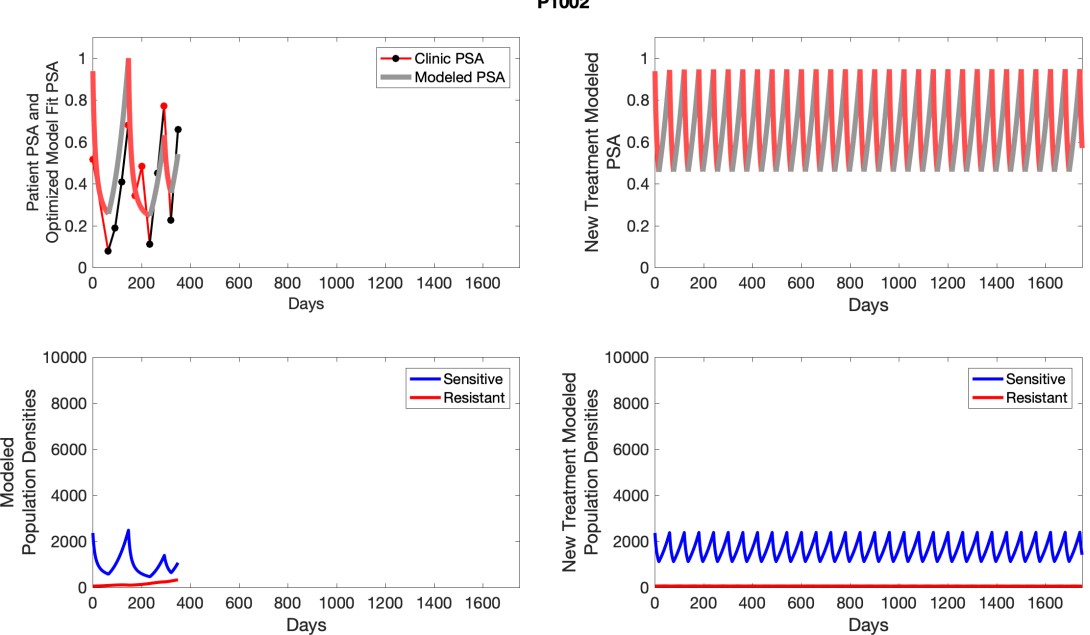

**Appendix 4—figure 17.** The left column shows actual outcome (upper panel) and computer simulations (lower panel) of the estimated population dynamics of the sensitive and resistant poulations for the indicated subject; the right column shows computer simulations based on an assumption that treatment was stopped immediately when the PSA fell below 0.5 of the pre-treatment and resumed immediately when PSA increased to the pretreatment value.

P1003

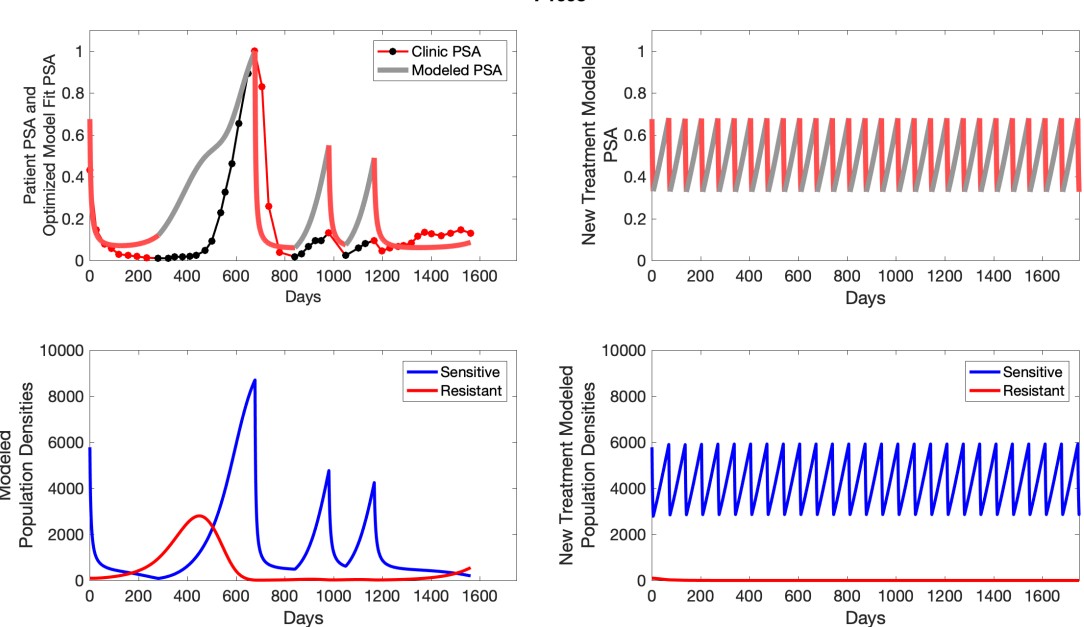

**Appendix 4—figure 18.** The left column shows actual outcome (upper panel) and computer simulations (lower panel) of the estimated population dynamics of the sensitive and resistant poulations for the indicated subject; the right column shows computer simulations based on an assumption that treatment was stopped immediately when the PSA fell below 0.5 of the pre-treatment and resumed immediately when PSA increased to the pretreatment value.

P1004

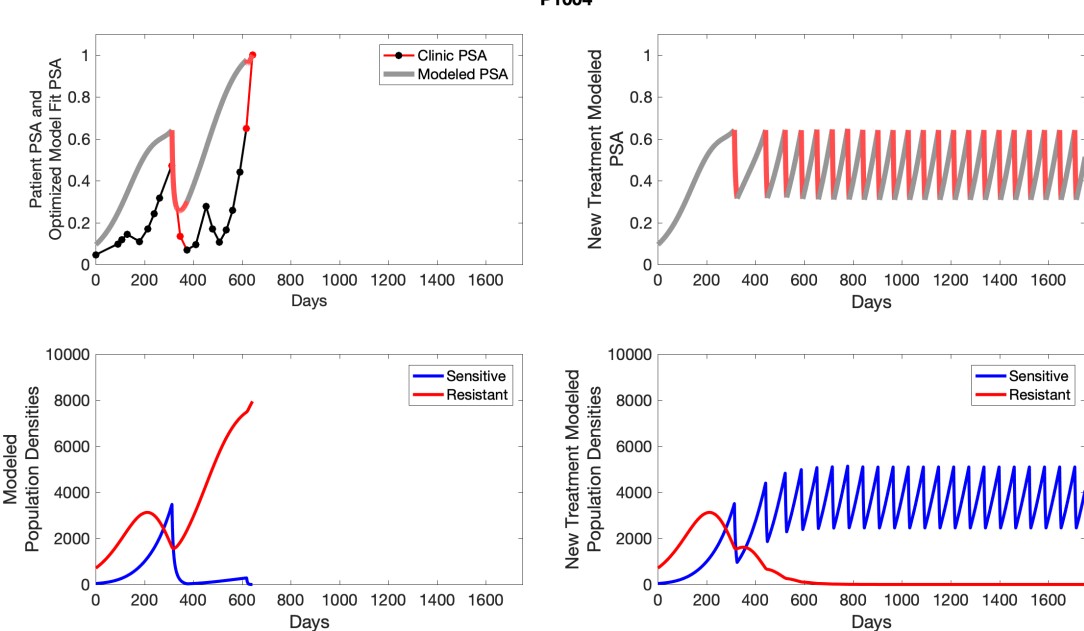

**Appendix 4—figure 19.** The left column shows actual outcome (upper panel) and computer simulations (lower panel) of the estimated population dynamics of the sensitive and resistant poulations for the indicated subject; the right column shows computer simulations based on an assumption that treatment was stopped immediately when the PSA fell below 0.5 of the pre-treatment and resumed immediately when PSA increased to the pretreatment value.

P1005

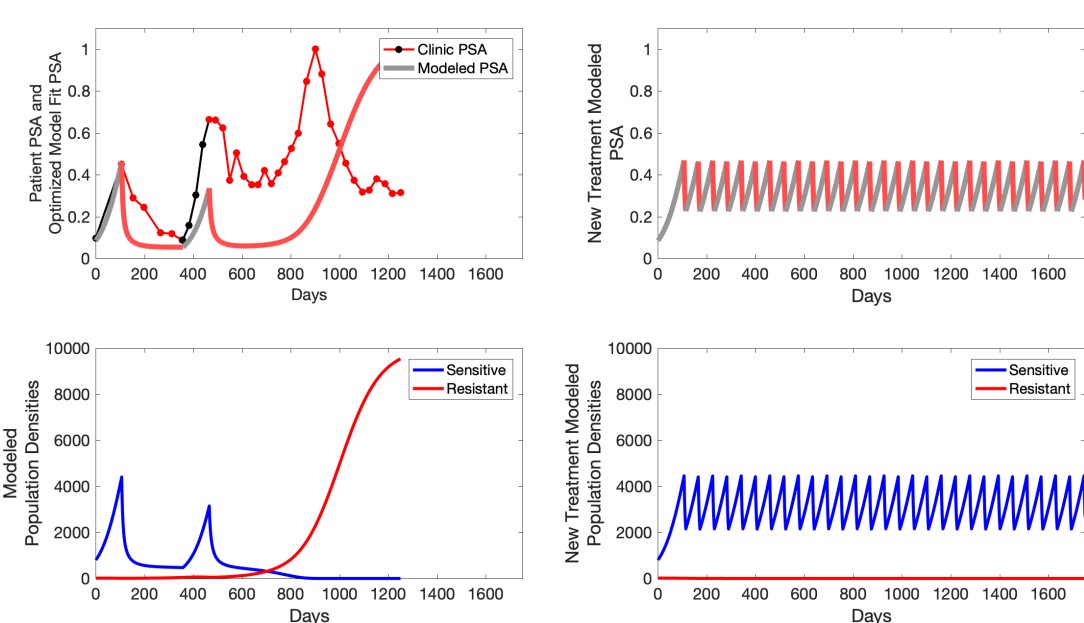

**Appendix 4—figure 20.** The left column shows actual outcome (upper panel) and computer simulations (lower panel) of the estimated population dynamics of the sensitive and resistant poulations for the indicated subject; the right column shows computer simulations based on an assumption that treatment was stopped immediately when the PSA fell below 0.5 of the pre-treatment and resumed immediately when PSA increased to the pretreatment value.

P1006

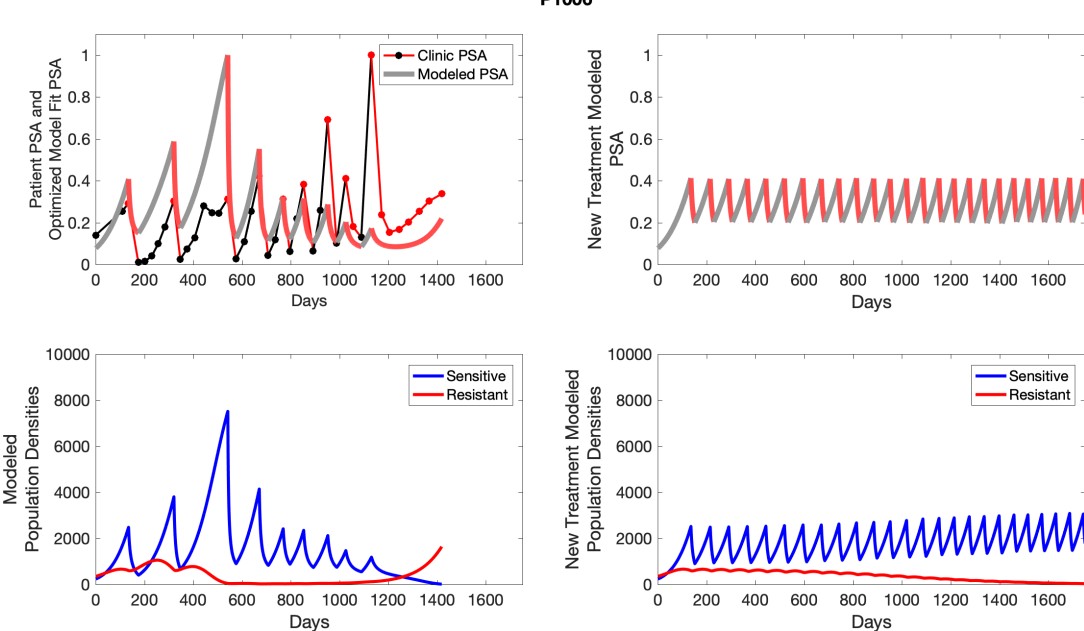

**Appendix 4—figure 21.** The left column shows actual outcome (upper panel) and computer simulations (lower panel) of the estimated population dynamics of the sensitive and resistant poulations for the indicated subject; the right column shows computer simulations based on an assumption that treatment was stopped immediately when the PSA fell below 0.5 of the pre-treatment and resumed immediately when PSA increased to the pretreatment value.

P1007

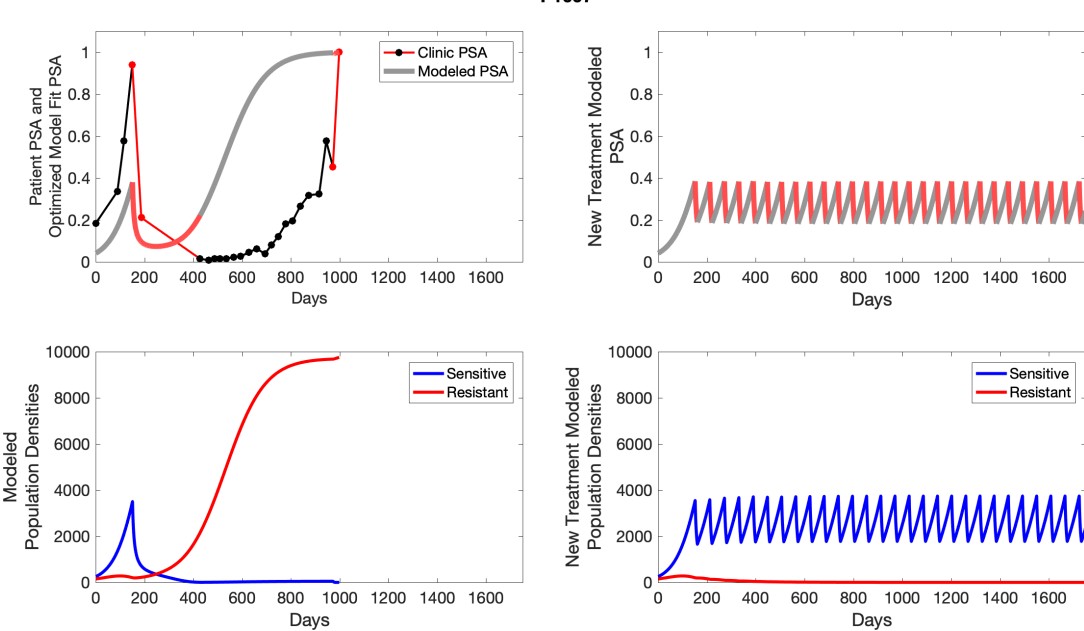

**Appendix 4—figure 22.** The left column shows actual outcome (upper panel) and computer simulations (lower panel) of the estimated population dynamics of the sensitive and resistant poulations for the indicated subject; the right column shows computer simulations based on an assumption that treatment was stopped immediately when the PSA fell below 0.5 of the pre-treatment and resumed immediately when PSA increased to the pretreatment value.

P1009

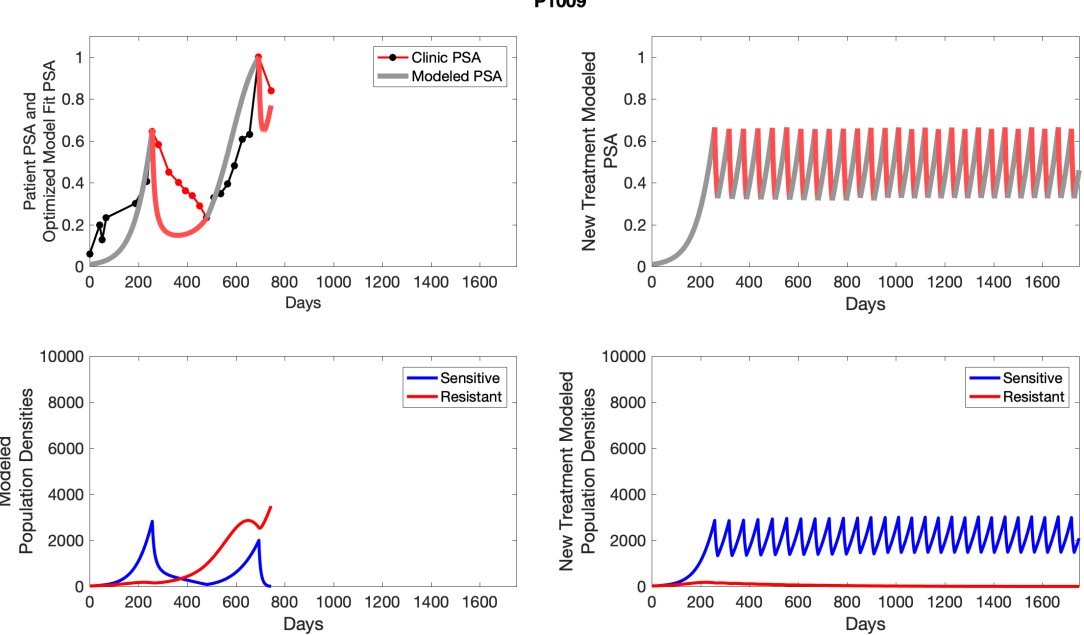

**Appendix 4—figure 23.** The left column shows actual outcome (upper panel) and computer simulations (lower panel) of the estimated population dynamics of the sensitive and resistant poulations for the indicated subject; the right column shows computer simulations based on an assumption that treatment was stopped immediately when the PSA fell below 0.5 of the pre-treatment and resumed immediately when PSA increased to the pretreatment value.

P1010

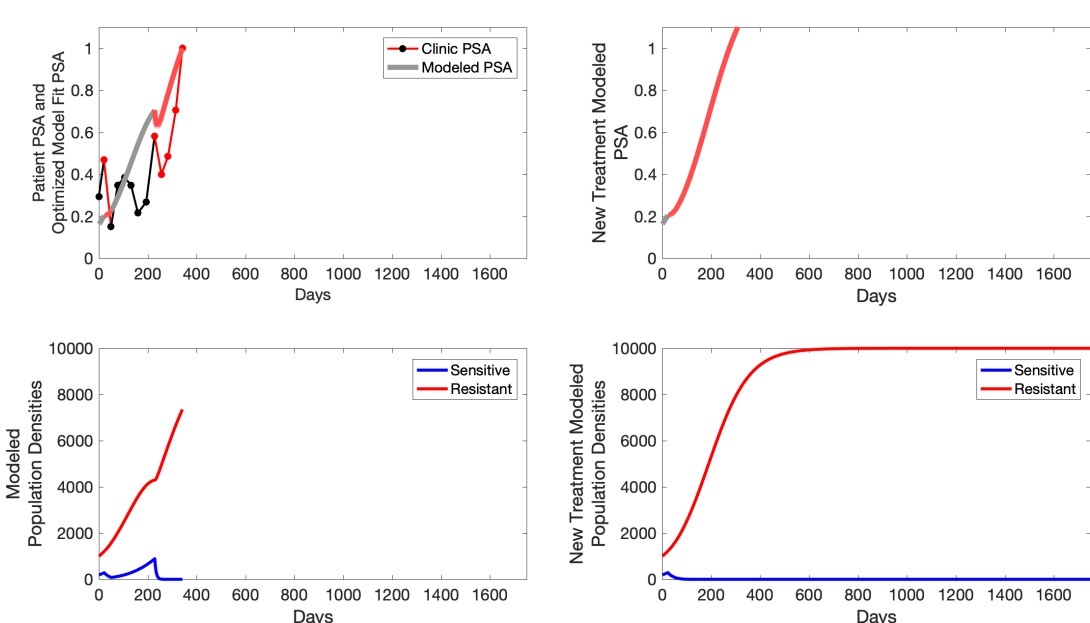

**Appendix 4—figure 24.** The left column shows actual outcome (upper panel) and computer simulations (lower panel) of the estimated population dynamics of the sensitive and resistant poulations for the indicated subject; the right column shows computer simulations based on an assumption that treatment was stopped immediately when the PSA fell below 0.5 of the pre-treatment and resumed immediately when PSA increased to the pretreatment value.

P1011

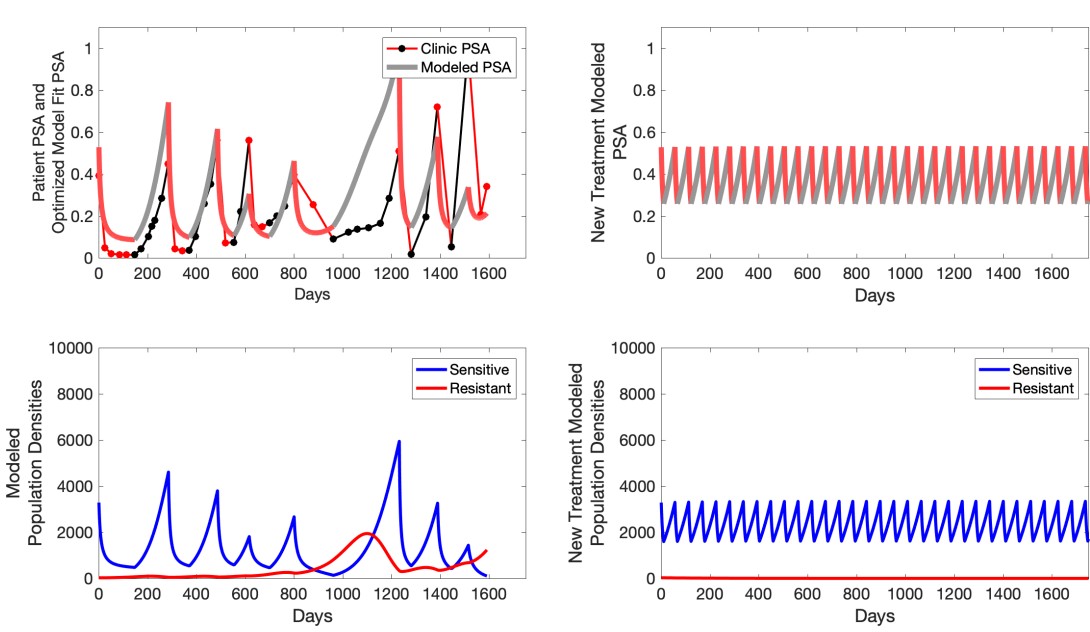

**Appendix 4—figure 25.** The left column shows actual outcome (upper panel) and computer simulations (lower panel) of the estimated population dynamics of the sensitive and resistant poulations for the indicated subject; the right column shows computer simulations based on an assumption that treatment was stopped immediately when the PSA fell below 0.5 of the pre-treatment and resumed immediately when PSA increased to the pretreatment value.

P1012

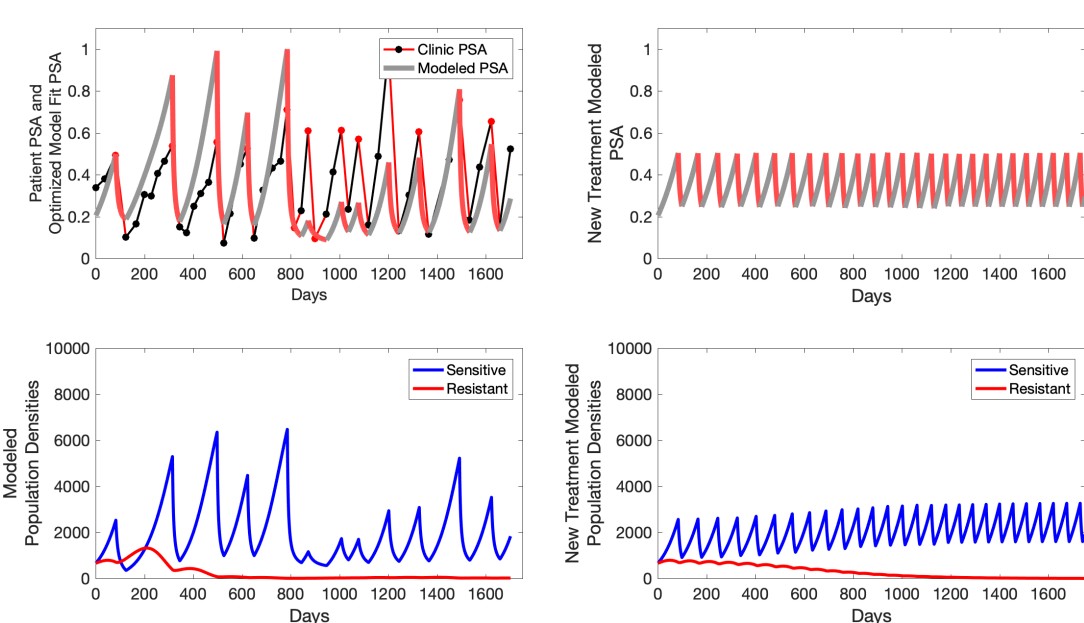

**Appendix 4—figure 26.** The left column shows actual outcome (upper panel) and computer simulations (lower panel) of the estimated population dynamics of the sensitive and resistant poulations for the indicated subject; the right column shows computer simulations based on an assumption that treatment was stopped immediately when the PSA fell below 0.5 of the pre-treatment and resumed immediately when PSA increased to the pretreatment value.

P1014

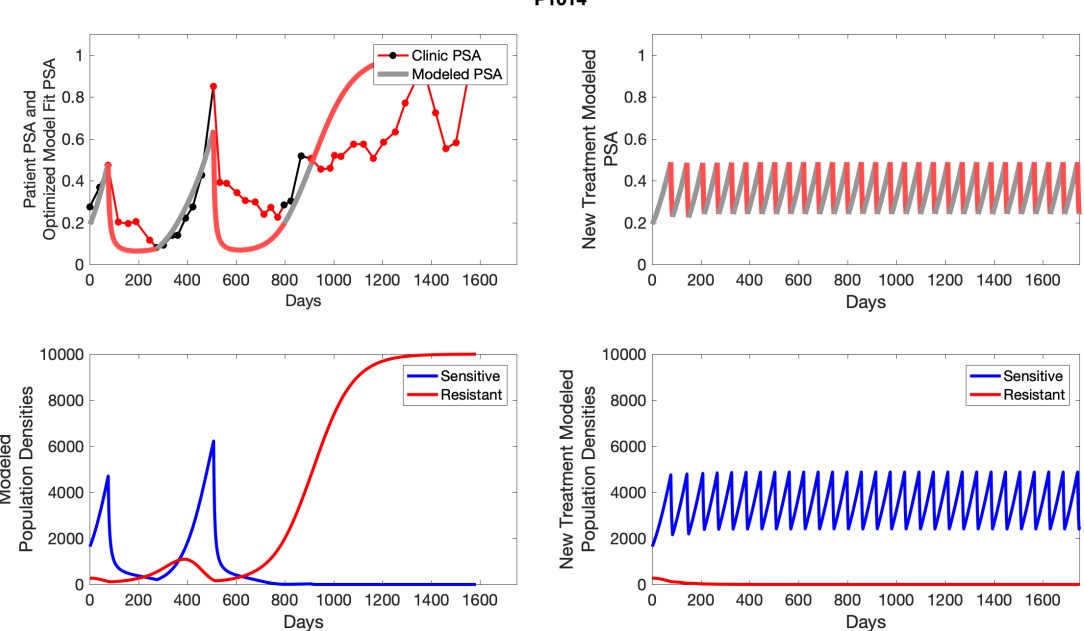

**Appendix 4—figure 27.** The left column shows actual outcome (upper panel) and computer simulations (lower panel) of the estimated population dynamics of the sensitive and resistant poulations for the indicated subject; the right column shows computer simulations based on an assumption that treatment was stopped immediately when the PSA fell below 0.5 of the pre-treatment and resumed immediately when PSA increased to the pretreatment value.

**P1015**

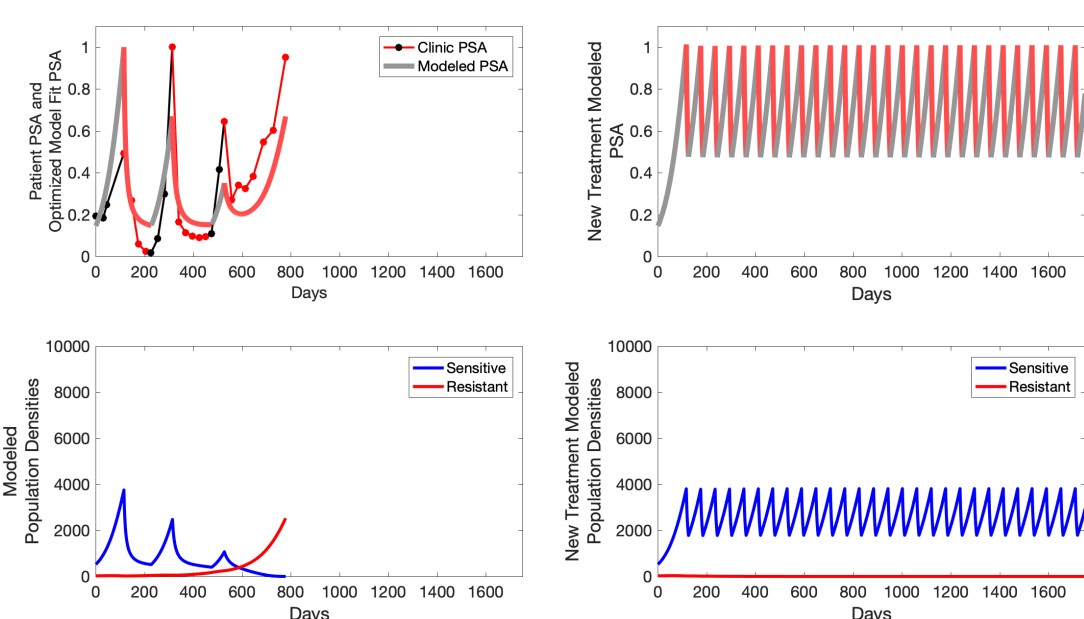

**Appendix 4—figure 28.** The left column shows actual outcome (upper panel) and computer simulations (lower panel) of the estimated population dynamics of the sensitive and resistant poulations for the indicated subject; the right column shows computer simulations based on an assumption that treatment was stopped immediately when the PSA fell below 0.5 of the pre-treatment and resumed immediately when PSA increased to the pretreatment value.

**P1016**

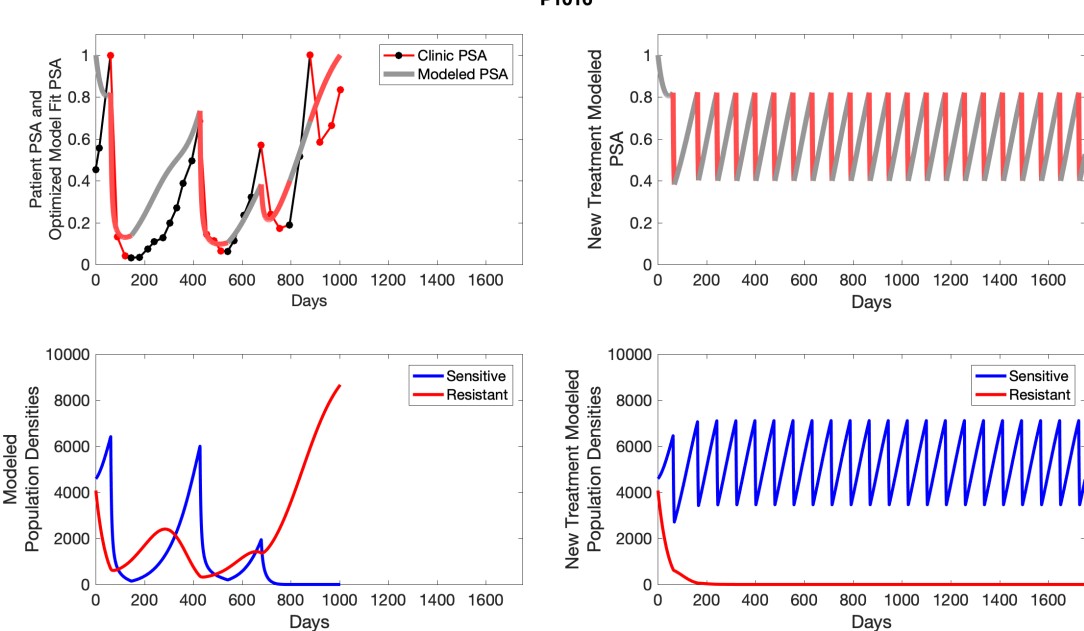

**Appendix 4—figure 29.** The left column shows actual outcome (upper panel) and computer simulations (lower panel) of the estimated population dynamics of the sensitive and resistant poulations for the indicated subject; the right column shows computer simulations based on an assumption that treatment was stopped immediately when the PSA fell below 0.5 of the pre-treatment and resumed immediately when PSA increased to the pretreatment value.

**P1017**

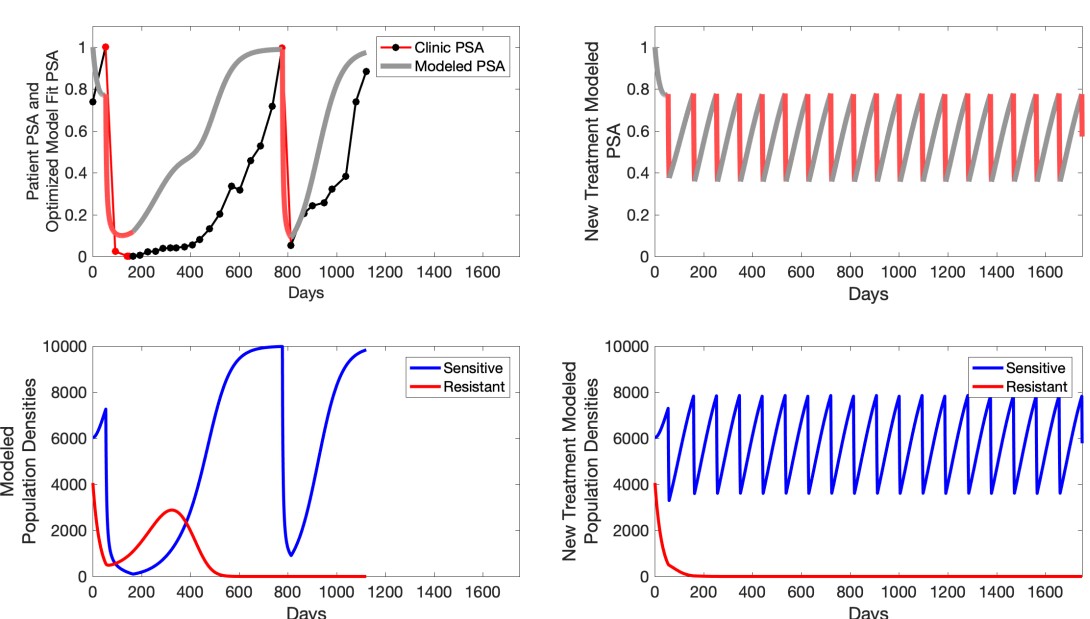

**Appendix 4—figure 30.** The left column shows actual outcome (upper panel) and computer simulations (lower panel) of the estimated population dynamics of the sensitive and resistant poulations for the indicated subject; the right column shows computer simulations based on an assumption that treatment was stopped immediately when the PSA fell below 0.5 of the pre-treatment and resumed immediately when PSA increased to the pretreatment value.

**P1018**

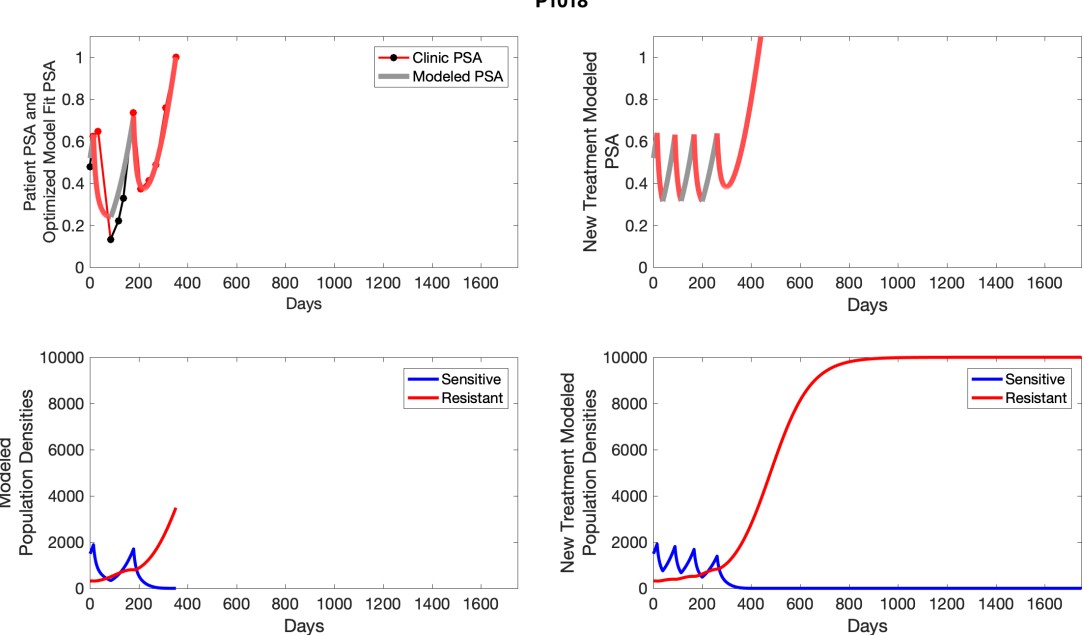

**Appendix 4—figure 31.** The left column shows actual outcome (upper panel) and computer simulations (lower panel) of the estimated population dynamics of the sensitive and resistant poulations for the indicated subject; the right column shows computer simulations based on an assumption that treatment was stopped immediately when the PSA fell below 0.5 of the pre-treatment and resumed immediately when PSA increased to the pretreatment value.

P1020

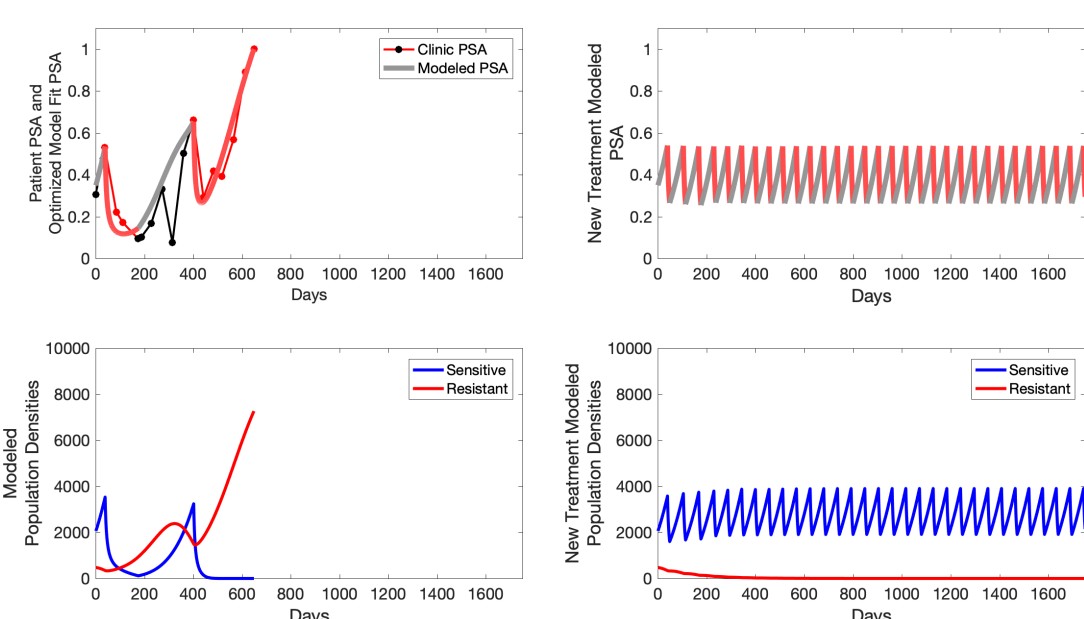

**Appendix 4—figure 32.** The left column shows actual outcome (upper panel) and computer simulations (lower panel) of the estimated population dynamics of the sensitive and resistant poulations for the indicated subject; the right column shows computer simulations based on an assumption that treatment was stopped immediately when the PSA fell below 0.5 of the pre-treatment and resumed immediately when PSA increased to the pretreatment value.

## Appendix 5

## Simulated standard of care

To evaluate SOC protocol, we ran a simulation for each patient (both cohorts). For each patient, we initiated model runs using the patient-specific parameters from *Appendix 2—table 3*. We considered continuous abiraterone therapy for both cohorts. SOC was initiated at the time abiraterone was first administered clinically to each patient, which was not always at t = 0. For the contemporaneous cohort that received SOC clinically, we ran the dynamics of SOC beyond clinical measurements to predict how the disease would have progressed in the absence of any other treatments. For the adaptive therapy patients, we could predict how each patient would have fared under SOC abiraterone.

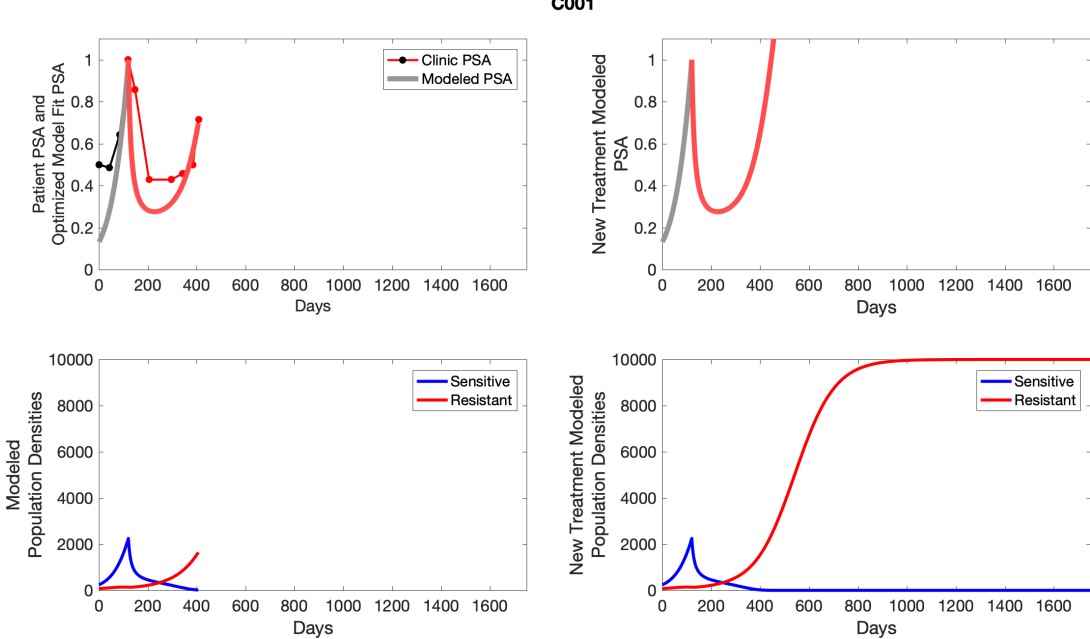

**Appendix 5—figure 1.** The left column shows actual PSA values (upper right panel) with the PSA dynamics of the fitted mathematical model for the indicated subject, the lower left panel shows computer simulations of the estimated population dynamics for sensitive and resistant cells; the right panels show computer simulations for extended therapy using the standard of care dosing with continuous MTD abiraterone.

**C002**

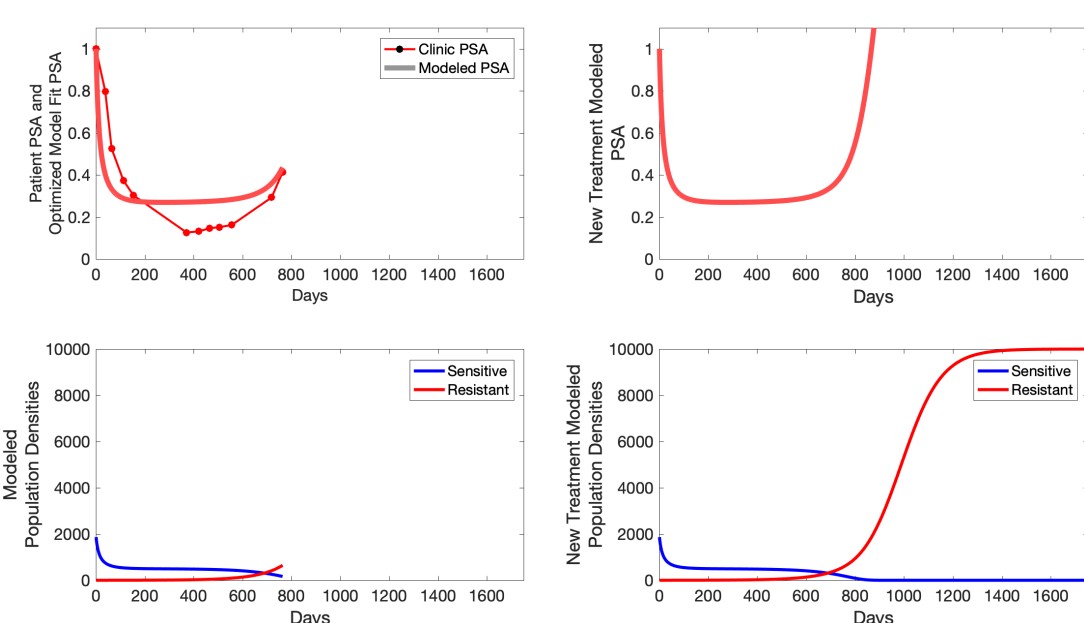

**Appendix 5—figure 2.** The left column shows actual PSA values (upper right panel) with the PSA dynamics of the fitted mathematical model for the indicated subject, the lower left panel shows computer simulations of the estimated population dynamics for sennsitive and resistant cells; the right panels show computer simulations for extended therapy using the standard of care dosing with continuous MTD abiraterone.

**C003**

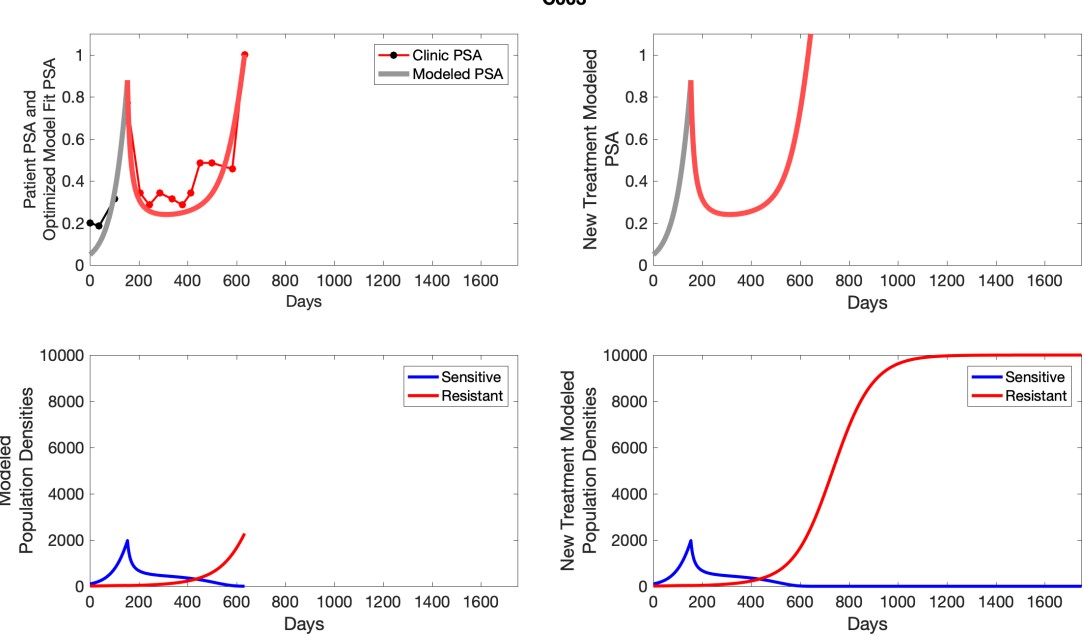

**Appendix 5—figure 3.** The left column shows actual PSA values (upper right panel) with the PSA dynamics of the fitted mathematical model for the indicated subject, the lower left panel shows computer simulations of the estimated population dynamics for sennsitive and resistant cells; the right panels show computer simulations for extended therapy using the standard of care dosing with continuous MTD abiraterone.

**C004**

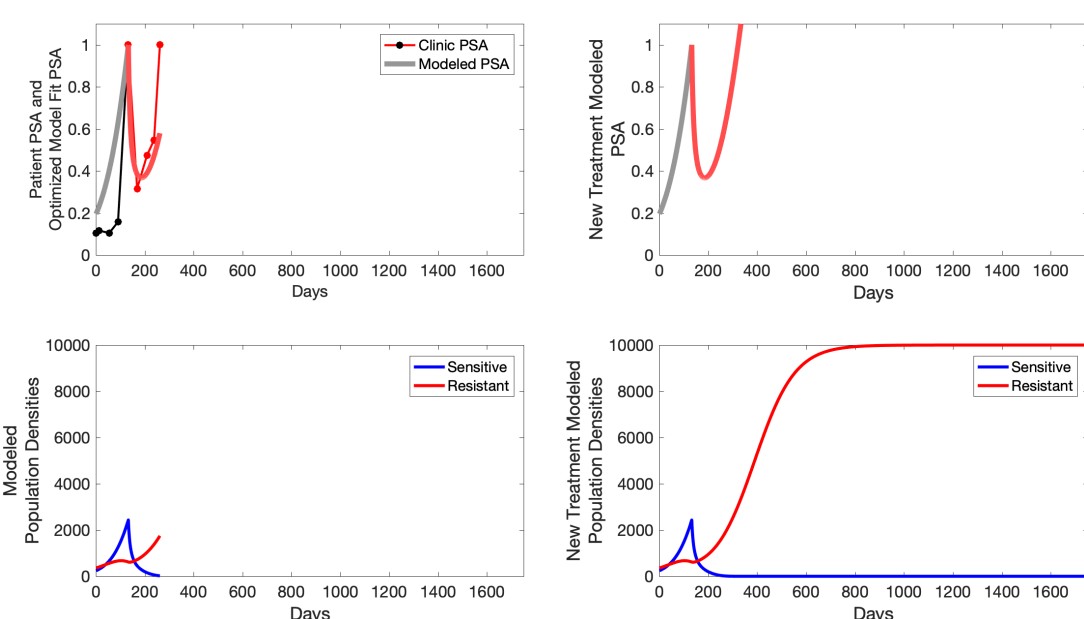

**Appendix 5—figure 4.** The left column shows actual PSA values (upper right panel) with the PSA dynamics of the fitted mathematical model for the indicated subject, the lower left panel shows computer simulations of the estimated population dynamics for sennsitive and resistant cells; the right panels show computer simulations for extended therapy using the standard of care dosing with continuous MTD abiraterone.

**C005**

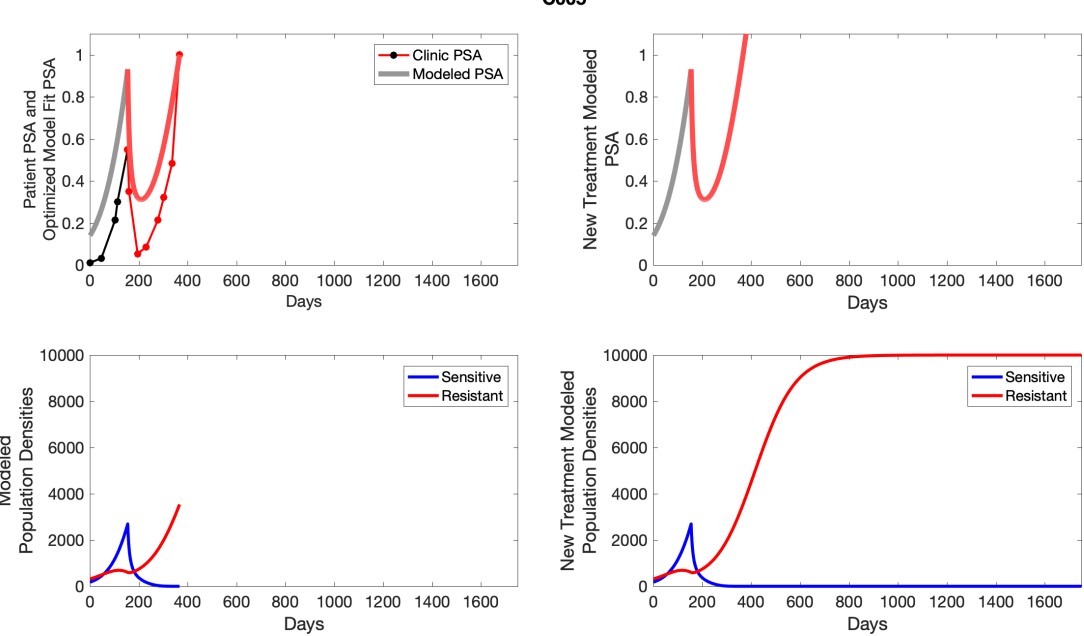

**Appendix 5—figure 5.** The left column shows actual PSA values (upper right panel) with the PSA dynamics of the fitted mathematical model for the indicated subject, the lower left panel shows computer simulations of the estimated population dynamics for sennsitive and resistant cells; the right panels show computer simulations for extended therapy using the standard of care dosing with continuous MTD abiraterone.

C006

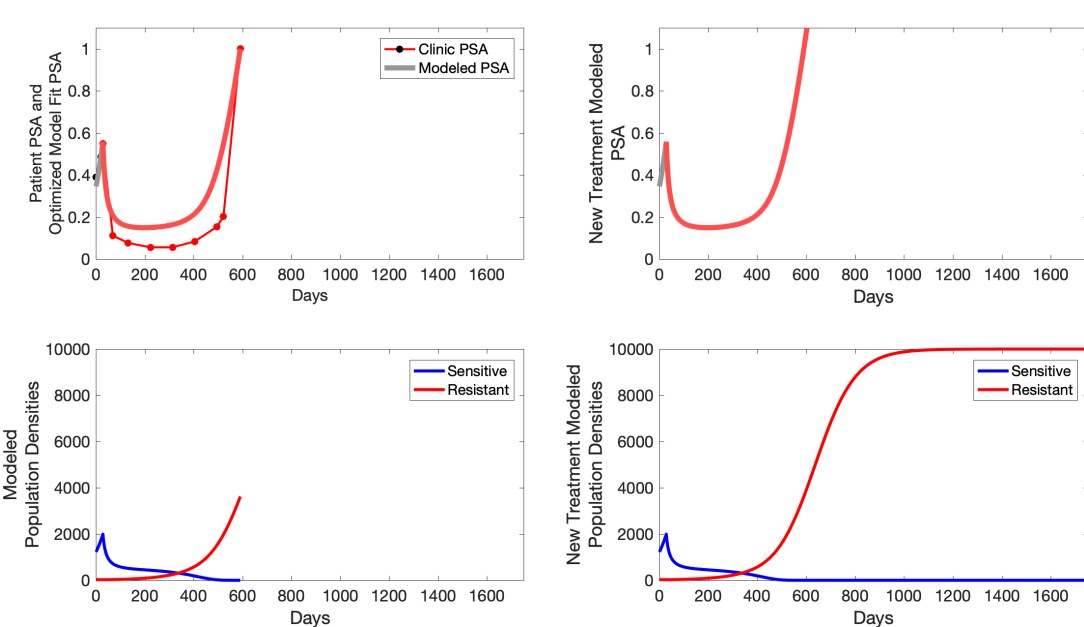

**Appendix 5—figure 6.** The left column shows actual PSA values (upper right panel) with the PSA dynamics of the fitted mathematical model for the indicated subject, the lower left panel shows computer simulations of the estimated population dynamics for sennsitive and resistant cells; the right panels show computer simulations for extended therapy using the standard of care dosing with continuous MTD abiraterone.

C007

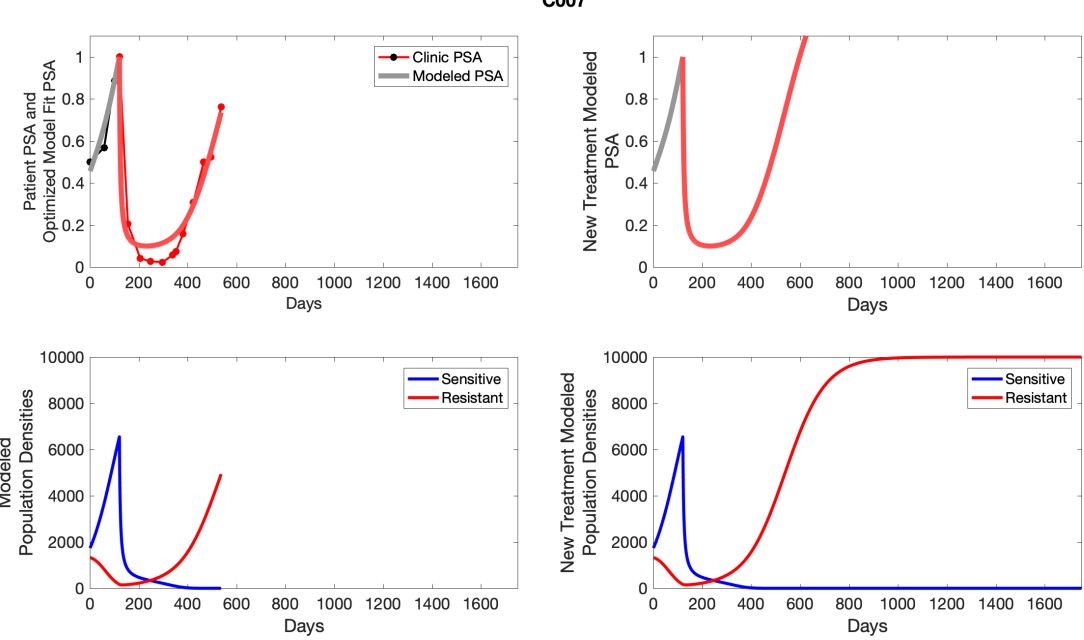

**Appendix 5—figure 7.** The left column shows actual PSA values (upper right panel) with the PSA dynamics of the fitted mathematical model for the indicated subject, the lower left panel shows computer simulations of the estimated population dynamics for sennsitive and resistant cells; the right panels show computer simulations for extended therapy using the standard of care dosing with continuous MTD abiraterone.

**C008**

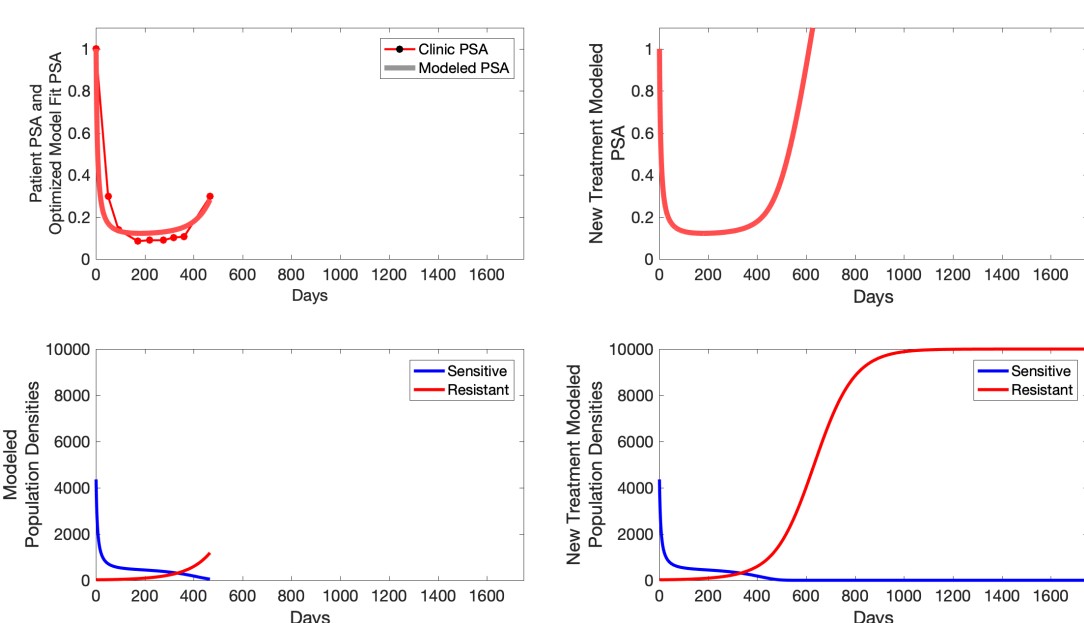

**Appendix 5—figure 8.** The left column shows actual PSA values (upper right panel) with the PSA dynamics of the fitted mathematical model for the indicated subject, the lower left panel shows computer simulations of the estimated population dynamics for sennsitive and resistant cells; the right panels show computer simulations for extended therapy using the standard of care dosing with continuous MTD abiraterone.

**C009**

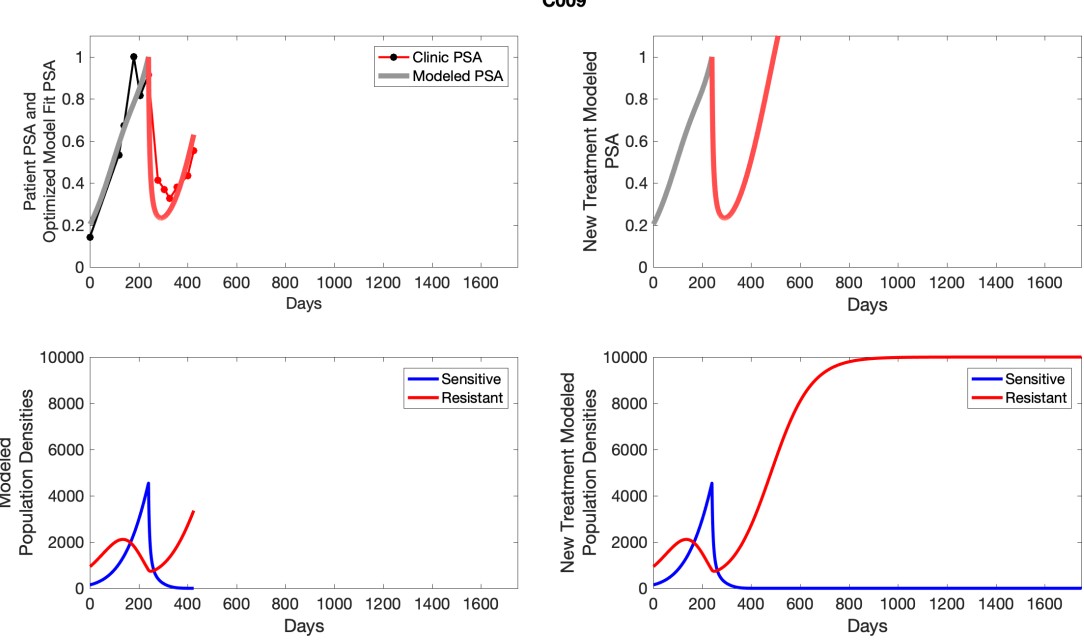

**Appendix 5—figure 9.** The left column shows actual PSA values (upper right panel) with the PSA dynamics of the fitted mathematical model for the indicated subject, the lower left panel shows computer simulations of the estimated population dynamics for sennsitive and resistant cells; the right panels show computer simulations for extended therapy using the standard of care dosing with continuous MTD abiraterone.

**C010**

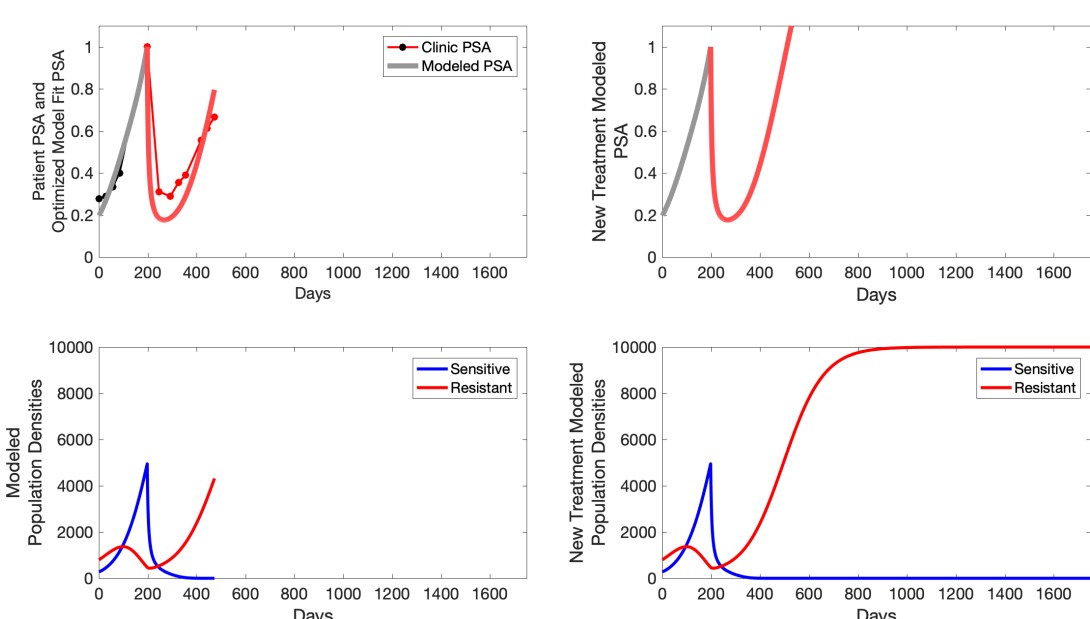

**Appendix 5—figure 10.** The left column shows actual PSA values (upper right panel) with the PSA dynamics of the fitted mathematical model for the indicated subject, the lower left panel shows computer simulations of the estimated population dynamics for sennsitive and resistant cells; the right panels show computer simulations for extended therapy using the standard of care dosing with continuous MTD abiraterone.

**C011**

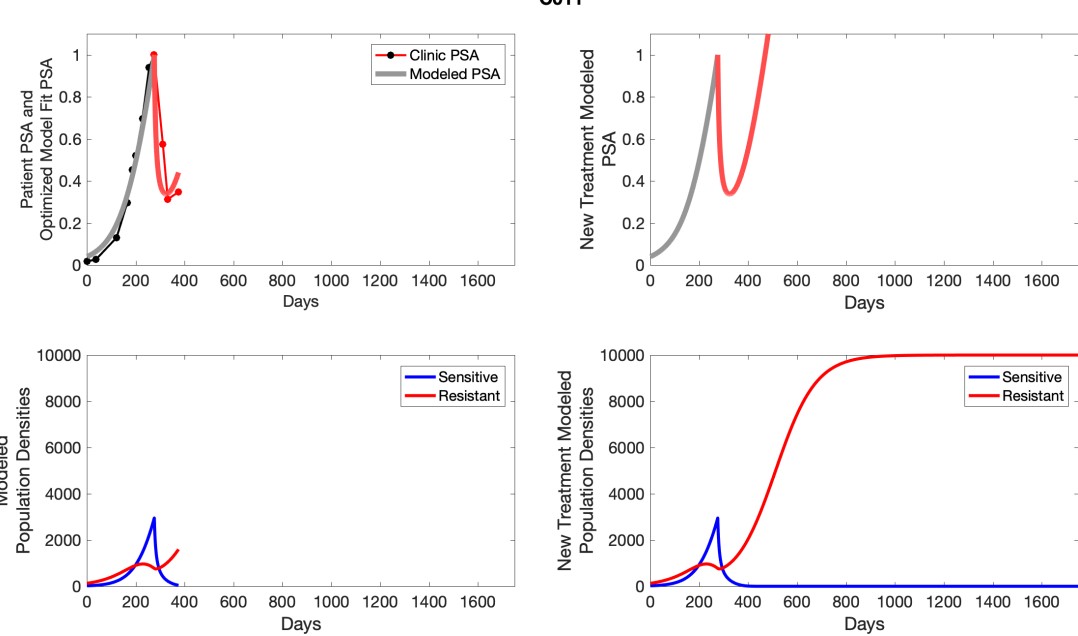

**Appendix 5—figure 11.** The left column shows actual PSA values (upper right panel) with the PSA dynamics of the fitted mathematical model for the indicated subject, the lower left panel shows computer simulations of the estimated population dynamics for sennsitive and resistant cells; the right panels show computer simulations for extended therapy using the standard of care dosing with continuous MTD abiraterone.

**C012**

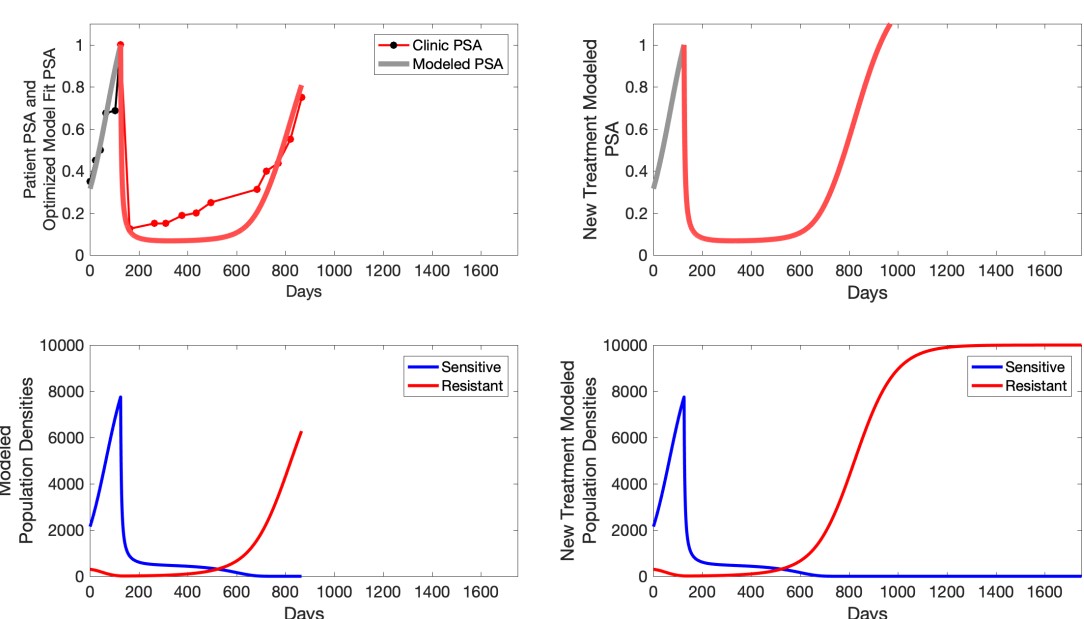

**Appendix 5—figure 12.** The left column shows actual PSA values (upper right panel) with the PSA dynamics of the fitted mathematical model for the indicated subject, the lower left panel shows computer simulations of the estimated population dynamics for sennsitive and resistant cells; the right panels show computer simulations for extended therapy using the standard of care dosing with continuous MTD abiraterone.

**C013**

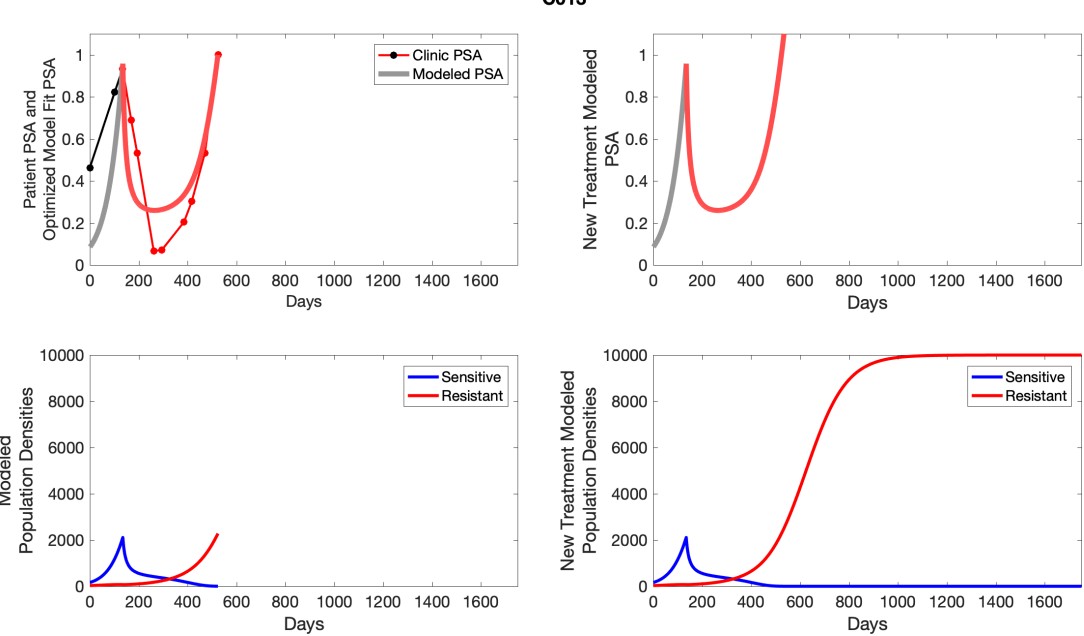

**Appendix 5—figure 13.** The left column shows actual PSA values (upper right panel) with the PSA dynamics of the fitted mathematical model for the indicated subject, the lower left panel shows computer simulations of the estimated population dynamics for sennsitive and resistant cells; the right panels show computer simulations for extended therapy using the standard of care dosing with continuous MTD abiraterone.

**C014**

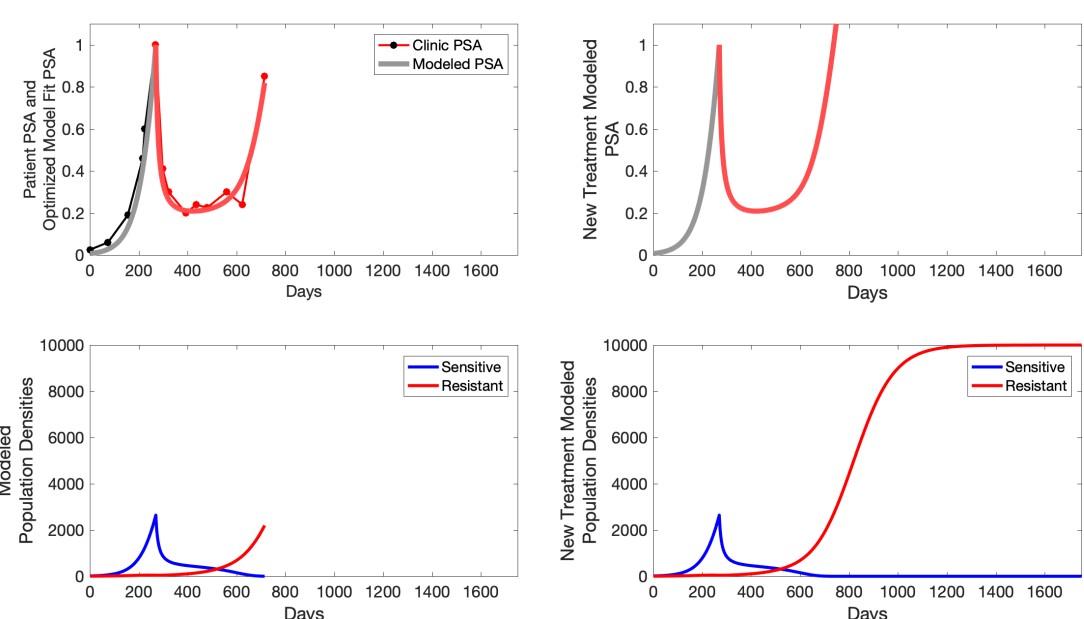

**Appendix 5—figure 14.** The left column shows actual PSA values (upper right panel) with the PSA dynamics of the fitted mathematical model for the indicated subject, the lower left panel shows computer simulations of the estimated population dynamics for sennsitive and resistant cells; the right panels show computer simulations for extended therapy using the standard of care dosing with continuous MTD abiraterone.

**C015**

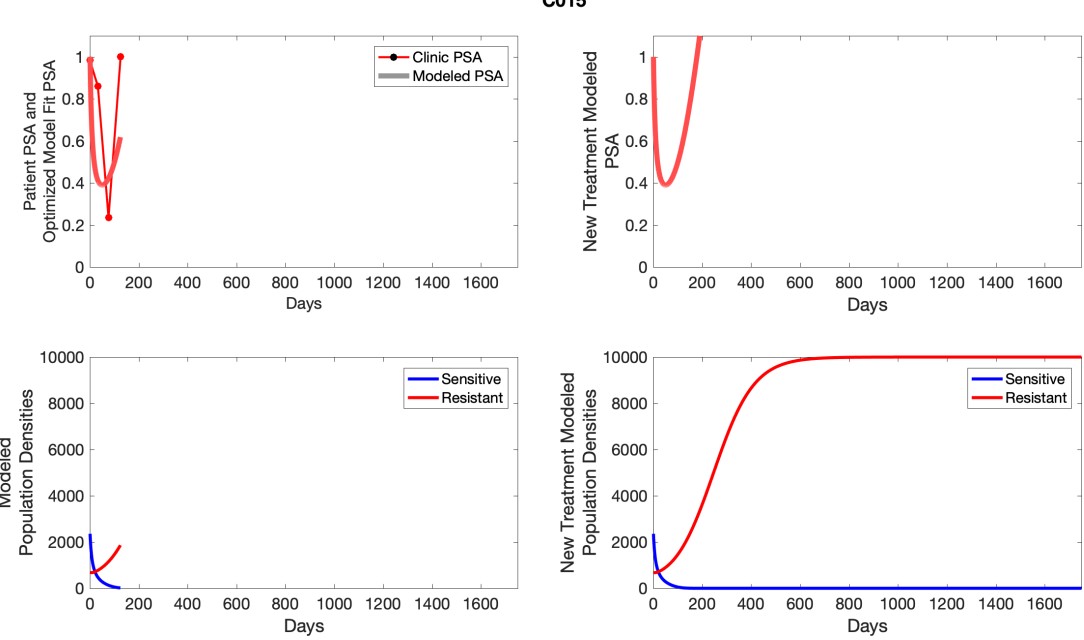

**Appendix 5—figure 15.** The left column shows actual PSA values (upper right panel) with the PSA dynamics of the fitted mathematical model for the indicated subject, the lower left panel shows computer simulations of the estimated population dynamics for sennsitive and resistant cells; the right panels show computer simulations for extended therapy using the standard of care dosing with continuous MTD abiraterone.

P1001

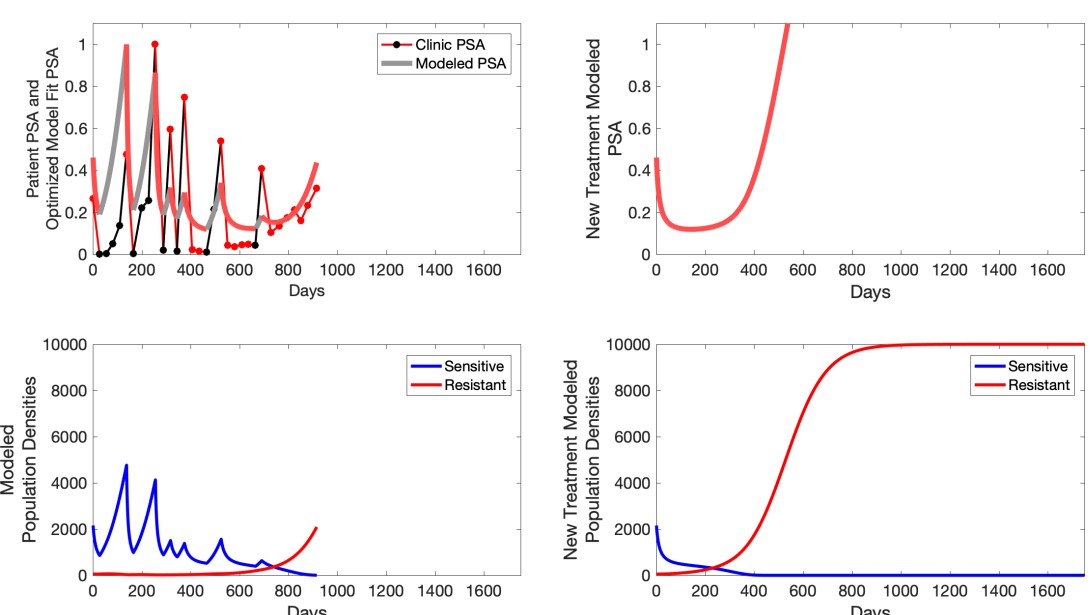

**Appendix 5—figure 16.** The left column shows actual PSA values (upper right panel) with the PSA dynamics of the fitted mathematical model for the indicated subject, the lower left panel shows computer simulations of the estimated population dynamics for sennsitive and resistant cells; the right panels show computer simulations for extended therapy using the standard of care dosing with continuous MTD abiraterone.

P1002

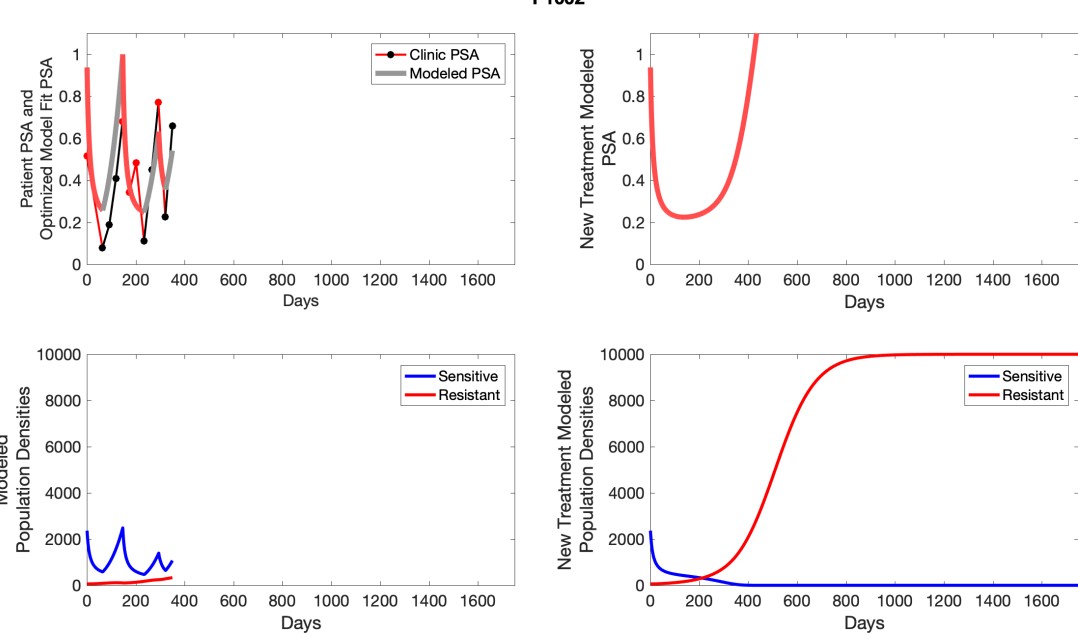

**Appendix 5—figure 17.** The left column shows actual PSA values (upper right panel) with the PSA dynamics of the fitted mathematical model for the indicated subject, the lower left panel shows computer simulations of the estimated population dynamics for sennsitive and resistant cells; the right panels show computer simulations for extended therapy using the standard of care dosing with continuous MTD abiraterone.

P1003

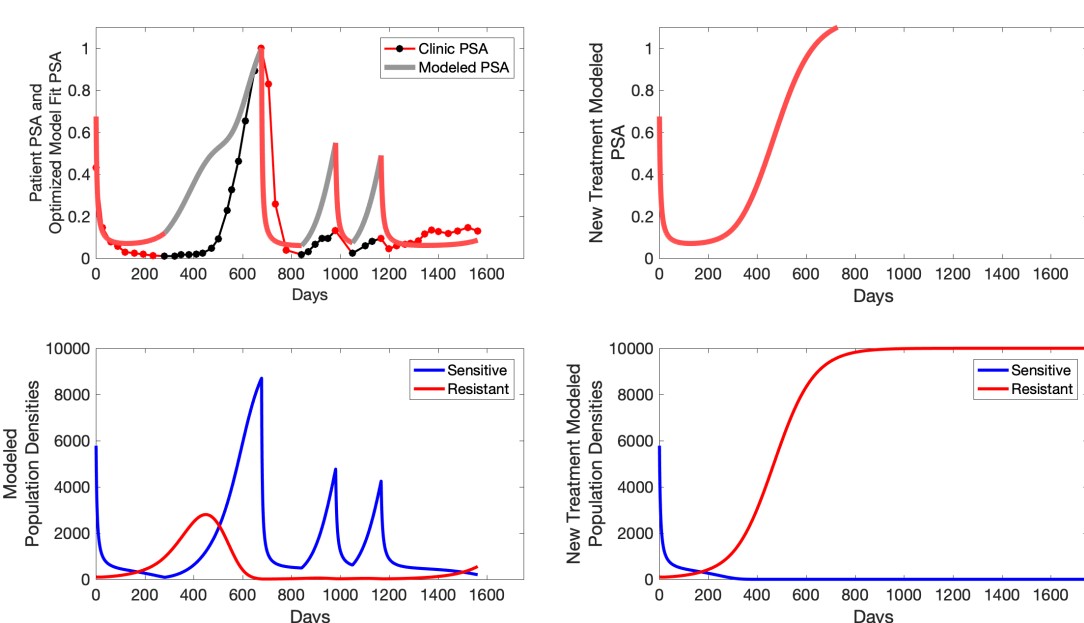

**Appendix 5—figure 18.** The left column shows actual PSA values (upper right panel) with the PSA dynamics of the fitted mathematical model for the indicated subject, the lower left panel shows computer simulations of the estimated population dynamics for sennsitive and resistant cells; the right panels show computer simulations for extended therapy using the standard of care dosing with continuous MTD abiraterone.

P1004

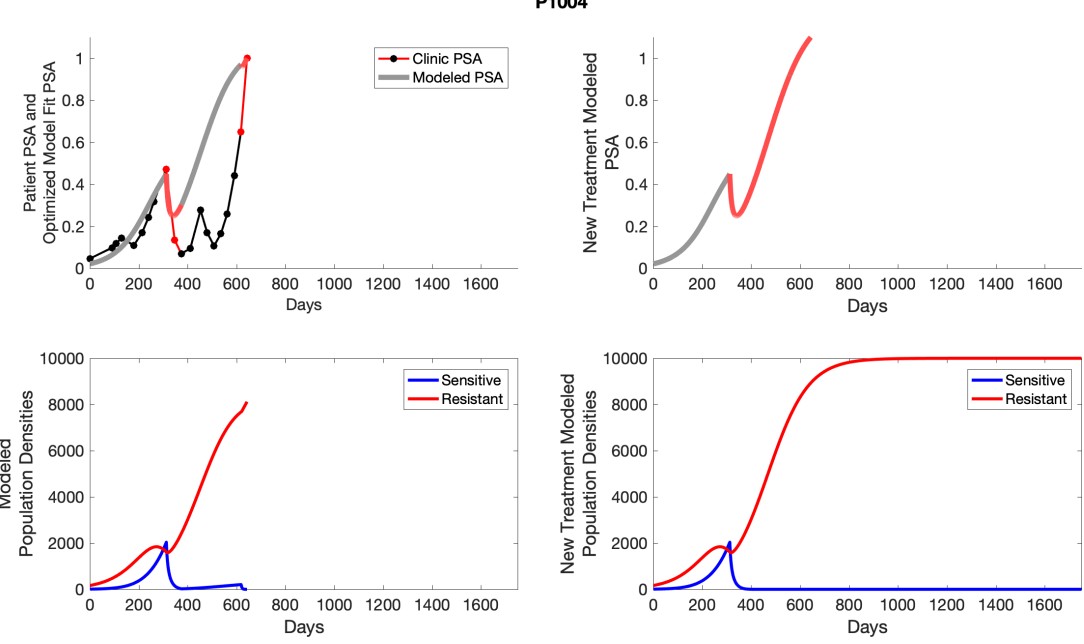

**Appendix 5—figure 19.** The left column shows actual PSA values (upper right panel) with the PSA dynamics of the fitted mathematical model for the indicated subject, the lower left panel shows computer simulations of the estimated population dynamics for sennsitive and resistant cells; the right panels show computer simulations for extended therapy using the standard of care dosing with continuous MTD abiraterone.

P1005

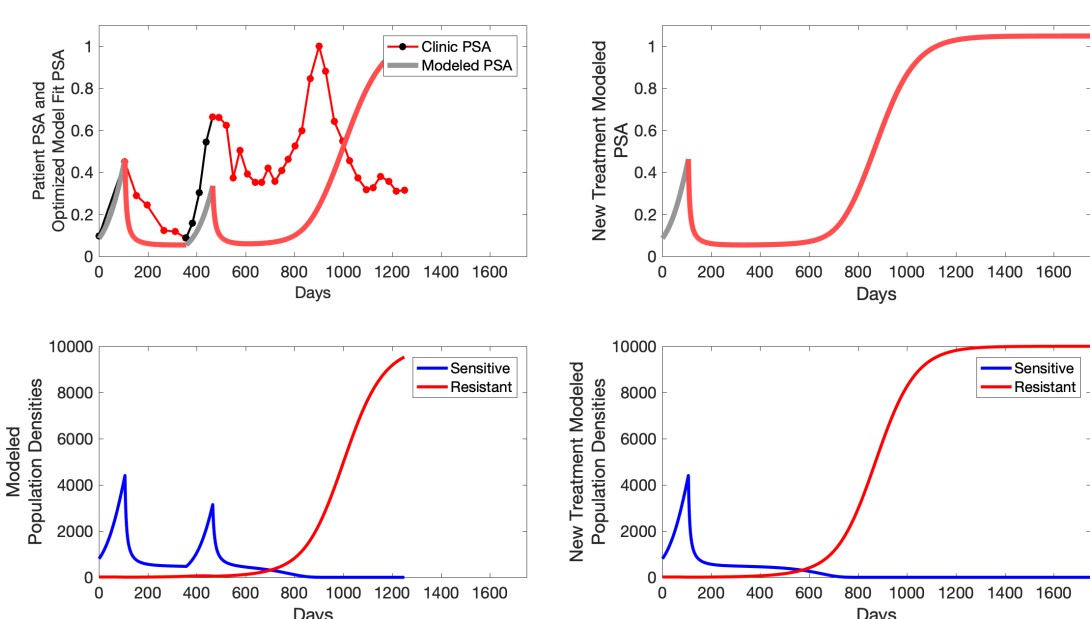

**Appendix 5—figure 20.** The left column shows actual PSA values (upper right panel) with the PSA dynamics of the fitted mathematical model for the indicated subject, the lower left panel shows computer simulations of the estimated population dynamics for sennsitive and resistant cells; the right panels show computer simulations for extended therapy using the standard of care dosing with continuous MTD abiraterone.

P1006

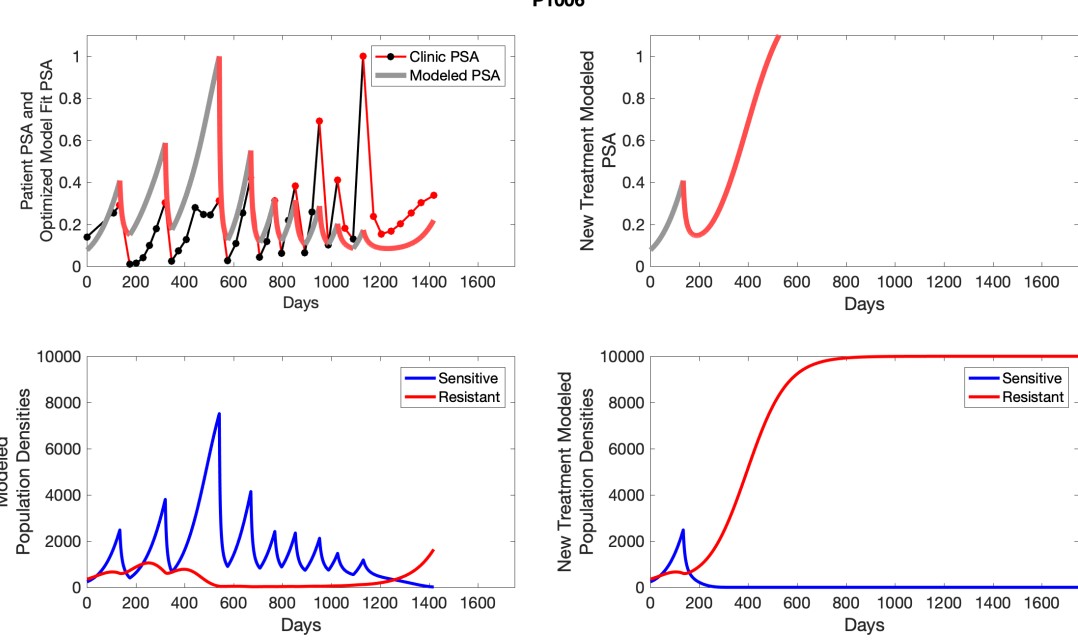

**Appendix 5—figure 21.** The left column shows actual PSA values (upper right panel) with the PSA dynamics of the fitted mathematical model for the indicated subject, the lower left panel shows computer simulations of the estimated population dynamics for sennsitive and resistant cells; the right panels show computer simulations for extended therapy using the standard of care dosing with continuous MTD abiraterone.

**P1007**

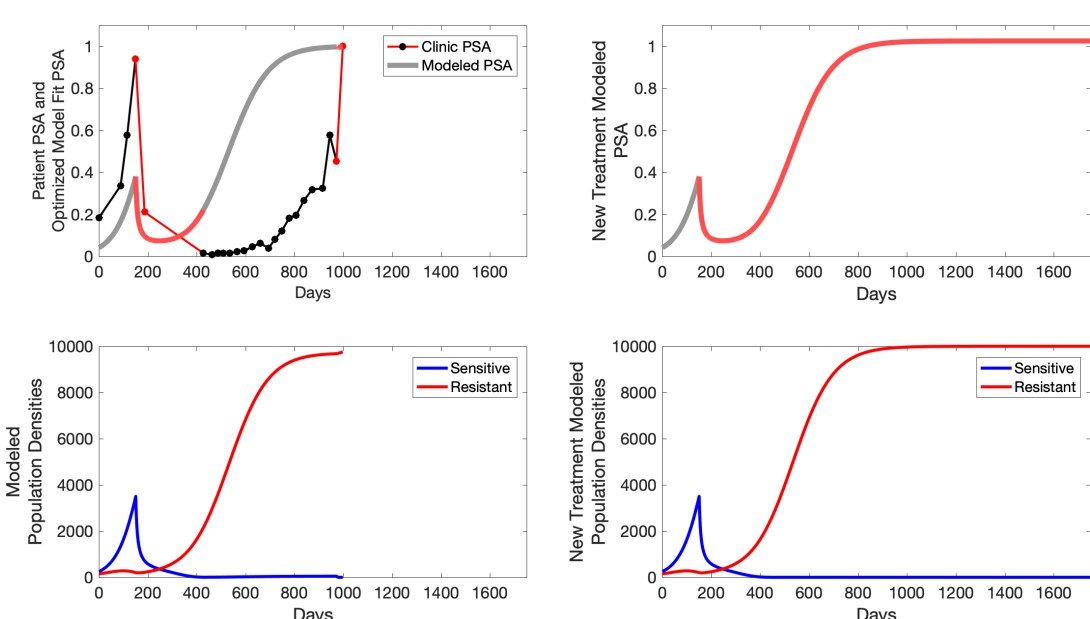

**Appendix 5—figure 22.** The left column shows actual PSA values (upper right panel) with the PSA dynamics of the fitted mathematical model for the indicated subject, the lower left panel shows computer simulations of the estimated population dynamics for sennsitive and resistant cells; the right panels show computer simulations for extended therapy using the standard of care dosing with continuous MTD abiraterone.

**P1009**

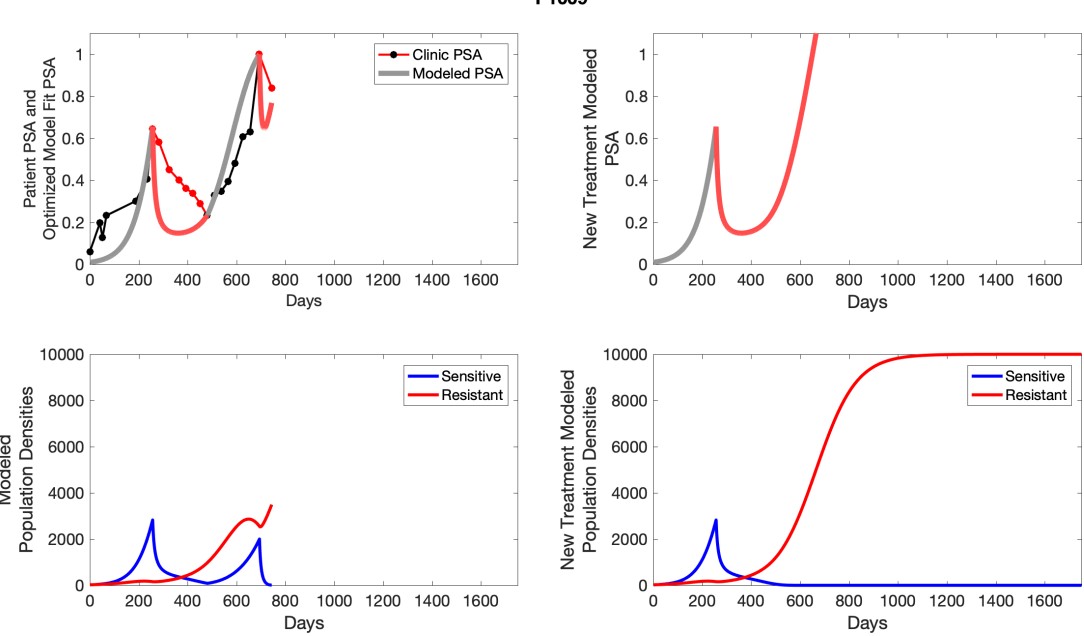

**Appendix 5—figure 23.** The left column shows actual PSA values (upper right panel) with the PSA dynamics of the fitted mathematical model for the indicated subject, the lower left panel shows computer simulations of the estimated population dynamics for sennsitive and resistant cells; the right panels show computer simulations for extended therapy using the standard of care dosing with continuous MTD abiraterone.

**P1010**

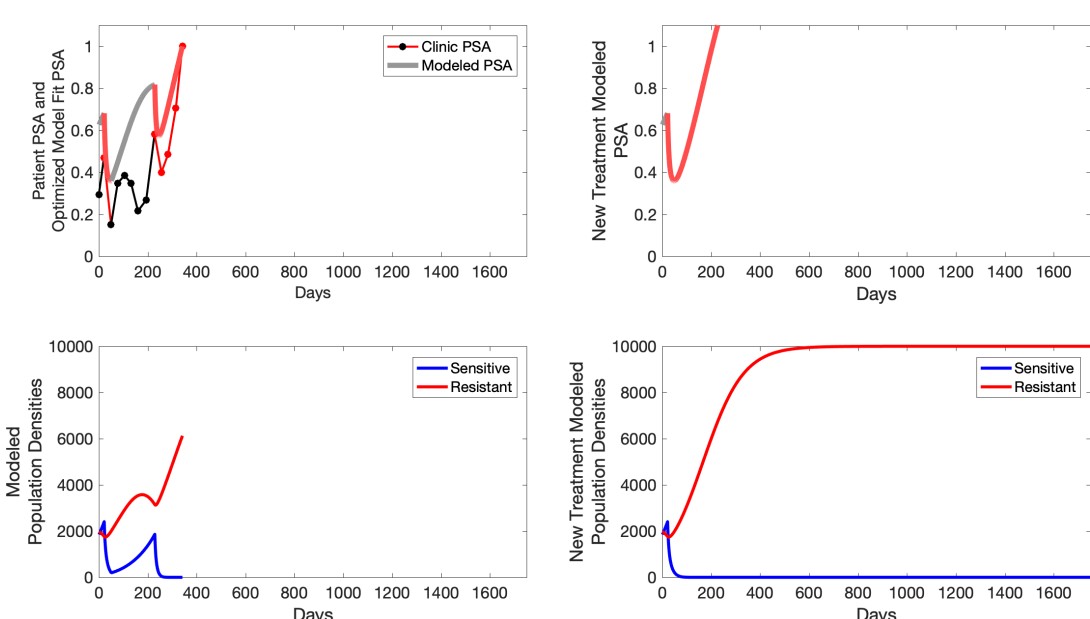

**Appendix 5—figure 24.** The left column shows actual PSA values (upper right panel) with the PSA dynamics of the fitted mathematical model for the indicated subject, the lower left panel shows computer simulations of the estimated population dynamics for sennsitive and resistant cells; the right panels show computer simulations for extended therapy using the standard of care dosing with continuous MTD abiraterone.

**P1011**

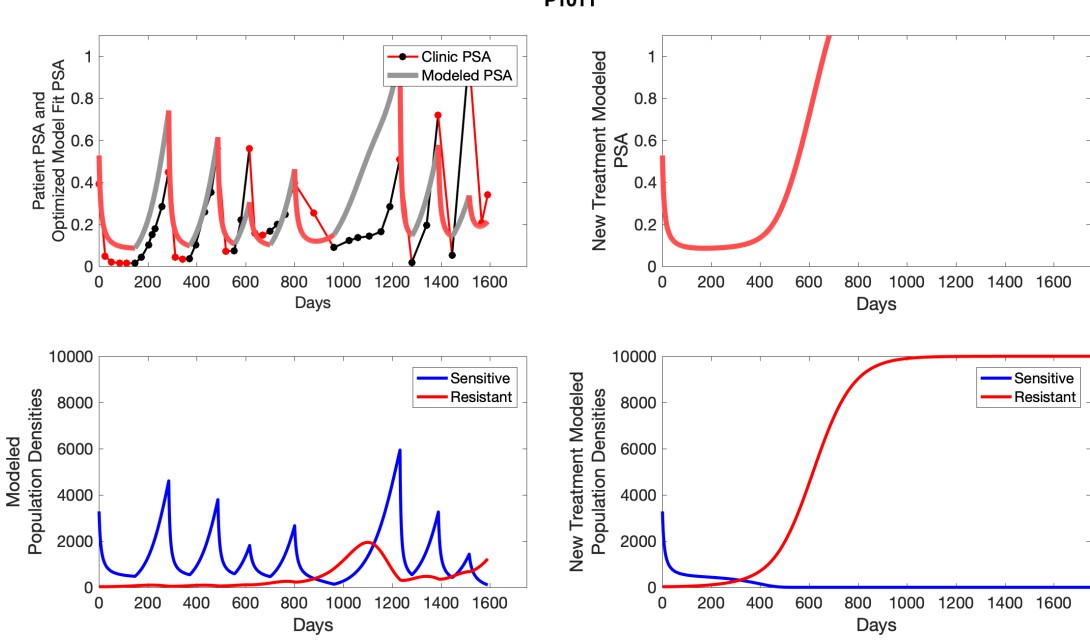

**Appendix 5—figure 25.** The left column shows actual PSA values (upper right panel) with the PSA dynamics of the fitted mathematical model for the indicated subject, the lower left panel shows computer simulations of the estimated population dynamics for sennsitive and resistant cells; the right panels show computer simulations for extended therapy using the standard of care dosing with continuous MTD abiraterone.

P1012

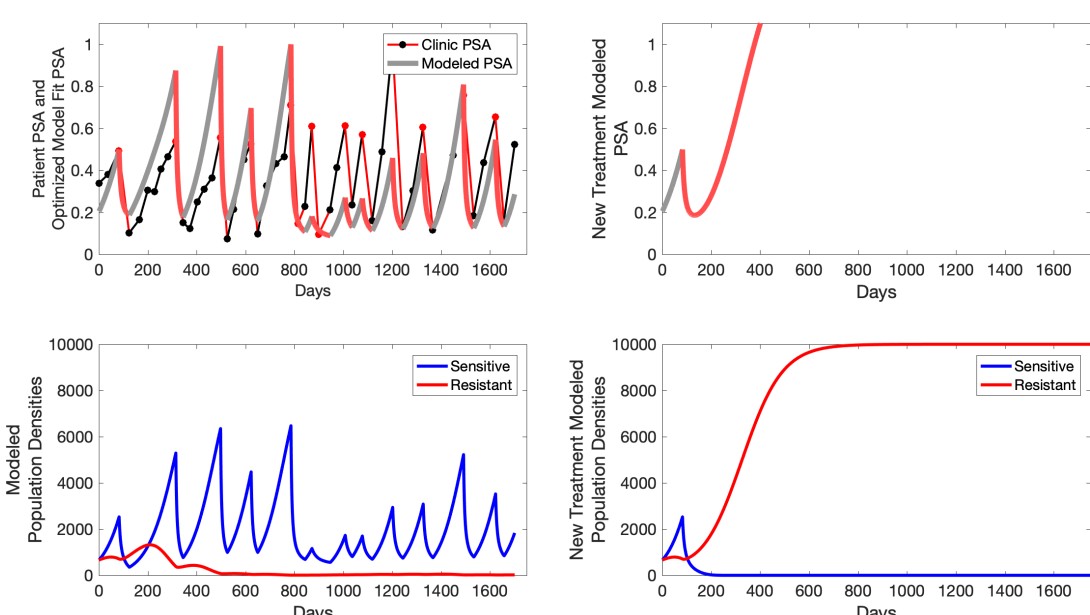

**Appendix 5—figure 26.** The left column shows actual PSA values (upper right panel) with the PSA dynamics of the fitted mathematical model for the indicated subject, the lower left panel shows computer simulations of the estimated population dynamics for sennsitive and resistant cells; the right panels show computer simulations for extended therapy using the standard of care dosing with continuous MTD abiraterone.

P1014

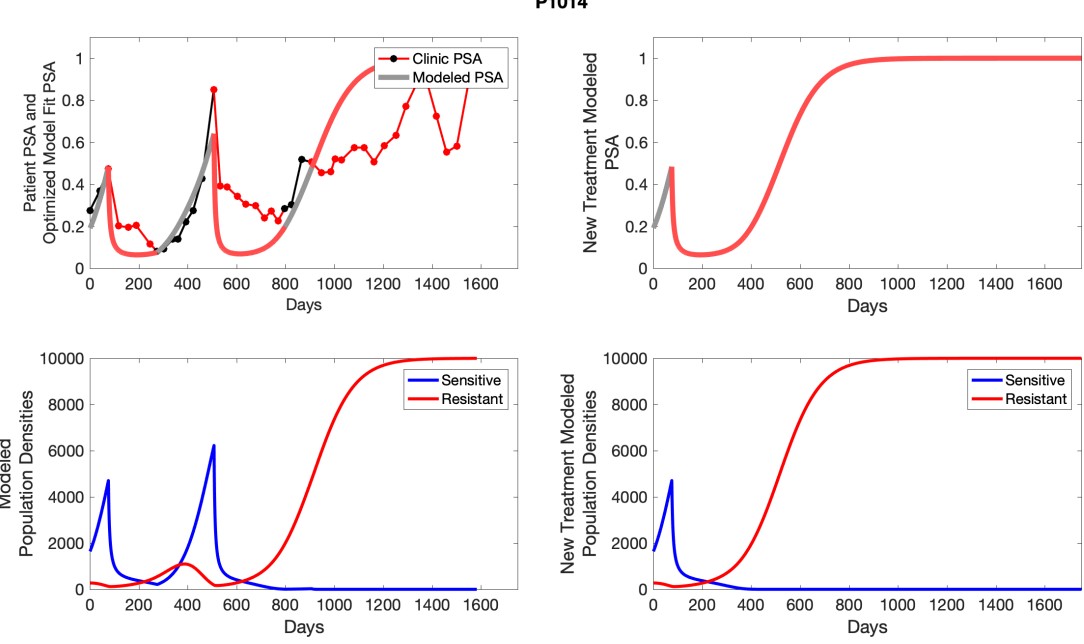

**Appendix 5—figure 27.** The left column shows actual PSA values (upper right panel) with the PSA dynamics of the fitted mathematical model for the indicated subject, the lower left panel shows computer simulations of the estimated population dynamics for sennsitive and resistant cells; the right panels show computer simulations for extended therapy using the standard of care dosing with continuous MTD abiraterone.

P1015

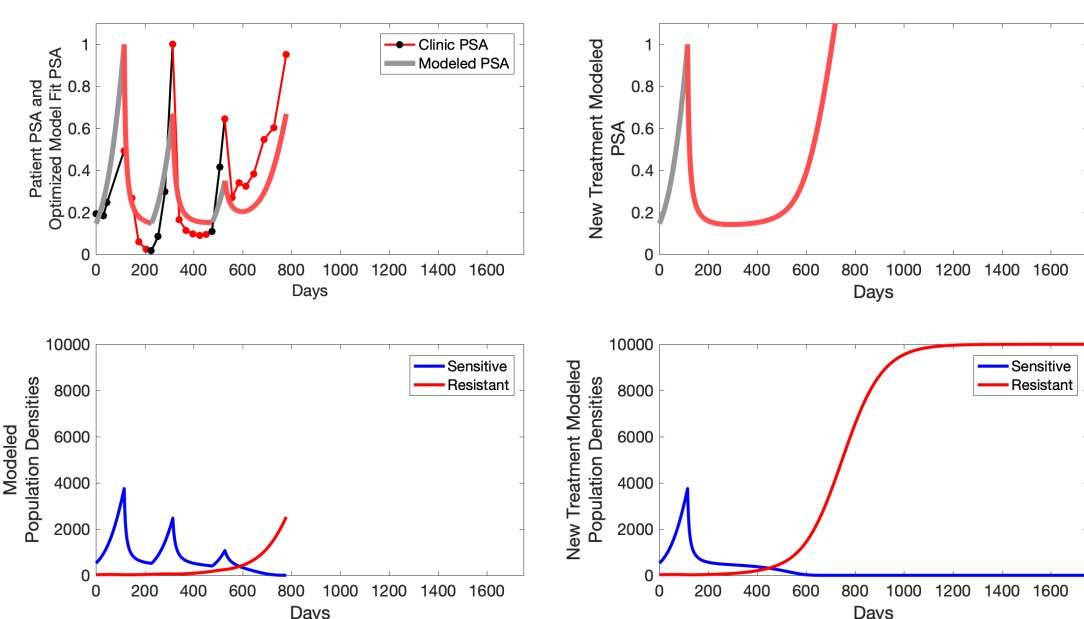

**Appendix 5—figure 28.** The left column shows actual PSA values (upper right panel) with the PSA dynamics of the fitted mathematical model for the indicated subject, the lower left panel shows computer simulations of the estimated population dynamics for sennsitive and resistant cells; the right panels show computer simulations for extended therapy using the standard of care dosing with continuous MTD abiraterone.

P1016

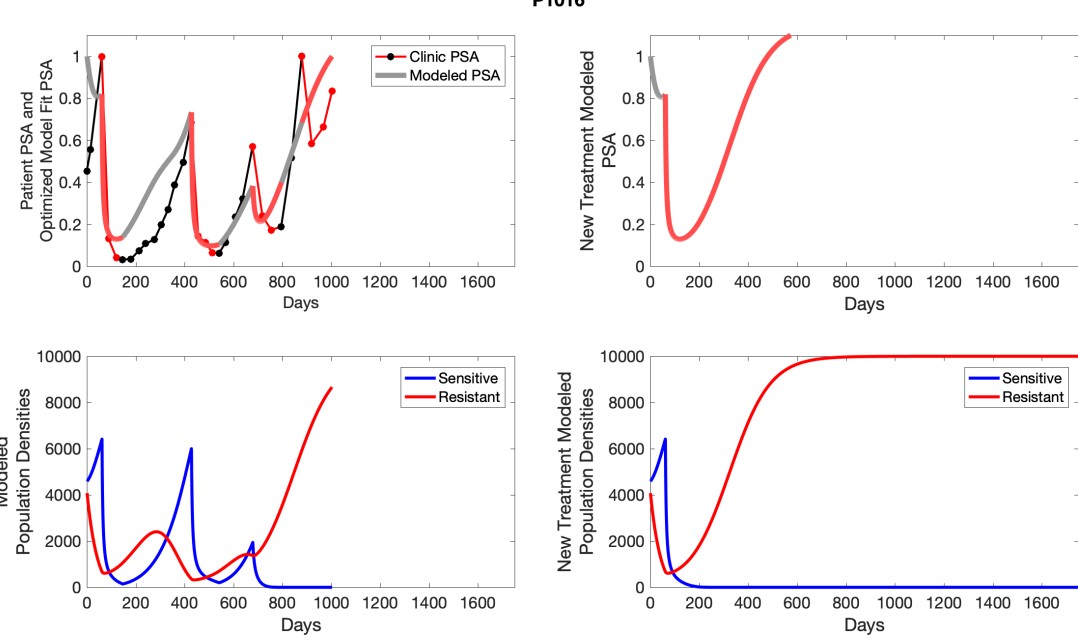

**Appendix 5—figure 29.** The left column shows actual PSA values (upper right panel) with the PSA dynamics of the fitted mathematical model for the indicated subject, the lower left panel shows computer simulations of the estimated population dynamics for sennsitive and resistant cells; the right panels show computer simulations for extended therapy using the standard of care dosing with continuous MTD abiraterone.

P1017

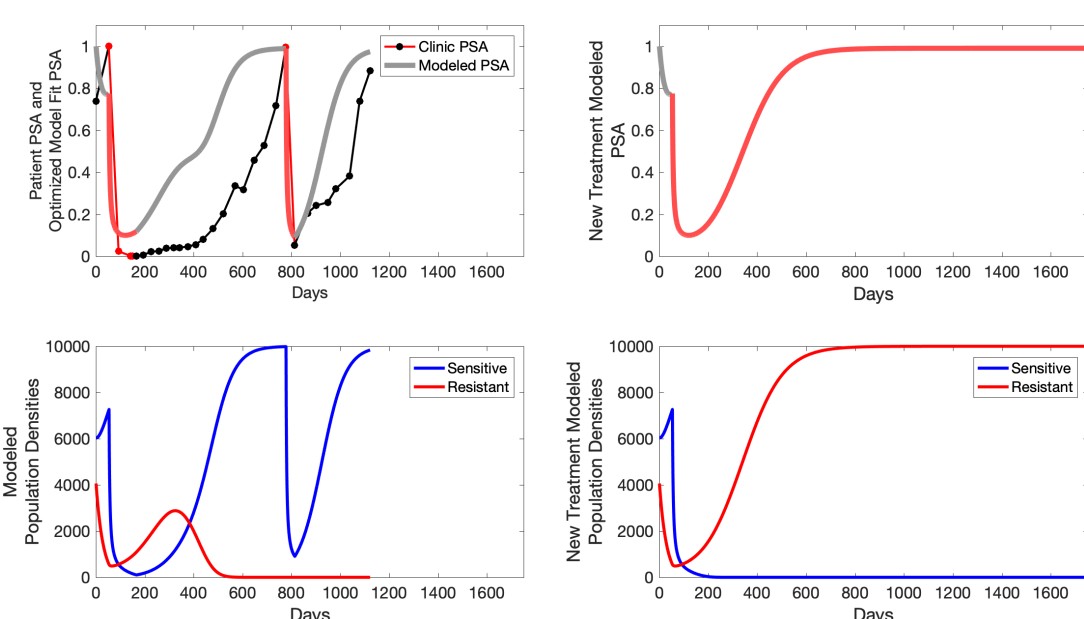

**Appendix 5—figure 30.** The left column shows actual PSA values (upper right panel) with the PSA dynamics of the fitted mathematical model for the indicated subject, the lower left panel shows computer simulations of the estimated population dynamics for sennsitive and resistant cells; the right panels show computer simulations for extended therapy using the standard of care dosing with continuous MTD abiraterone.

P1018

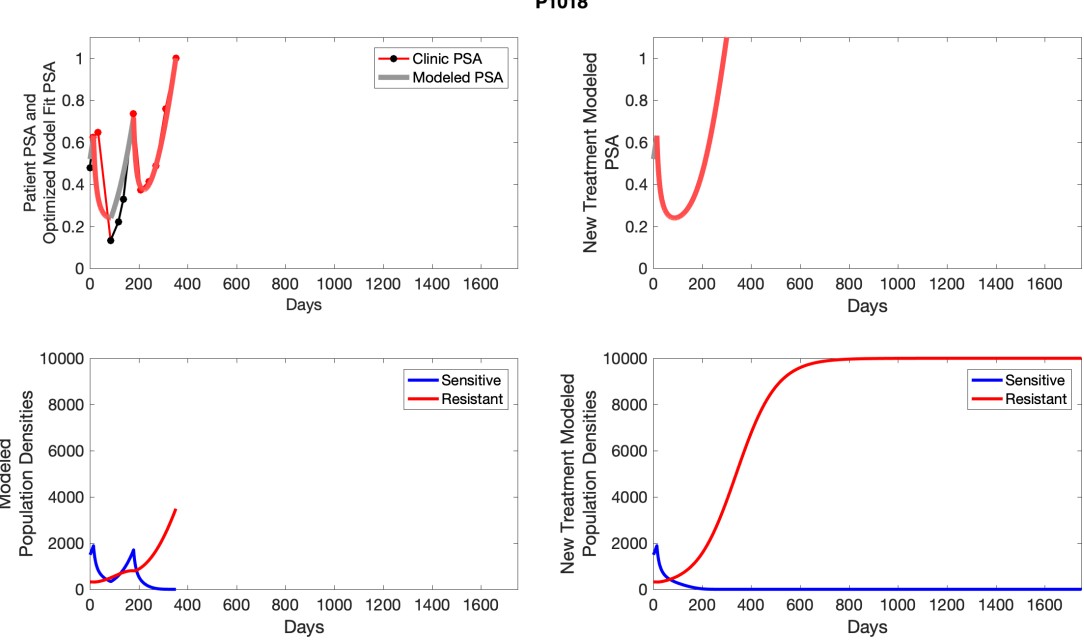

**Appendix 5—figure 31.** The left column shows actual PSA values (upper right panel) with the PSA dynamics of the fitted mathematical model for the indicated subject, the lower left panel shows computer simulations of the estimated population dynamics for sennsitive and resistant cells; the right panels show computer simulations for extended therapy using the standard of care dosing with continuous MTD abiraterone.

P1020

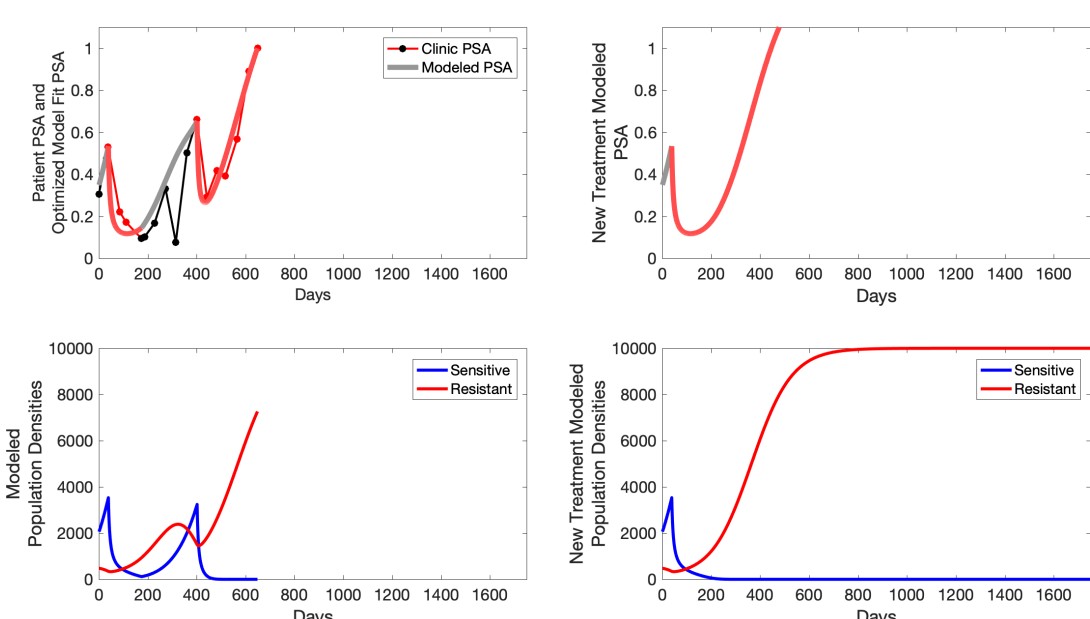

**Appendix 5—figure 32.** The left column shows actual PSA values (upper right panel) with the PSA dynamics of the fitted mathematical model for the indicated subject, the lower left panel shows computer simulations of the estimated population dynamics for sennsitive and resistant cells; the right panels show computer simulations for extended therapy using the standard of care dosing with continuous MTD abirateron.

## Appendix 6

## Simulated intermittent therapy

To evaluate an intermittent therapy (one that had not been used as a trial arm), we ran a simulation for each patient (both cohorts) with a treatment protocol that started with an 8-month (243-day) induction period beginning at the time abiraterone was first administered clinically to each patient, which was not always at t = 0. Following the induction period, abiraterone was then discontinued until either another 243 days had passed or until the patient's PSA returned to the patient's initial baseline PSA level. If abiraterone is reinstated, it remains for 243 days. For each patient, we initiated model runs using the patient-specific parameters from *Appendix 2—table 3*.

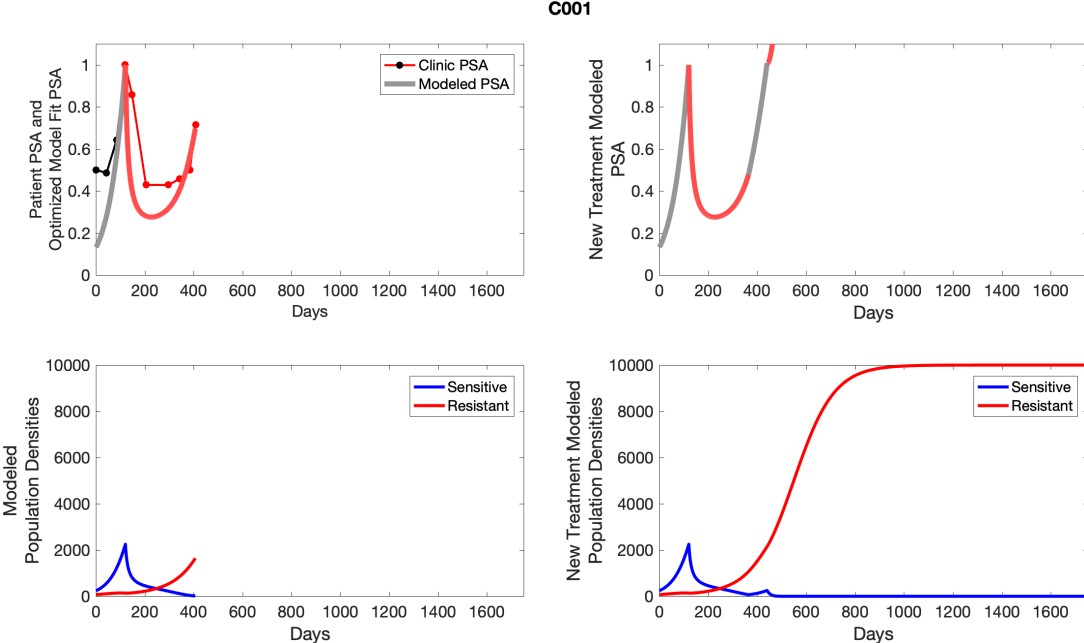

**Appendix 6—figure 1.** The left column shows PSA changes during treatment and associated model fit for each captioned patient (upper panel) as well as computer simulations for the population dynamics of the resistant and sensitive cells; the right column shows computer simulation in the same patient for a typical "intermittent therapy" protocol in which cycling therapy is insituted only after an 8 month induction period of continuous MTD application of abiraterone.

**C002**

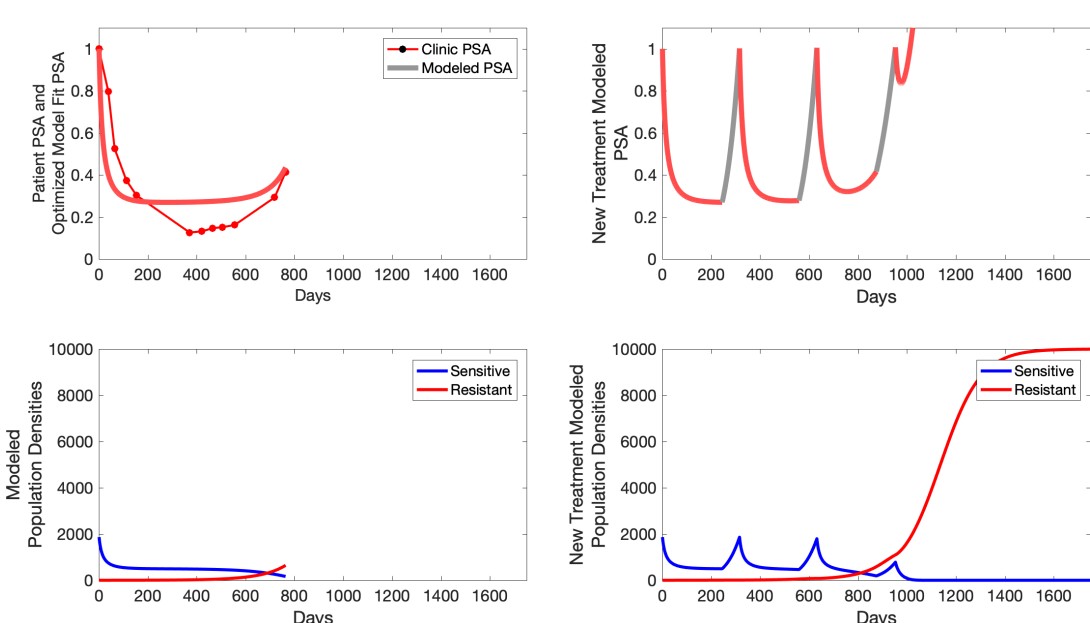

**Appendix 6—figure 2.** The left column shows PSA changes during treatment and associated model fit for each captioned patient (upper panel) as well as computer simulations for the population dynamics of the resistant and sensitive cells (lower panel); the right column shows computer simulation in the same patient for a typical "intermittent therapy" protocol in which cycling therapy is insituted only after an 8 month induction period of continuous MTD application of abiraterone.

**C003**

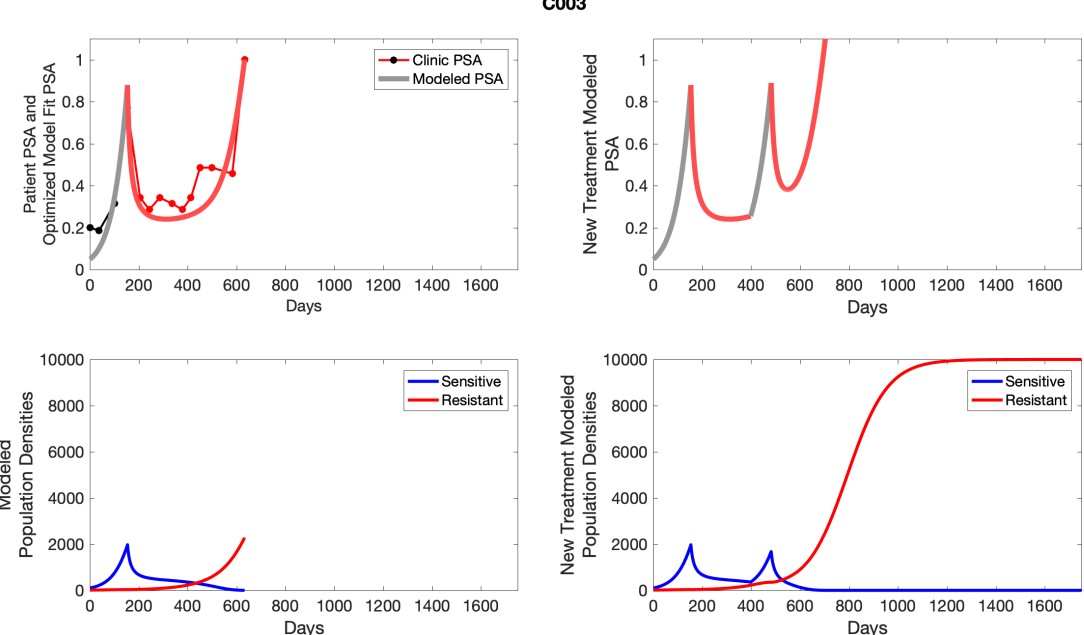

**Appendix 6—figure 3.** The left column shows PSA changes during treatment and associated model fit for each captioned patient (upper panel) as well as computer simulations for the population dynamics of the resistant and sensitive cells; the right column shows computer simulation in the same patient for a typical "intermittent therapy" protocol in which cycling therapy is insituted only after an 8 month induction period of continuous MTD application of abiraterone.

**C004**

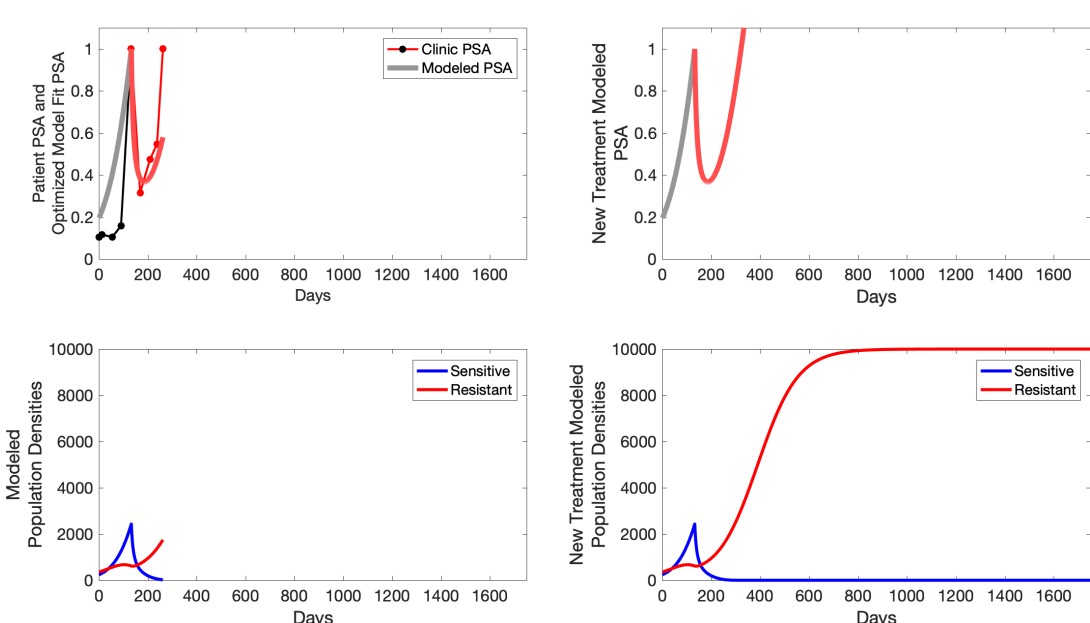

**Appendix 6—figure 4.** The left column shows PSA changes during treatment and associated model fit for each captioned patient (upper panel) as well as computer simulations for the population dynamics of the resistant and sensitive cells; the right column shows computer simulation in the same patient for a typical "intermittent therapy" protocol in which cycling therapy is insituted only after an 8 month induction period of continuous MTD application of abiraterone.

**C005**

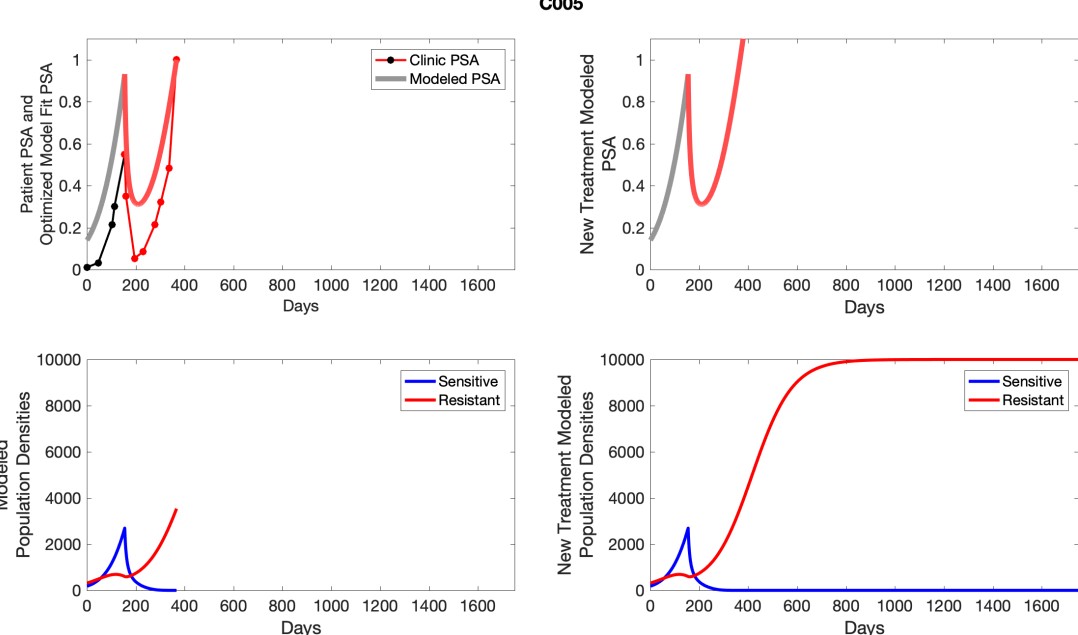

**Appendix 6—figure 5.** The left column shows PSA changes during treatment and associated model fit for each captioned patient (upper panel) as well as computer simulations for the population dynamics of the resistant and sensitive cells; the right column shows computer simulation in the same patient for a typical "intermittent therapy" protocol in which cycling therapy is insituted only after an 8 month induction period of continuous MTD application of abiraterone.

**C006**

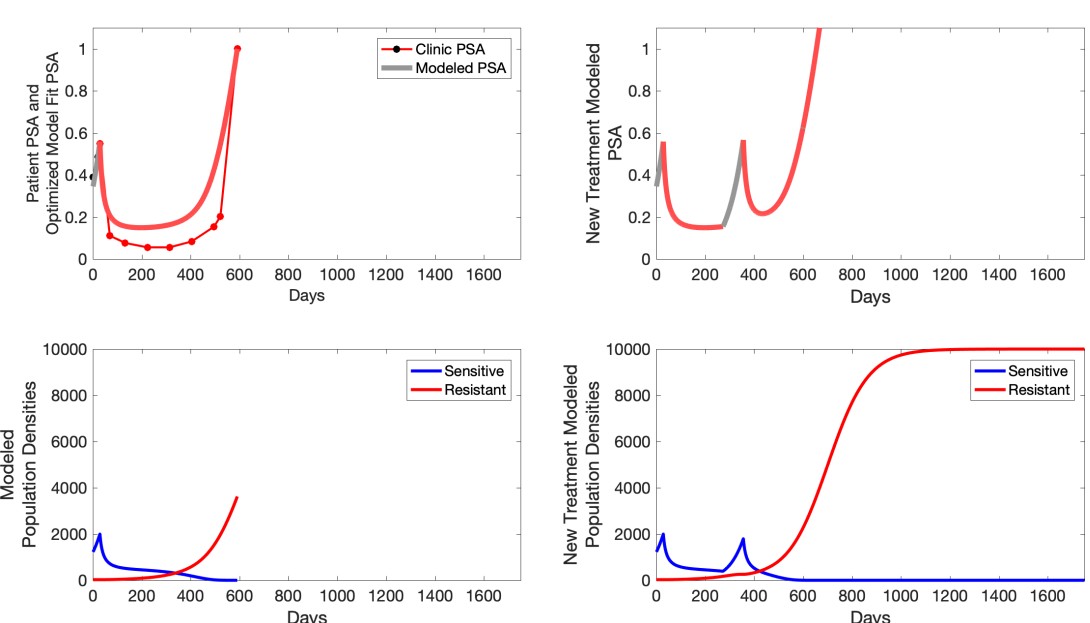

**Appendix 6—figure 6.** The left column shows PSA changes during treatment and associated model fit for each captioned patient (upper right panel) as well as computer simulations for the population dynamics of the resistant and sensitive cells; the right column shows computer simulation in the same patient for a typical "intermittent therapy" protocol in which cycling therapy is insituted only after an 8 month induction period of continuous MTD application of abiraterone.

**C007**

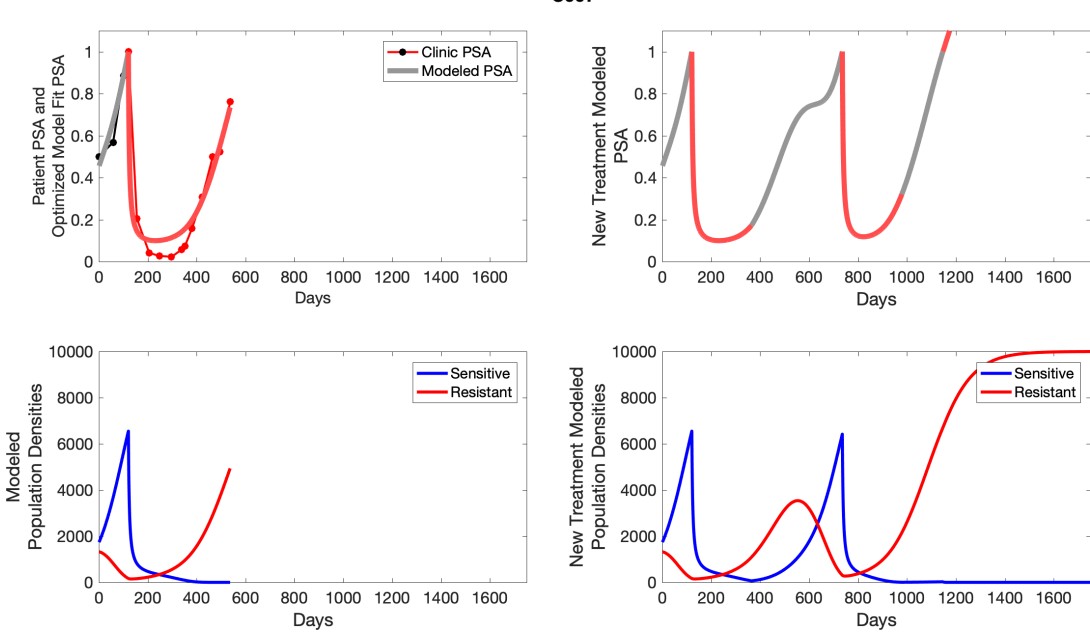

**Appendix 6—figure 7.** The left column shows PSA changes during treatment and associated model fit for each captioned patient (upper panel) as well as computer simulations for the population dynamics of the resistant and sensitive cells; the right column shows computer simulation in the same patient for a typical "intermittent therapy" protocol in which cycling therapy is insituted only after an 8 month induction period of continuous MTD application of abiraterone.

**C008**

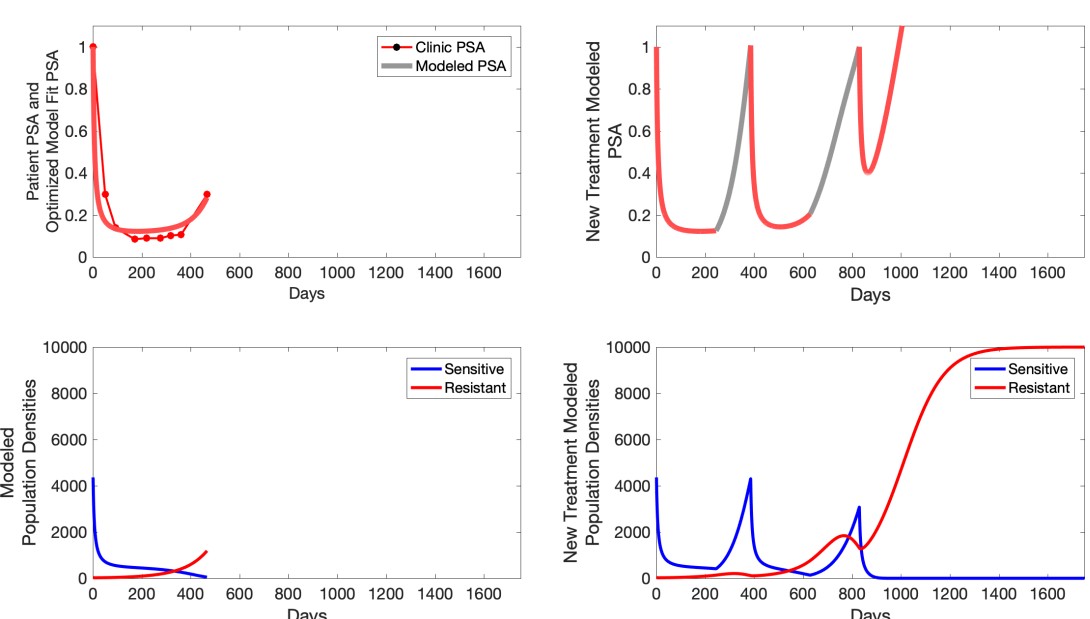

**Appendix 6—figure 8.** The left column shows PSA changes during treatment and associated model fit for each captioned patient (upper panel) as well as computer simulations for the population dynamics of the resistant and sensitive cells; the right column shows computer simulation in the same patient for a typical "intermittent therapy" protocol in which cycling therapy is insituted only after an 8 month induction period of continuous MTD application of abiraterone.

**C009**

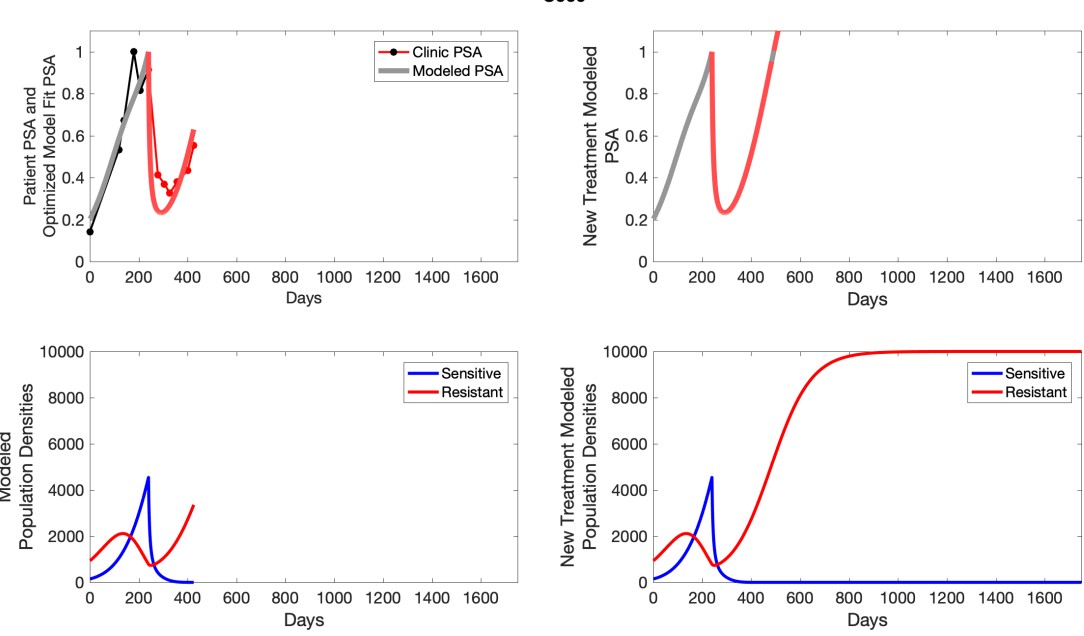

**Appendix 6—figure 9.** The left column shows PSA changes during treatment and associated model fit for each captioned patient (upper panel) as well as computer simulations for the population dynamics of the resistant and sensitive cells; the right column shows computer simulation in the same patient for a typical "intermittent therapy" protocol in which cycling therapy is insituted only after an 8 month induction period of continuous MTD application of abiraterone.

**C010**

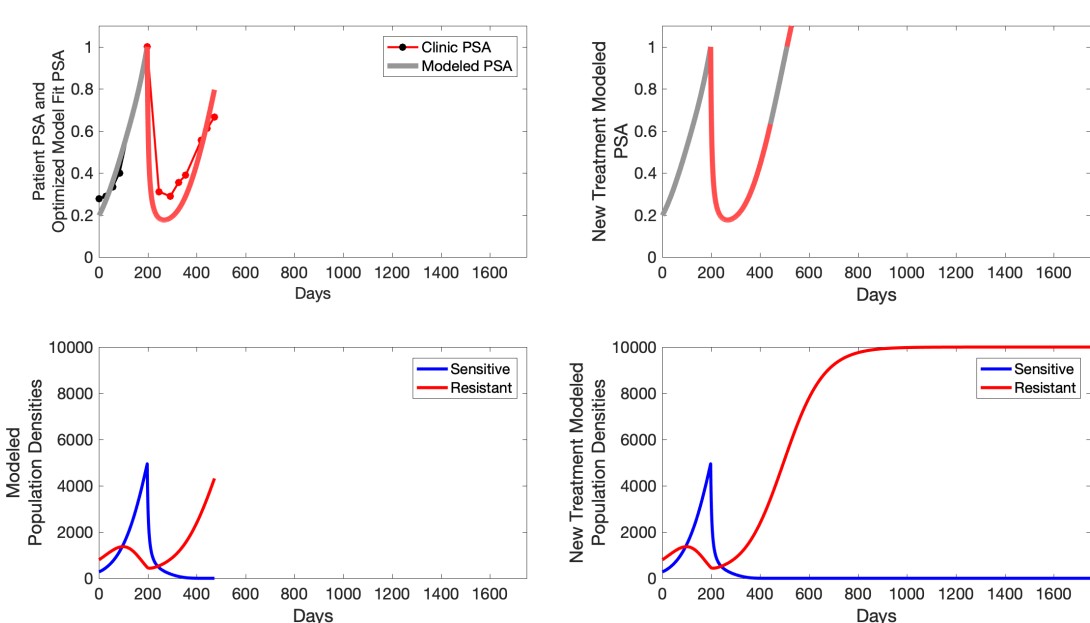

**Appendix 6—figure 10.** The left column shows PSA changes during treatment and associated model fit for each captioned patient (upper panel) as well as computer simulations for the population dynamics of the resistant and sensitive cells; the right column shows computer simulation in the same patient for a typical "intermittent therapy" protocol in which cycling therapy is insituted only after an 8 month induction period of continuous MTD application of abiraterone.

**C011**

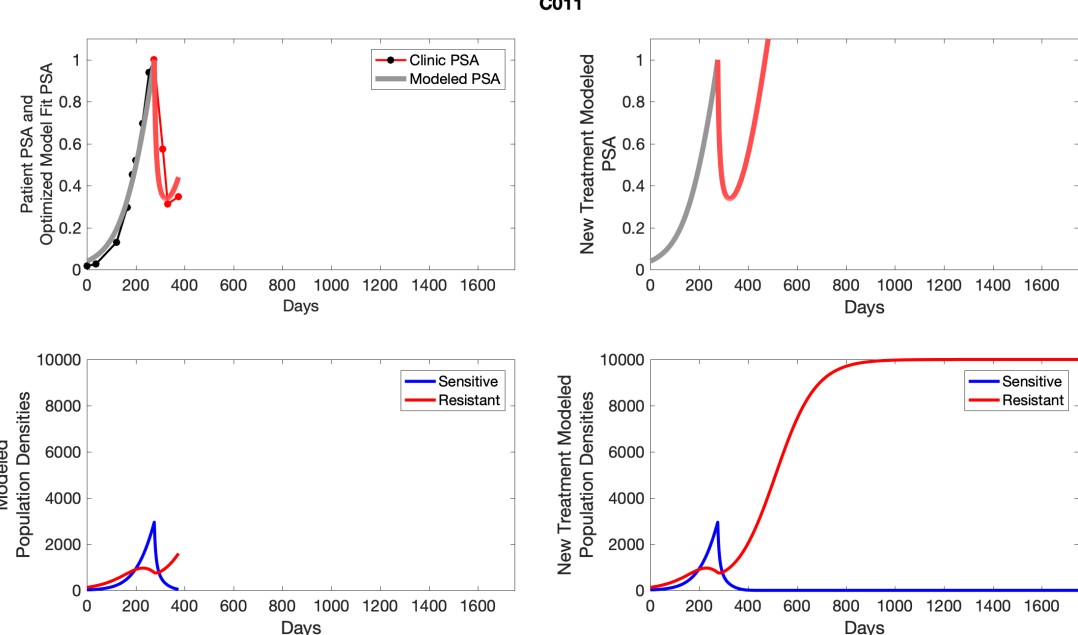

**Appendix 6—figure 11.** The left column shows PSA changes during treatment and associated model fit for each captioned patient (upper panel) as well as computer simulations for the population dynamics of the resistant and sensitive cells; the right column shows computer simulation in the same patient for a typical "intermittent therapy" protocol in which cycling therapy is insituted only after an 8 month induction period of continuous MTD application of abiraterone.

**C012**

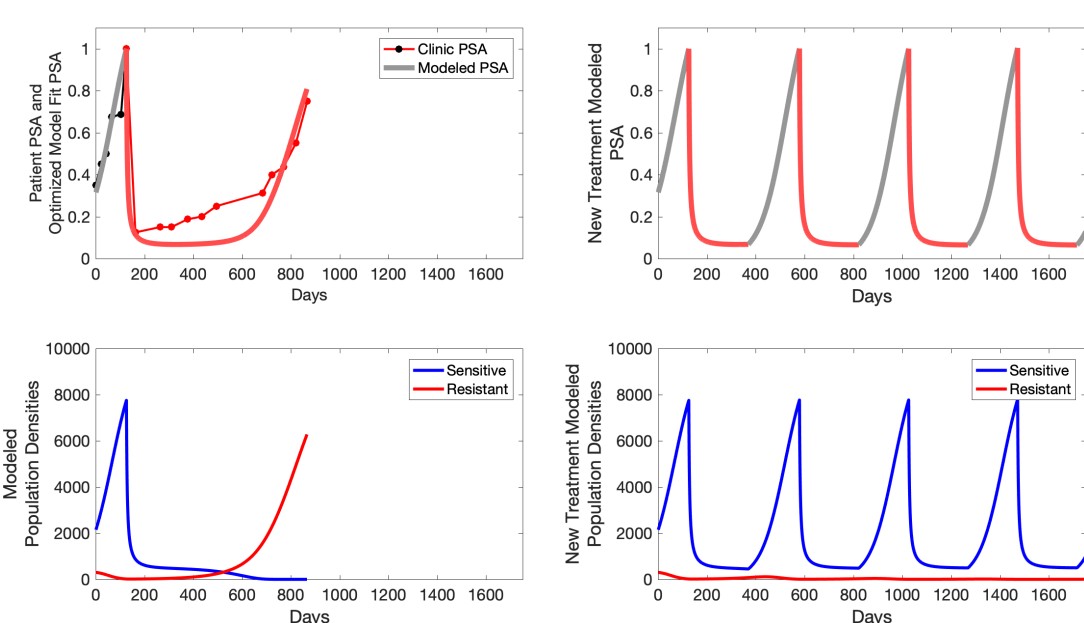

**Appendix 6—figure 12.** The left column shows PSA changes during treatment and associated model fit for each captioned patient (upper panel) as well as computer simulations for the population dynamics of the resistant and sensitive cells; the right column shows computer simulation in the same patient for a typical "intermittent therapy" protocol in which cycling therapy is insituted only after an 8 month induction period of continuous MTD application of abiraterone.

**C013**

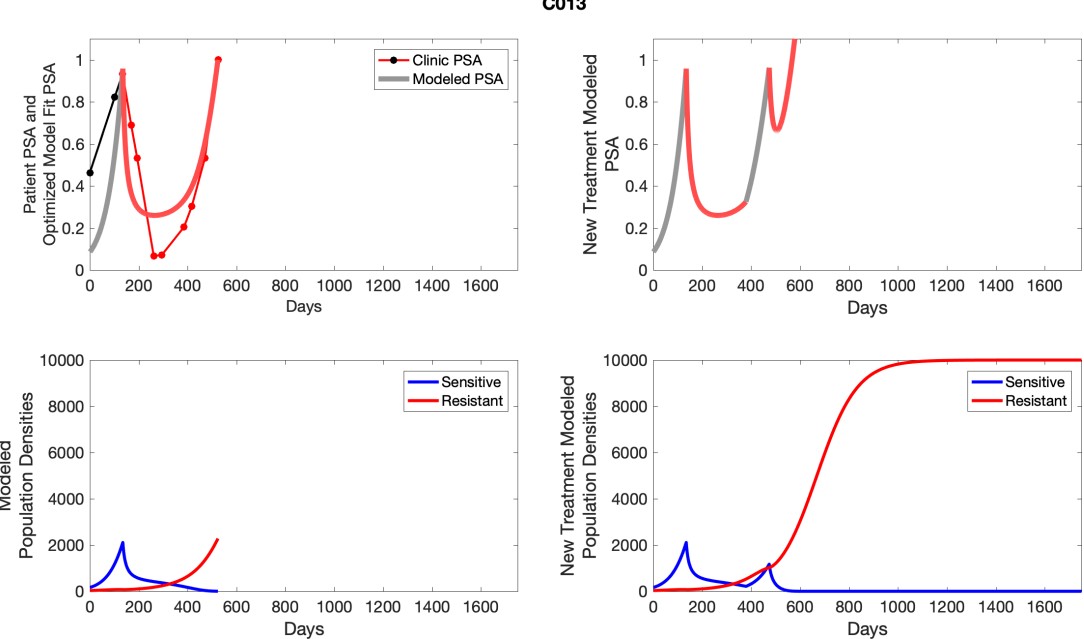

**Appendix 6—figure 13.** The left column shows PSA changes during treatment and associated model fit for each captioned patient (upper panel) as well as computer simulations for the population dynamics of the resistant and sensitive cells; the right column shows computer simulation in the same patient for a typical "intermittent therapy" protocol in which cycling therapy is insituted only after an 8 month induction period of continuous MTD application of abiraterone.

**C014**

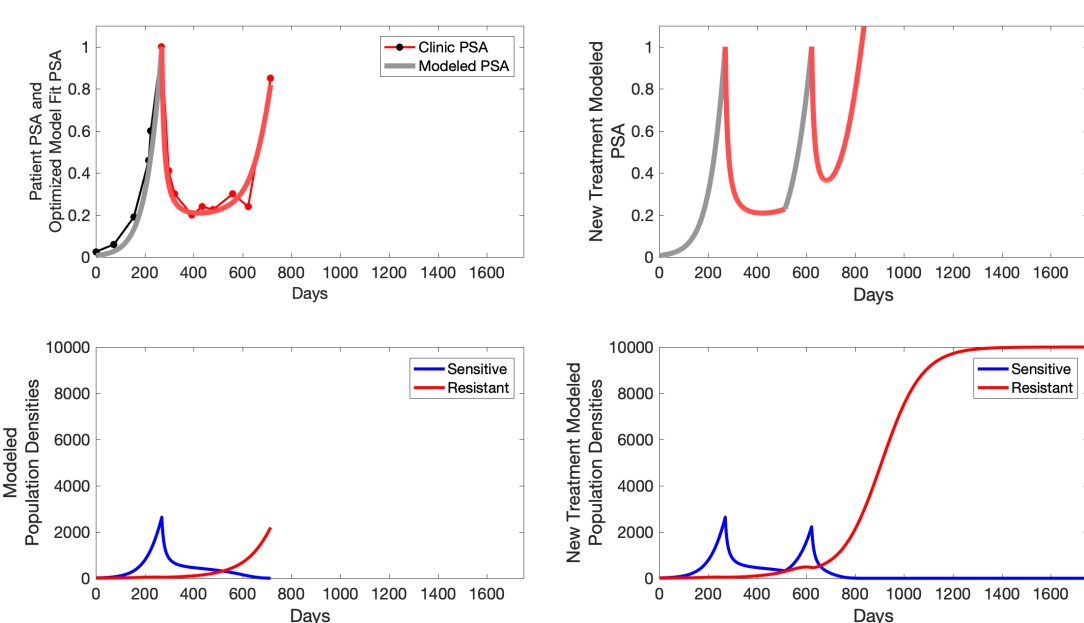

**Appendix 6—figure 14.** The left column shows PSA changes during treatment and associated model fit for each captioned patient (upper right panel) as well as computer simulations for the population dynamics of the resistant and sensitive cells; the right column shows computer simulation in the same patient for a typical "intermittent therapy" protocol in which cycling therapy is insituted only after an 8 month induction period of continuous MTD application of abiraterone.

**C015**

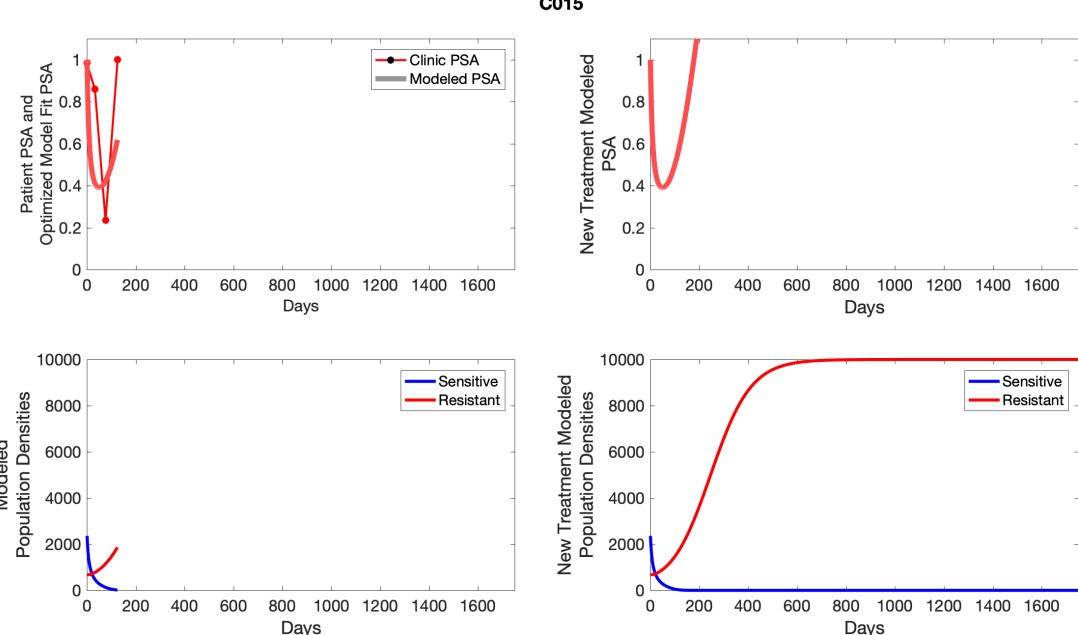

**Appendix 6—figure 15.** The left column shows PSA changes during treatment and associated model fit for each captioned patient (upper panel) as well as computer simulations for the population dynamics of the resistant and sensitive cells; the right column shows computer simulation in the same patient for a typical "intermittent therapy" protocol in which cycling therapy is insituted only after an 8 month induction period of continuous MTD application of abiraterone.

**P1001**

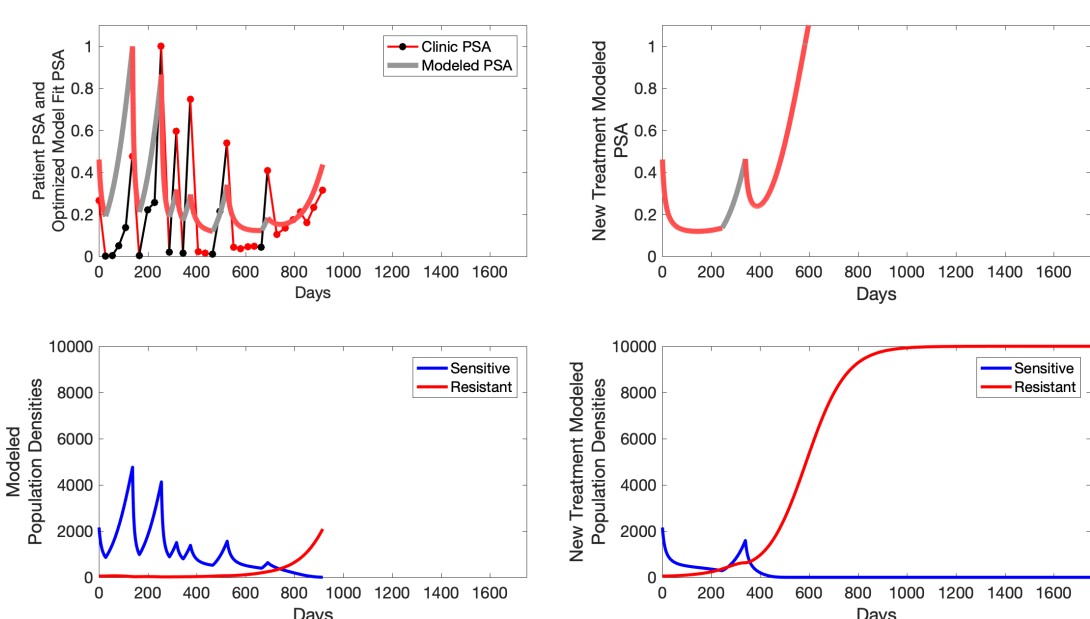

**Appendix 6—figure 16.** The left column shows PSA changes during treatment and associated model fit for each captioned patient (upper panel) as well as computer simulations for the population dynamics of the resistant and sensitive cells; the right column shows computer simulation in the same patient for a typical "intermittent therapy" protocol in which cycling therapy is insituted only after an 8 month induction period of continuous MTD application of abiraterone.

**P1002**

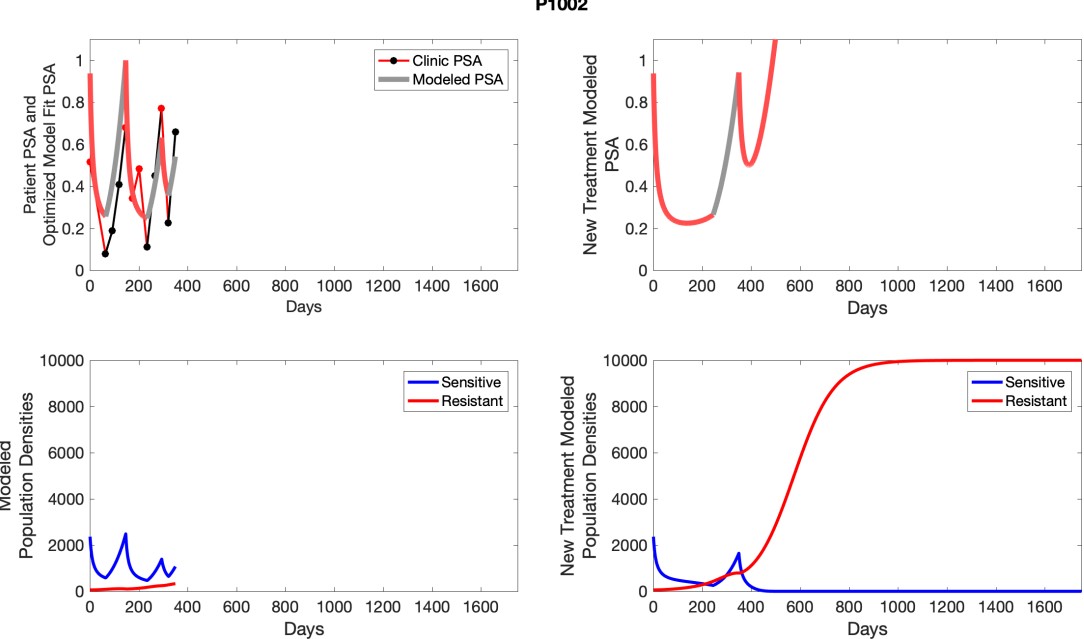

**Appendix 6—figure 17.** The left column shows PSA changes during treatment and associated model fit for each captioned patient (upper panel) as well as computer simulations for the population dynamics of the resistant and sensitive cells; the right column shows computer simulation in the same patient for a typical "intermittent therapy" protocol in which cycling therapy is insituted only after an 8 month induction period of continuous MTD application of abiraterone.

P1003

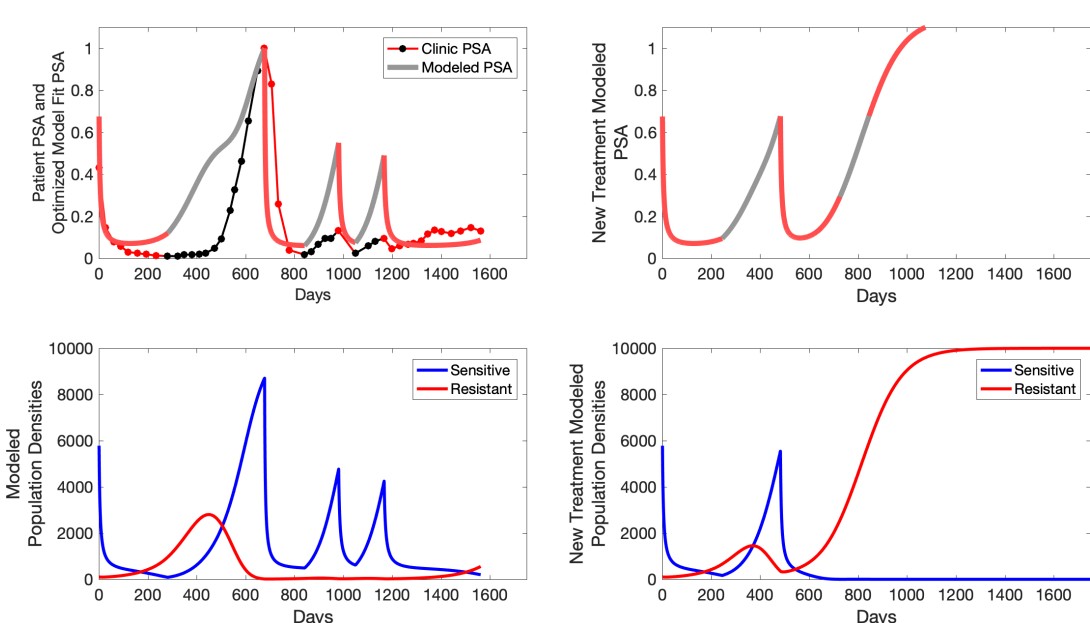

**Appendix 6—figure 18.** The left column shows PSA changes during treatment and associated model fit for each captioned patient (upper panel) as well as computer simulations for the population dynamics of the resistant and sensitive cells; the right column shows computer simulation in the same patient for a typical "intermittent therapy" protocol in which cycling therapy is insituted only after an 8 month induction period of continuous MTD application of abiraterone.

P1004

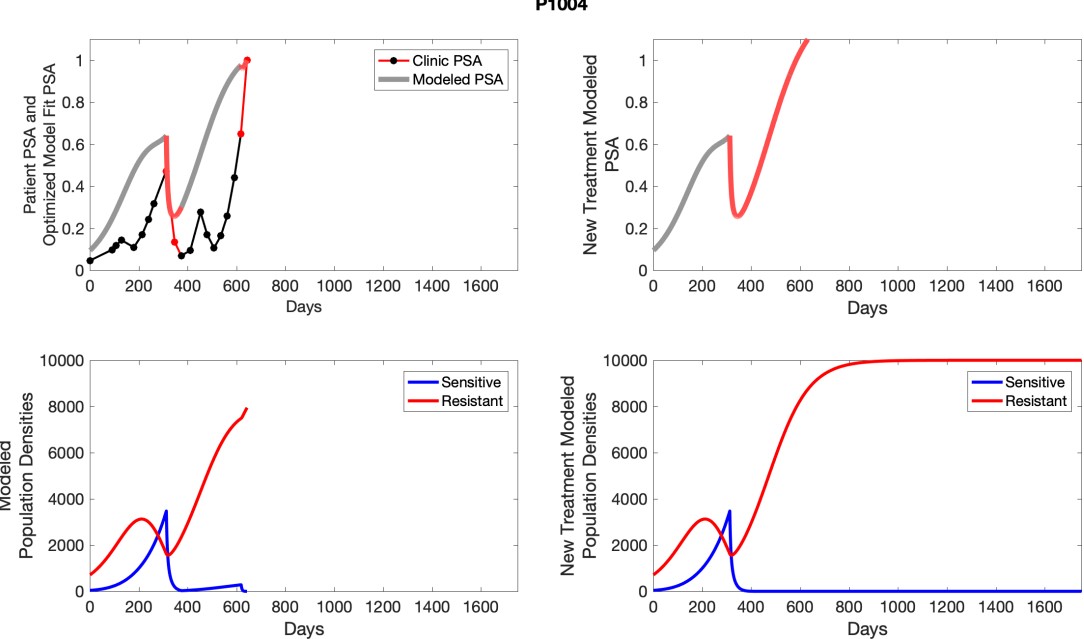

**Appendix 6—figure 19.** The left column shows PSA changes during treatment and associated model fit for each captioned patient (upper panel) as well as computer simulations for the population dynamics of the resistant and sensitive cells; the right column shows computer simulation in the same patient for a typical "intermittent therapy" protocol in which cycling therapy is insituted only after an 8 month induction period of continuous MTD application of abiraterone.

P1005

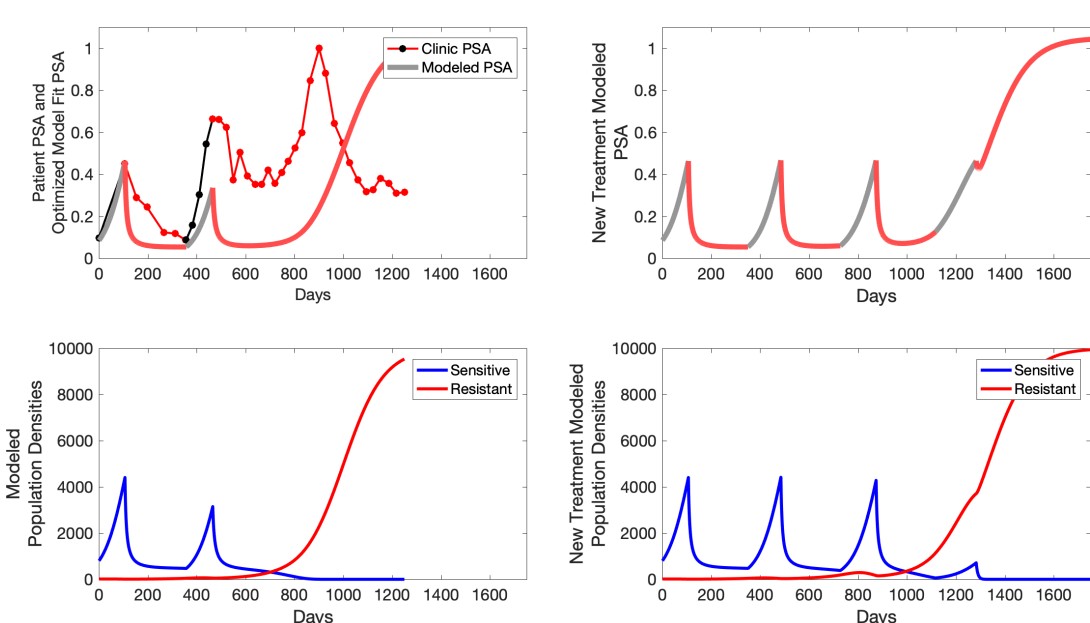

**Appendix 6—figure 20.** The left column shows PSA changes during treatment and associated model fit for each captioned patient (upper panel) as well as computer simulations for the population dynamics of the resistant and sensitive cells; the right column shows computer simulation in the same patient for a typical "intermittent therapy" protocol in which cycling therapy is insituted only after an 8 month induction period of continuous MTD application of abiraterone.

P1006

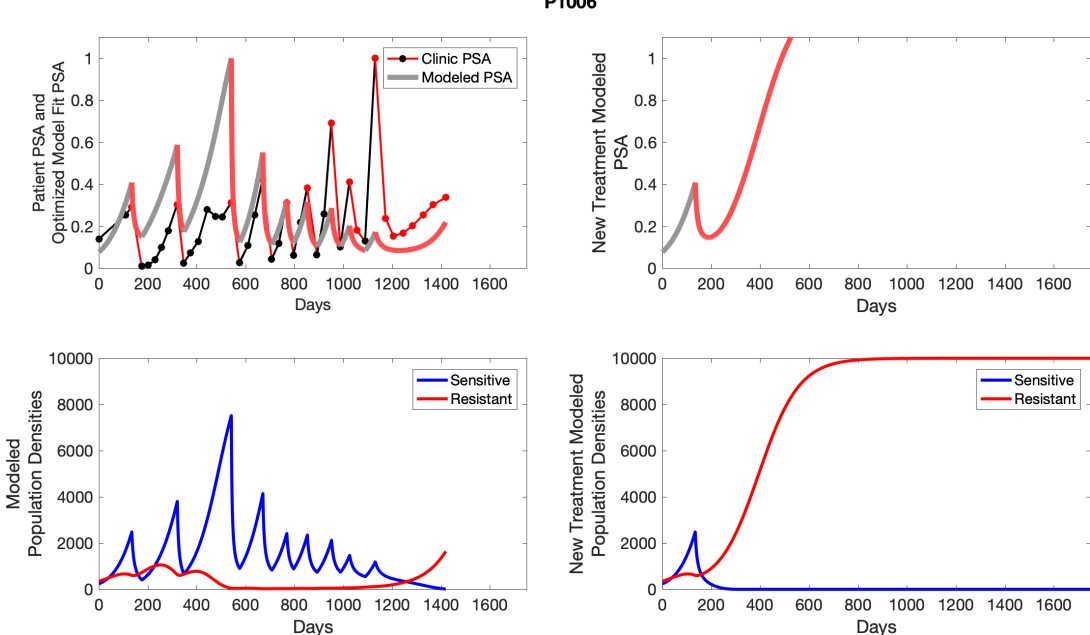

**Appendix 6—figure 21.** The left column shows PSA changes during treatment and associated model fit for each captioned patient (upper panel) as well as computer simulations for the population dynamics of the resistant and sensitive cells; the right column shows computer simulation in the same patient for a typical "intermittent therapy" protocol in which cycling therapy is insituted only after an 8 month induction period of continuous MTD application of abiraterone.

P1007

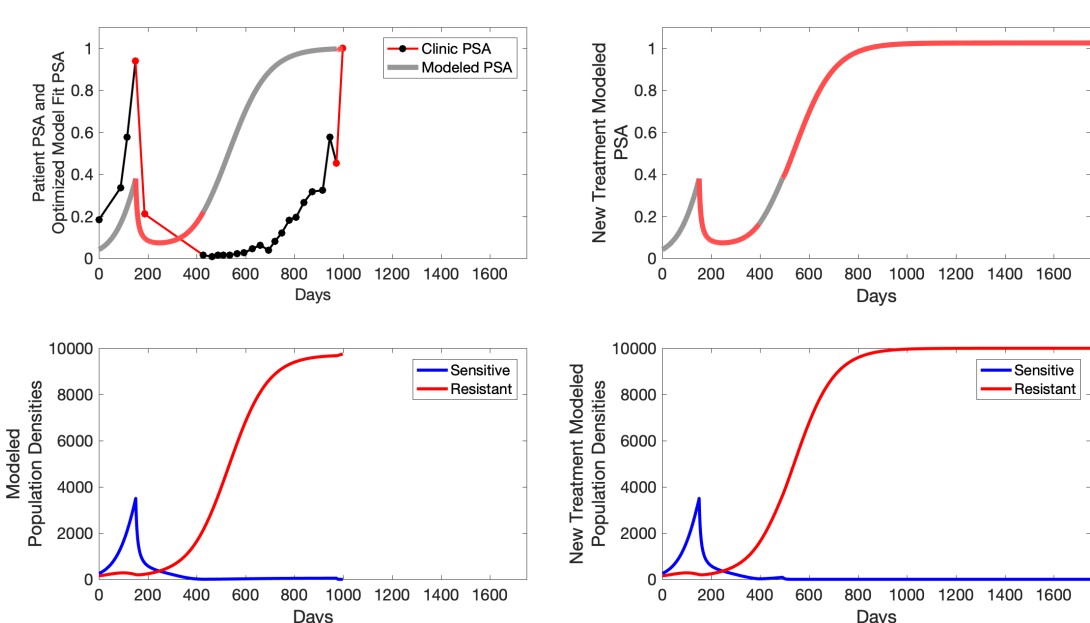

**Appendix 6—figure 22.** The left column shows PSA changes during treatment and associated model fit for each captioned patient (upper panel) as well as computer simulations for the population dynamics of the resistant and sensitive cells; the right column shows computer simulation in the same patient for a typical "intermittent therapy" protocol in which cycling therapy is insituted only after an 8 month induction period of continuous MTD application of abiraterone.

P1009

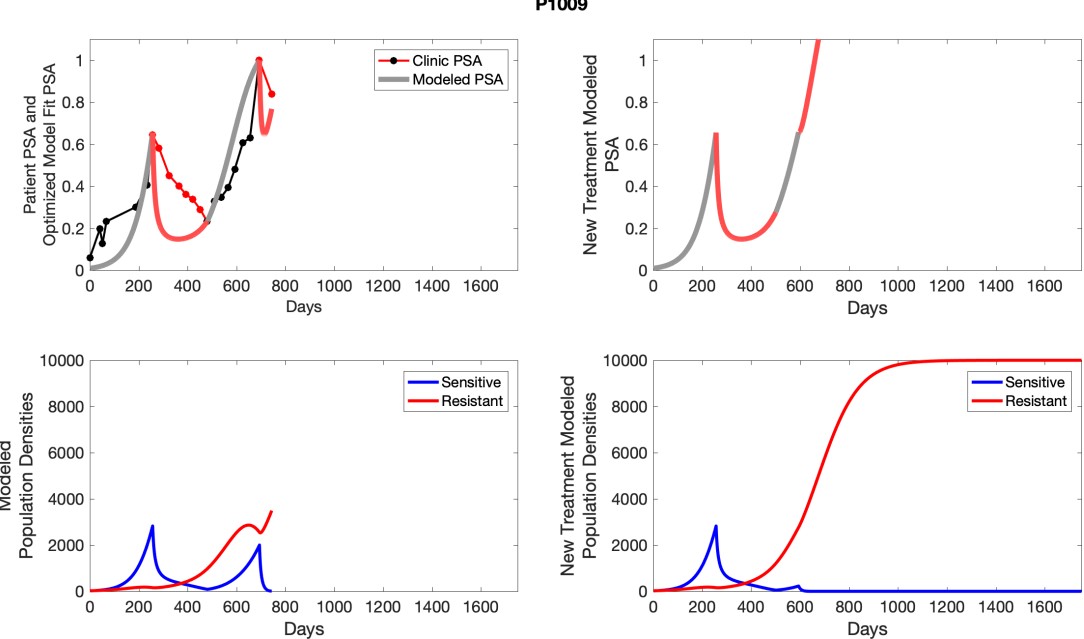

**Appendix 6—figure 23.** The left column shows PSA changes during treatment and associated model fit for each captioned patient (upper panel) as well as computer simulations for the population dynamics of the resistant and sensitive cells; the right column shows computer simulation in the same patient for a typical "intermittent therapy" protocol in which cycling therapy is insituted only after an 8 month induction period of continuous MTD application of abiraterone.

P1010

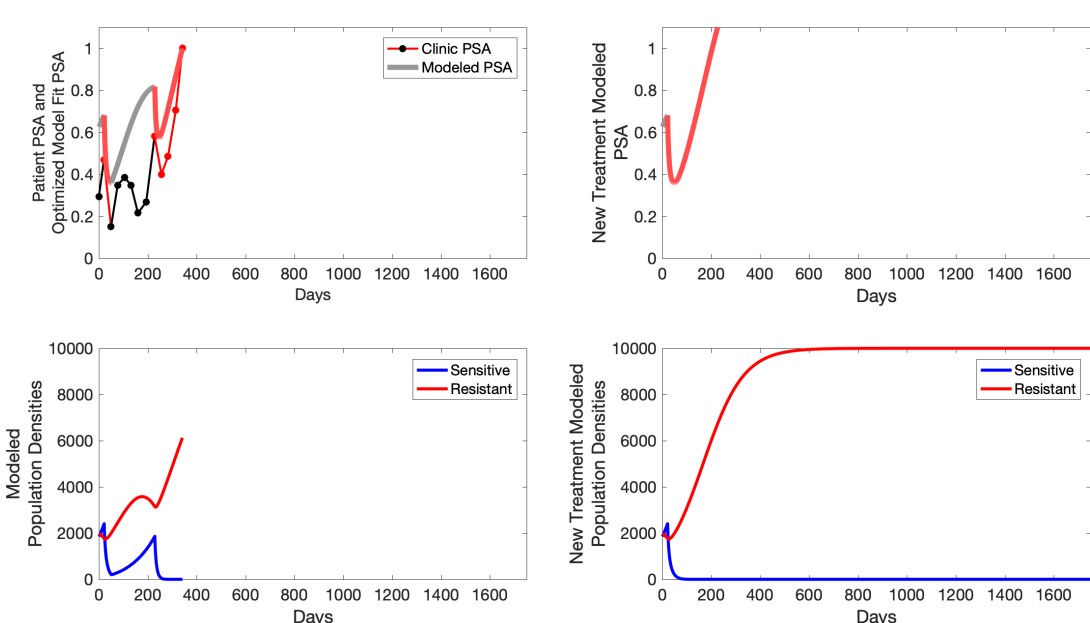

**Appendix 6—figure 24.** The left column shows PSA changes during treatment and associated model fit for each captioned patient (upper panel) as well as computer simulations for the population dynamics of the resistant and sensitive cells; the right column shows computer simulation in the same patient for a typical "intermittent therapy" protocol in which cycling therapy is insituted only after an 8 month induction period of continuous MTD application of abiraterone.

P1011

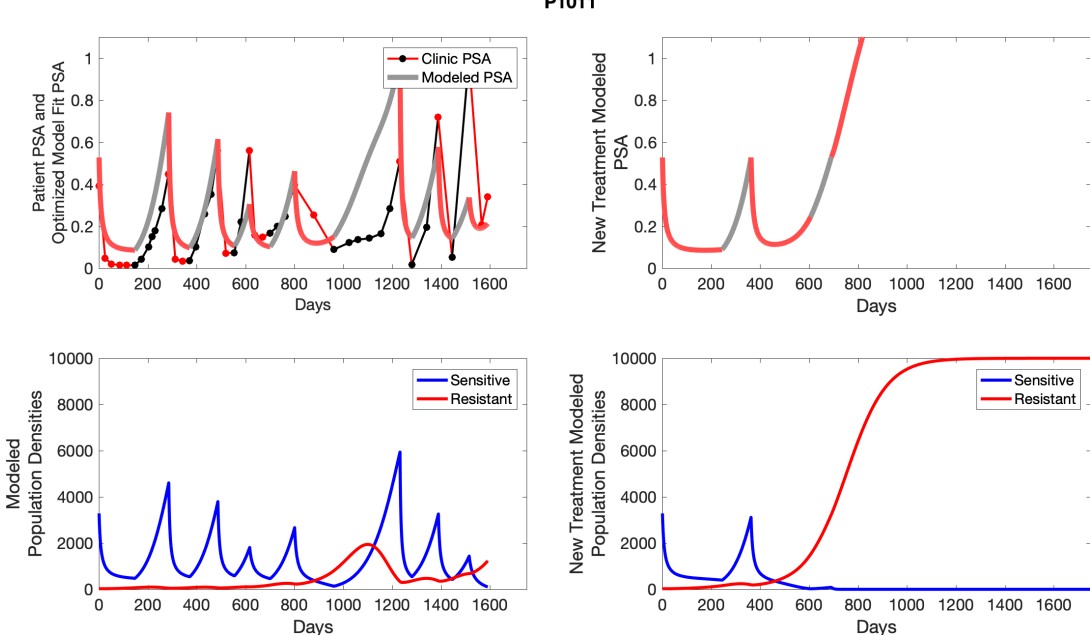

**Appendix 6—figure 25.** The left column shows PSA changes during treatment and associated model fit for each captioned patient (upper panel) as well as computer simulations for the population dynamics of the resistant and sensitive cells; the right column shows computer simulation in the same patient for a typical "intermittent therapy" protocol in which cycling therapy is insituted only after an 8 month induction period of continuous MTD application of abiraterone.

P1012

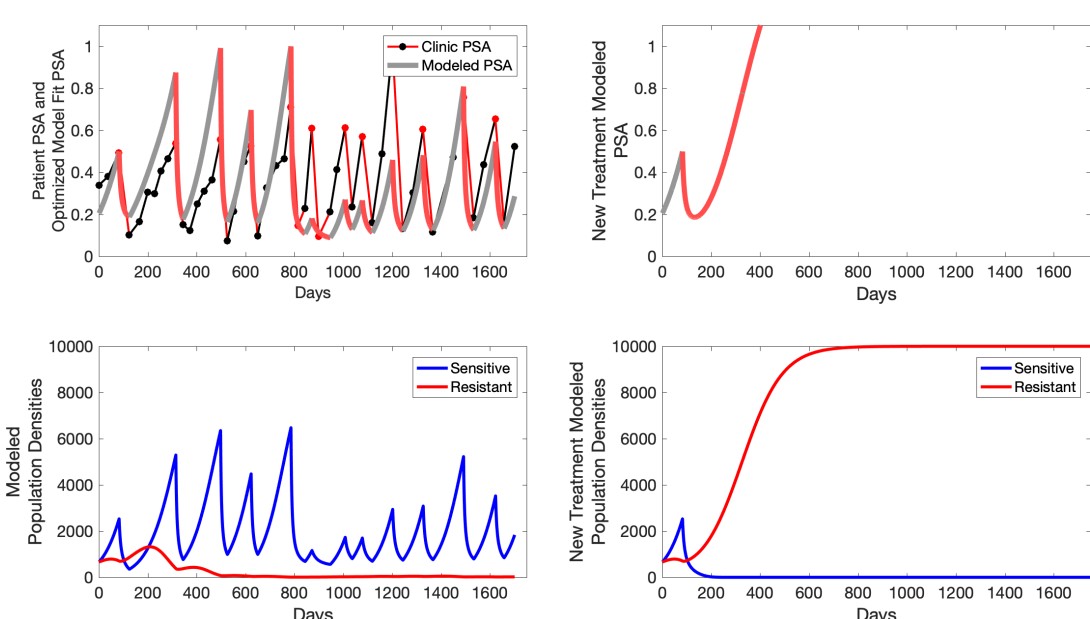

**Appendix 6—figure 26.** The left column shows PSA changes during treatment and associated model fit for each captioned patient (upper panel) as well as computer simulations for the population dynamics of the resistant and sensitive cells; the right column shows computer simulation in the same patient for a typical "intermittent therapy" protocol in which cycling therapy is insituted only after an 8 month induction period of continuous MTD application of abiraterone.

P1014

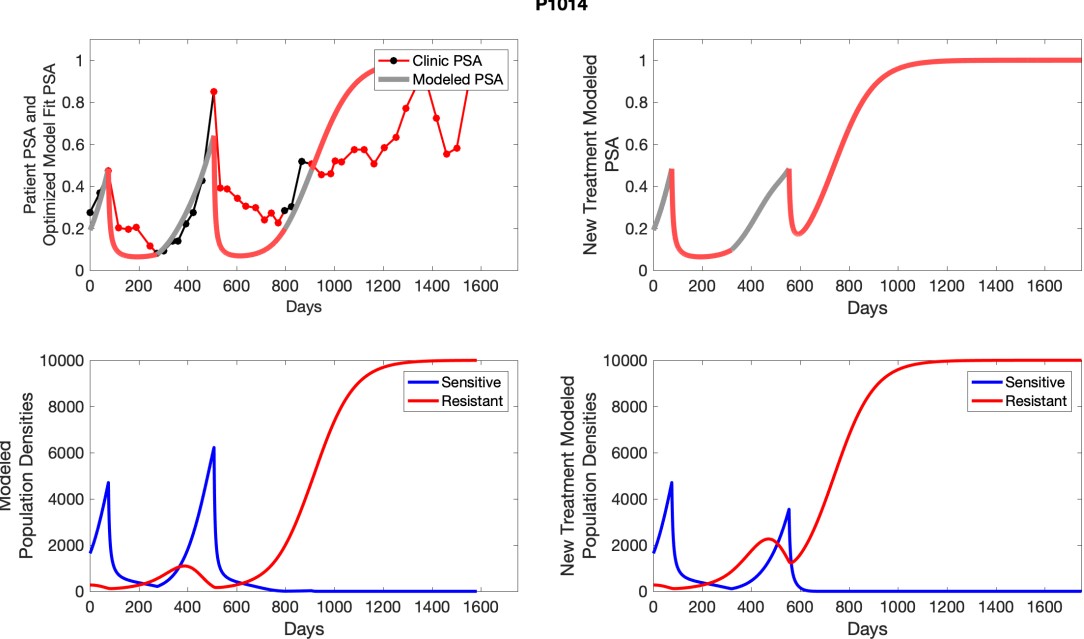

**Appendix 6—figure 27.** The left column shows PSA changes during treatment and associated model fit for each captioned patient (upper panel) as well as computer simulations for the population dynamics of the resistant and sensitive cells; the right column shows computer simulation in the same patient for a typical "intermittent therapy" protocol in which cycling therapy is insituted only after an 8 month induction period of continuous MTD application of abiraterone.

**P1015**

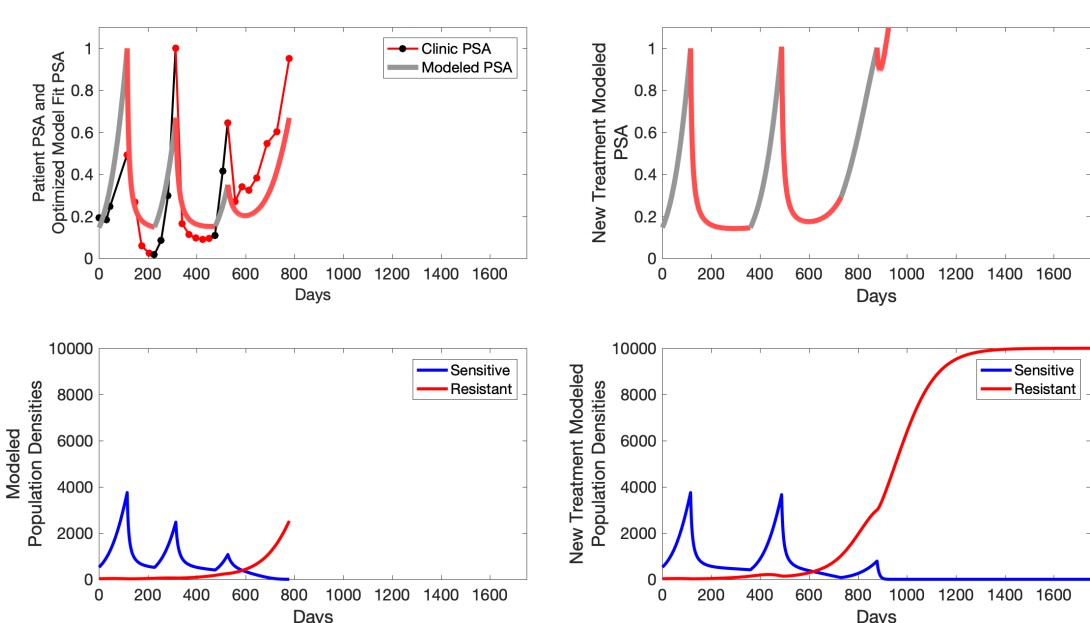

**Appendix 6—figure 28.** The left column shows PSA changes during treatment and associated model fit for each captioned patient (upper panel) as well as computer simulations for the population dynamics of the resistant and sensitive cells; the right column shows computer simulation in the same patient for a typical "intermittent therapy" protocol in which cycling therapy is insituted only after an 8 month induction period of continuous MTD application of abiraterone.

**P1016**

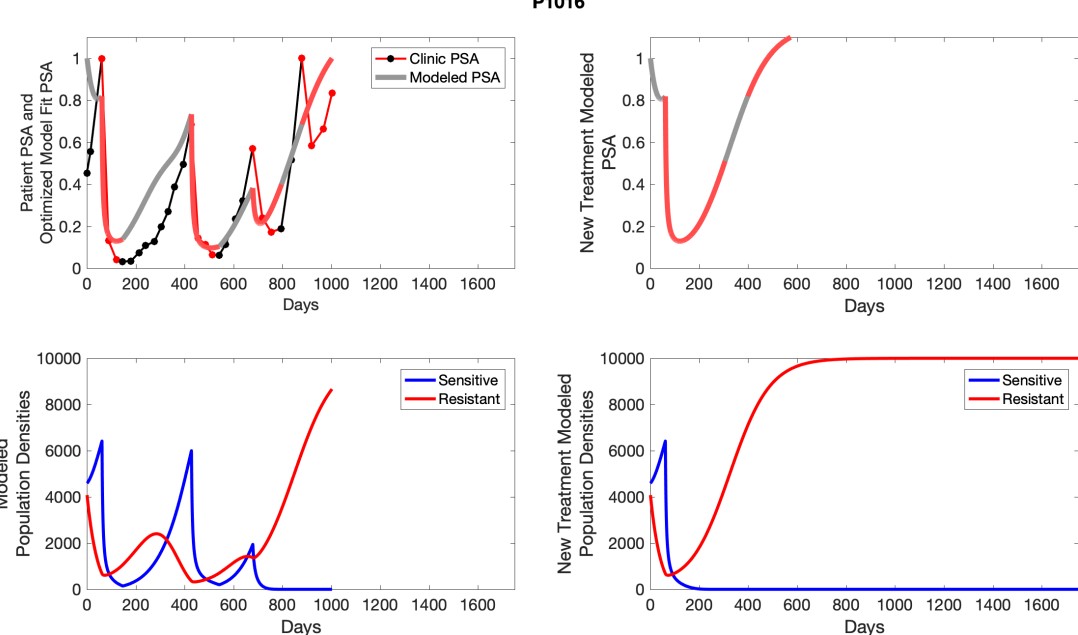

**Appendix 6—figure 29.** The left column shows PSA changes during treatment and associated model fit for each captioned patient (upper right panel) as well as computer simulations for the population dynamics of the resistant and sensitive cells; the right column shows computer simulation in the same patient for a typical "intermittent therapy" protocol in which cycling therapy is insituted only after an 8 month induction period of continuous MTD application of abiraterone.

P1017

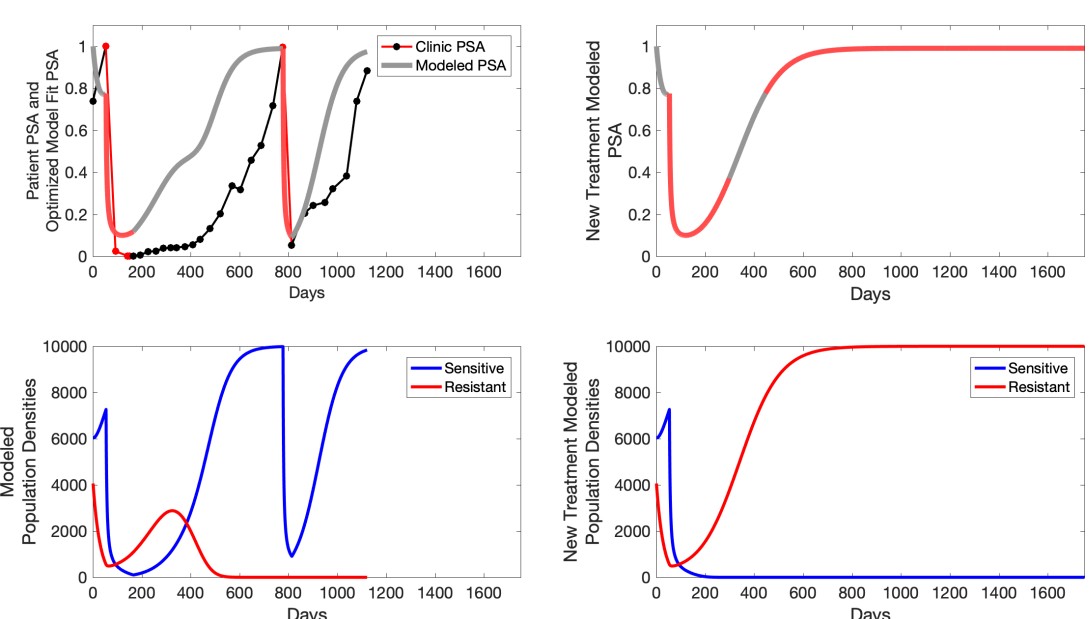

**Appendix 6—figure 30.** The left column shows PSA changes during treatment and associated model fit for each captioned patient (upper panel) as well as computer simulations for the population dynamics of the resistant and sensitive cells; the right column shows computer simulation in the same patient for a typical "intermittent therapy" protocol in which cycling therapy is insituted only after an 8 month induction period of continuous MTD application of abiraterone.

P1018

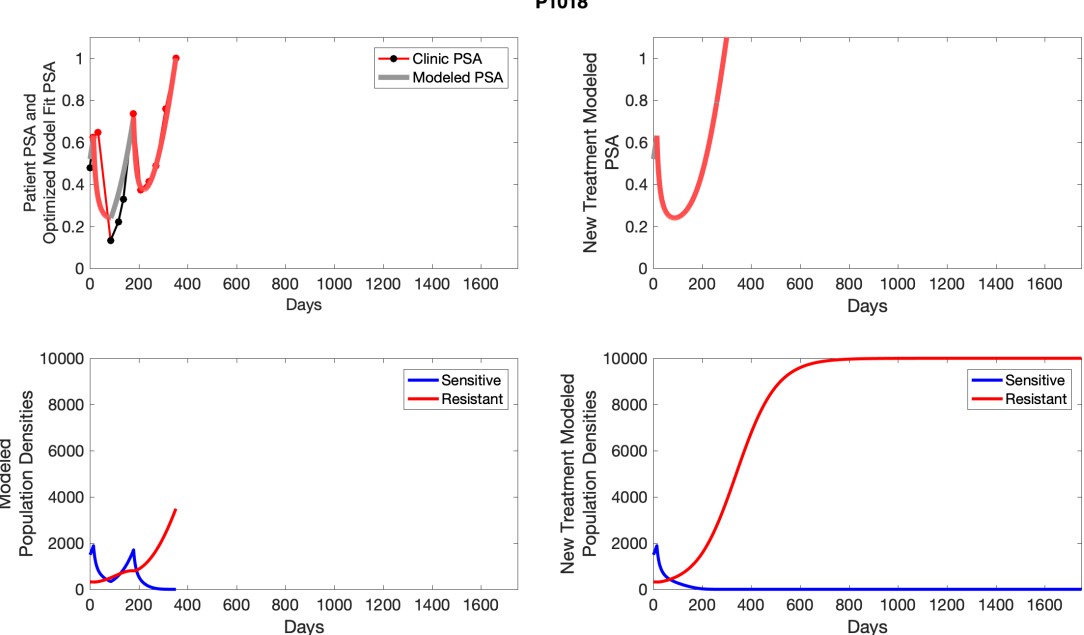

**Appendix 6—figure 31.** The left column shows PSA changes during treatment and associated model fit for each captioned patient (upper panel) as well as computer simulations for the population dynamics of the resistant and sensitive cells; the right column shows computer simulation in the same patient for a typical "intermittent therapy" protocol in which cycling therapy is insituted only after an 8 month induction period of continuous MTD application of abiraterone.

**P1020**

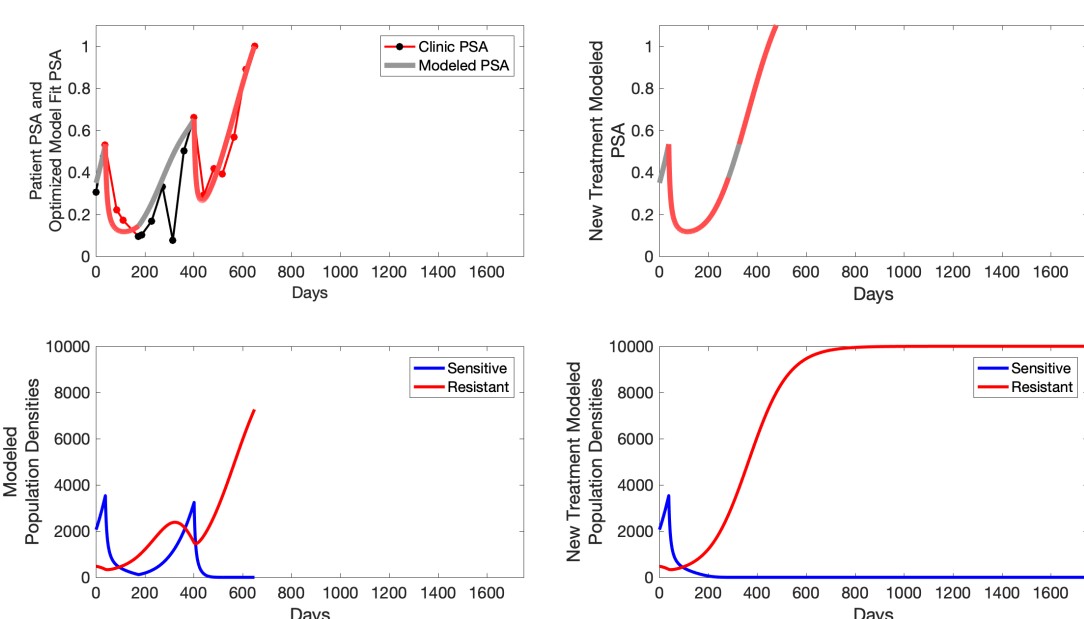

**Appendix 6—figure 32.** The left column shows PSA changes during treatment and associated model fit for each captioned patient (upper panel) as well as computer simulations for the population dynamics of the resistant and sensitive cells; the right column shows computer simulation in the same patient for a typical "intermittent therapy" protocol in which cycling therapy is insituted only after an 8 month induction period of continuous MTD application of abiraterone.

