## [Editor Report]

Zhang et al. use evolution-guided mathematical models to guide the timing and dosing of abiraterone treatment in castrate-resistant prostate cancer. While the sample size is limited, the implications of the study outcome are broad and compelling, and the article importantly highlights the transformative potential of deeply interdisciplinary research.

---

## [Decision Letter]

**Decision letter after peer review:**

Thank you for submitting your article "Evolution-based mathematical models significantly prolong response to Abiraterone in metastatic castrate resistant prostate cancer and identify strategies to further improve outcomes" for consideration by *eLife*. Your article has been reviewed by 2 peer reviewers, and the evaluation has been overseen by George Perry as the Reviewing and Senior Editor. The reviewers have opted to remain anonymous.

Essential revisions:

The reviewers are positive about your paper but requesting several points of clarification in their below reviews. Please address these points in your revision.

*Reviewer #1 (Recommendations for the authors):*

I suggest the authors clearly describe what has been published and what the difference is in the current model. It seems the mathematical model described 204-227 is the same as the original model in Zhang J et al. 2017, however, many details were missing (e.g. what are the cell types, why some parameters are set in a certain way) and just reading this part alone without reading Zhang J et al. 2017 is confusing.

Many mathematical details are missing: Line 427 says "The full details of the mathematical model are presented in Supplementary Mathematics section below." While there is nothing under the Supplementary Mathematics section on the last page. Line 246-249 only vaguely say there are patient-specific parameters and there are shared ones. However, there are important assumptions about the model that should be introduced more clearly. I found out from the supplement that competitive coefficients are set as the same across all patients, while the authors also showed that these are critical to determining outcomes. The authors should provide more explanations or analysis about why is alright to set the same if infer in ADT more successful patients and not successful patients if their values are similar.

The authors should give more explanations about how to understand the figures. The figure 3a is for a patient with an initial resistant fraction of 0.03 but based on the density value for the red line and blue line in the bottom panel of figure 3a, it doesn't seem like the fraction is 0.03. I have the same question for all such figures in the paper. And figure legend doesn't really match the plot for the top panel of figure 3a, does the red color indicate treatment? And line style indicates real/modeled?

I also suggest the authors illustrate more of how the clinical guidance can be derived from the current analysis. Eg. "include more rapid withdrawal of therapy when PSA crosses the 50% threshold". Some simulations showing the impact of withdrawal at the different thresholds will be useful.

The manuscript has not talked about Figure 4 (I didn't find where it is referenced).

Analysis code should be available for review.

*Reviewer #2 (Recommendations for the authors):*

There is a significantly longer median time to progression, 33 months in the patients that received adaptive therapy group compared to 14.3 months in the SOC cohort, and OS was 58.5 months compared to 31.3 months. It would be very informative to highlight what percentage of mCRPC patients could benefit from evolution-informed treatments, considering those that have <50% PSA decline after the first arbiterone therapy (statistically), those that progress while on arbiterone treatment, and those that have evolutionary dynamics that violate the assumptions of the model.

The mathematical model is clearly described. As it takes a significant portion of the Results section, it would be helpful for the reader to highlight the changes and advantages compared to the previously published model in 2017, Nat Comms.

In addition, while the manuscript is clearly written, it would help a lot for the reader if the abbreviations such as TTP in the abstract, TP, T+, and T- are described in their first use (Line 206 or Abstract), without the need to read previous work from the authors.

I couldn't find the information on how many patients didn't undergo castration or a table with a summary of the clinical features for each patient, ideally along with PSA levels, duration of arbiterone treatment, and the number of arbiterone treatments.

The results show no statistically significant difference between the two patient groups (SOC and adaptive therapy group) in terms of tumour stage, Gleason score, and pretreatment PSA.

Unfortunately, the sample size wouldn't allow for multivariate analysis. What about based on whether patients underwent castration-resistant therapy?

The SOC cohort had prior therapy history (Line 407), but it is not clear from the text whether patients from the adaptive therapy cohort also had prior therapy history.

Some figures in the supplementary would need a higher resolution

Is there a difference in the competition coefficient ratios between SOC and adaptive therapy cohorts, at the start of treatment and at progression?

---

## [Author Response]

Essential revisions:The reviewers are positive about your paper but requesting several points of clarification in their below reviews. Please address these points in your revision.Reviewer #1 (Recommendations for the authors):I suggest the authors clearly describe what has been published and what the difference is in the current model. It seems the mathematical model described 204-227 is the same as the original model in Zhang J et al. 2017, however, many details were missing (e.g. what are the cell types, why some parameters are set in a certain way) and just reading this part alone without reading Zhang J et al. 2017 is confusing.

Thank you for pointing this out. As noted in our response to reviewer 2, we have added a full paragraph explaining the tumor subtypes and the reasoning behind the post treatment model simulations.

Many mathematical details are missing: Line 427 says "The full details of the mathematical model are presented in Supplementary Mathematics section below." While there is nothing under the Supplementary Mathematics section on the last page. Line 246-249 only vaguely say there are patient-specific parameters and there are shared ones. However, there are important assumptions about the model that should be introduced more clearly. I found out from the supplement that competitive coefficients are set as the same across all patients, while the authors also showed that these are critical to determining outcomes. The authors should provide more explanations or analysis about why is alright to set the same if infer in ADT more successful patients and not successful patients if their values are similar.

We have now extensively revised the mathematics section to provide more details about the methods. The reviewer makes a very good point that key parameters in the model may differ among individuals and a new generation of clinical biomarkers to provide these estimates will ultimately be necessary to a priori predict patient-specific treatment optimization. We have added 3 sentences to the Discussion section to make this point explicit and summarize:

“Thus, ideally, every cancer patient should have a unique mathematical model that is continuously updated throughout the treatment arc – similar to, for example, the models and computer simulations used to track storms.”

The authors should give more explanations about how to understand the figures. The figure 3a is for a patient with an initial resistant fraction of 0.03 but based on the density value for the red line and blue line in the bottom panel of figure 3a, it doesn't seem like the fraction is 0.03. I have the same question for all such figures in the paper. And figure legend doesn't really match the plot for the top panel of figure 3a, does the red color indicate treatment? And line style indicates real/modeled?

Thank you, these are all good points. Figure 3 is confusing and the “0.03” was a typo. To clarify this, we have added data on model-estimated pre-treatment fraction in Figure 2 and we have added a more extensive discussion of this in the methods section as well as explanatory text to the figure captions. In Figures 4 through 6, we have added more explanatory text in the figure caption to clarify each of the lines in the two panels.

I also suggest the authors illustrate more of how the clinical guidance can be derived from the current analysis. Eg. "include more rapid withdrawal of therapy when PSA crosses the 50% threshold". Some simulations showing the impact of withdrawal at the different thresholds will be useful.

This is an interesting comment because the answer is complicated and one that we have internally discussed extensively. The brief answer is that, in general, the best results are obtained by shorter duration of treatment. That is, if we stop treatment when the PSA reaches 80% of the pre-treatment value, tumor control is maintained for longer than if we use a 50% threshold. We have added a new figure (Figure 7) and text that demonstrates this. Furthermore, in pre-clinical experiments, we found the best outcomes are obtained when we maintain the tumor volume stable by continuously adjusting the dose of drug applied (1) – this is also now briefly discussed. However, in designing this first trial using evolution-base treatment, it was felt the most clinically practical approach was using the 50% threshold since the other approaches required more frequent PSA monitoring and clinical appointments. We have added some text in the discussion to make note of this.

The manuscript has not talked about Figure 4 (I didn't find where it is referenced).

Thank you, that has been added

Analysis code should be available for review.

Thank you, we have added that the code is available upon request. We have also added submitted the code and anonymized clinical data to Code Ocean and will activate the site upon acceptance of the paper.

Reviewer #2 (Recommendations for the authors):There is a significantly longer median time to progression, 33 months in the patients that received adaptive therapy group compared to 14.3 months in the SOC cohort, and OS was 58.5 months compared to 31.3 months. It would be very informative to highlight what percentage of mCRPC patients could benefit from evolution-informed treatments, considering those that have <50% PSA decline after the first arbiterone therapy (statistically), those that progress while on arbiterone treatment, and those that have evolutionary dynamics that violate the assumptions of the model.

Thank you. We have added a panel to Figure 3 that shows time to radiographic progression plotted against the mathematically estimated pre-treatment fraction of resistant cells. This demonstrates that, at all ranges of this fraction, the patients on evolution-based treatment had a longer TTP than those receiving standard of care MTD dosing.

The mathematical model is clearly described. As it takes a significant portion of the Results section, it would be helpful for the reader to highlight the changes and advantages compared to the previously published model in 2017, Nat Comms.

A more detailed explanation of the math model has been added to the text (see response to Reviewer 1). We have also referenced more technical presentation of the model that has been published in mathematical biology journals for the mathematically-minded readers.

In addition, while the manuscript is clearly written, it would help a lot for the reader if the abbreviations such as TTP in the abstract, TP, T+, and T- are described in their first use (Line 206 or Abstract), without the need to read previous work from the authors.

Thank you for pointing this out. As above, we have added the definition of TTP to the abstract and manuscript text. We have added a full paragraph to better define the TP, T+, and T- subpopulations and their eco-evolutionary interactions.

I couldn't find the information on how many patients didn't undergo castration or a table with a summary of the clinical features for each patient, ideally along with PSA levels, duration of arbiterone treatment, and the number of arbiterone treatments.

The PSA data throughout the course of treatment is provided for each patient in both cohorts in the Supplemental Figures

The results show no statistically significant difference between the two patient groups (SOC and adaptive therapy group) in terms of tumour stage, Gleason score, and pretreatment PSA.Unfortunately, the sample size wouldn't allow for multivariate analysis. What about based on whether patients underwent castration-resistant therapy?

We have included this in the new Table 1. Only Sipuleucel-T had been administered in the castrate resistant setting. 5 patients in the control and 6 patients in the adaptive group had received this treatment.

The SOC cohort had prior therapy history (Line 407), but it is not clear from the text whether patients from the adaptive therapy cohort also had prior therapy history.

This has now been added in Table 1.

Some figures in the supplementary would need a higher resolution

Thank you for point this out. In our initial submission, we submitted lower resolution images due to limitations in the file sizes. We have now submitted the higher resolution images

Is there a difference in the competition coefficient ratios between SOC and adaptive therapy cohorts, at the start of treatment and at progression?

This is a good point and the ratio almost certainly varies somewhat among patients. Please see our response above. Furthermore, it is possible the ratio changed with time during treatment. However, the mathematical methods used to estimate required a global approach using all patients in both cohorts. We have added text to make this clear and used this as an argument to justify additional measurement of evolution-specific data during subsequent trials.

1. Enriquez-Navas PM, Kam Y, Das T, Hassan S, Silva A, Foroutan P, Ruiz E, Martinez G, Minton S, Gillies RJ, Gatenby RA. Exploiting evolutionary principles to prolong tumor control in preclinical models of breast cancer. Sci Transl Med. 2016;8(327):327ra24. Epub 2016/02/26. doi: 10.1126/scitranslmed.aad7842. PubMed PMID: 26912903; PMCID: PMC4962860.